# First-Order Methods for Linearly Constrained Bilevel Optimization

**Guy Kornowski**[†]    **Swati Padmanabhan**[‡]    **Kai Wang**[§]    **Jimmy Zhang**[¶]    **Suvrit Sra**[||]

## Abstract

Algorithms for bilevel optimization often encounter Hessian computations, which are prohibitive in high dimensions. While recent works offer first-order methods for unconstrained bilevel problems, the *constrained* setting remains relatively underexplored. We present first-order linearly constrained optimization methods with finite-time hypergradient stationarity guarantees. For linear *equality* constraints, we attain $\epsilon$-stationarity in $\widetilde{O}(\epsilon^{-2})$ gradient oracle calls, which is nearly-optimal. For linear *inequality* constraints, we attain $(\delta, \epsilon)$-Goldstein stationarity in $\widetilde{O}(d\delta^{-1}\epsilon^{-3})$ gradient oracle calls, where $d$ is the upper-level dimension. Finally, we obtain for the linear inequality setting dimension-free rates of $\widetilde{O}(\delta^{-1}\epsilon^{-4})$ oracle complexity under the additional assumption of oracle access to the optimal dual variable. Along the way, we develop new nonsmooth nonconvex optimization methods with inexact oracles. Our numerical experiments verify these guarantees.

## 1   Introduction

Bilevel optimization [1–4], an important problem in optimization, is defined as follows:

$$\text{minimize}_{x \in \mathcal{X}} \ F(x) \coloneqq f(x, y^*(x)) \ \text{subject to} \ y^*(x) \in \arg\min_{y \in S(x)} g(x, y). \tag{1.1}$$

Here, the value of the upper-level problem at any point $x$ depends on the solution of the lower-level problem. This framework has recently found numerous applications in meta-learning [5–8], hyperparameter optimization [9–11], and reinforcement learning [12–15]. Its growing importance has spurred increasing efforts towards designing computationally efficient algorithms for it.

As demonstrated by [16], a key computational step in algorithms for bilevel optimization is estimating $dy^*(x)/dx$, the gradient of the lower-level solution. This gradient estimation problem has been extensively studied in differentiable optimization [17, 18] by applying the implicit function theorem to the KKT system of the given problem [19–24]. However, this technique typically entails computing (or estimating) second-order derivatives, which can be prohibitive in high dimensions [25–27].

Recently, [28] made a big leap forward towards addressing this computational bottleneck. Restricting themselves to the class of unconstrained bilevel optimization, they proposed a fully first-order method with finite-time stationarity guarantees. While a remarkable breakthrough, [28] does not directly extend to the important setting of *constrained* bilevel optimization. This motivates the question:

*Can we develop a first-order algorithm for constrained bilevel optimization?*

[1]Full version: https://arxiv.org/abs/2412.12771.  GK, SP, KW, ZZ contributed equally; authors ordered alphabetically.

[†]Weizmann Institute of Science. `guy.kornowski@weizmann.ac.il`

[‡]Massachusetts Institute of Technology. `pswt@mit.edu`

[§]Georgia Institute of Technology. `kwang692@gatech.edu`

[¶]Purdue University. `zhan5111@purdue.edu`

[||]Massachusetts Institute of Technology. `suvrit@mit.edu`

38th Conference on Neural Information Processing Systems (NeurIPS 2024).

Besides being natural from the viewpoint of complexity theory, this question is well-grounded in applications such as mechanism design [29, 30], resource allocation [31–34], and decision-making under uncertainty [20, 35, 36]. Our primary contribution is *an affirmative answer to the highlighted question for bilevel programs with linear constraints*, an important problem class often arising in adversarial training, decentralized meta learning, and sensor networks (see [37]). While there have been some other recent works [37–39] on this problem, our work is first-order (as opposed to [37]) and offers, in our view, a stronger guarantee on stationarity (compared to [38, 39])— cf. Section 1.2.

## 1.1 Our contributions

We provide first-order algorithms (with associated finite-time convergence guarantees) for linearly constrained bilevel programs (Problem 1.1). By "first-order", we mean that we use only zeroth and first-order oracle access to $f$ and $g$. Our assumptions for each of our contributions are in Section 2.1.

**(1) Linear equality constraints.** As our first contribution, we design first-order algorithms for solving Problem 1.1 where the lower-level constraint set $S(x) := \{y : Ax - By - b = 0\}$ comprises linear equality constraints, and $\mathcal{X}$ a convex compact set. With appropriate regularity assumptions on $f$ and $g$, we show in this case smoothness in $x$ of the hyperobjective $F$. Inspired by ideas from Kwon et al. [40], we use implicit differentiation of the KKT matrix of a slightly perturbed version of the lower-level problem to design a first-order approximation to $\nabla F$. Constructing our first-order approximation entails solving a strongly convex optimization problem on affine constraints, which can be done efficiently. With this inexact gradient oracle in hand, we then run projected gradient descent, which converges in $\widetilde{O}(\epsilon^{-2})$ iterations for smooth functions.

**Theorem 1.1** (Informal; cf. Theorem 3.1). *Given Problem 1.1 with linear equality constraints $S(x) := \{y : Ax - By - b = 0\}$ and $\mathcal{X}$ a convex compact set, under regularity assumptions on $f$ and $g$ (Assumptions 2.2 and 2.3), there exists an algorithm, which in $\widetilde{O}(\epsilon^{-2})$ oracle calls to $f$ and $g$, converges to an $\epsilon$-stationary point of $F$.*

For linear equality constrained bilevel optimization, this is the first first-order result attaining $\epsilon$-stationarity of $F$ with assumptions solely on the constituent functions $f$ and $g$ and none on $F$ — cf. Section 1.2 for a discussion of the results of Khanduri et al. [37] for this setting.

**(2) Linear inequality constraints.** Next, we provide first-order algorithms for solving Problem 1.1 where the lower-level constraint set $S(x) := \{y : Ax - By - b \leq 0\}$ comprises linear inequality constraints, and the upper-level variable is unconstrained.

Our measure of convergence of algorithms in this case is that of $(\delta, \epsilon)$-stationarity [41]: for a Lipschitz function, we say that a point $x$ is $(\delta, \epsilon)$-stationary if within a $\delta$-ball around $x$ there exists a convex combination of subgradients of the function with norm at most $\epsilon$ (cf. Definition 2.1).

To motivate this notion of convergence, we note that the hyperobjective $F$ (in Problem 1.1) as a function of $x$ could be nonsmooth and nonconvex (and Lipschitz, as we later prove). Minimizing such a function in general is known to be intractable [42], necessitating local notions of stationarity. Indeed, not only is it impossible to attain $\epsilon$-stationarity in finite time [43], even getting *near* an approximate stationary point of an arbitrary Lipschitz function is impossible unless the number of queries is exponential in the dimension [44]. Consequently, for this function class, $(\delta, \epsilon)$-stationarity has recently emerged [43] to be a natural and algorithmically tractable notion of stationarity. We give the following guarantee under regularity assumptions on $f$ and $g$.

**Theorem 1.2** (Informal; Theorem 4.1). *Consider Problem 1.1 with linear inequality constraints $S(x) := \{y : Ax - By - b \leq 0\}$. Under mild assumptions on $f$ and $g$ (Assumption 2.2) and the lower-level primal solution $y^*$ (Assumption 2.4), there exists an algorithm, which converges to a $(\delta, \epsilon)$-stationary point of $F$ in $\widetilde{O}(d\delta^{-1}\epsilon^{-3})$ oracle calls to $f$ and $g$, where $d$ is the upper-level variable dimension.*

To the best of our knowledge, this is the first result to offer a first-order finite-time stationarity guarantee on the hyperobjective for linear inequality constrained bilevel optimization (cf. Section 1.2 for a discussion of related work [37–39]). We obtain our guarantee in Theorem 1.2 by first invoking a result by Zhang and Lan [45] to obtain inexact hyperobjective values of $F$ using only $\widetilde{O}(1)$ oracle calls to $f$ and $g$. We also show (Lemma 4.3) that this hyperobjective $F$ is Lipschitz. We then employ our inexact zeroth-order oracle for $F$ in Algorithm 2 designed to minimize Lipschitz nonsmooth nonconvex functions (in particular, $F$), with the following convergence guarantee.

**Theorem 1.3** (see Theorem C.1). *Given L-Lipschitz $F : \mathbb{R}^d \to \mathbb{R}$ and $|\widetilde{F}(\cdot) - F(\cdot)| \leq \epsilon$, there exists an algorithm, which, in $\widetilde{O}(d\delta^{-1}\epsilon^{-3})$ calls to $\widetilde{F}(\cdot)$, outputs $x^{\text{out}}$ with $\mathbb{E}[\text{dist}(0, \partial_\delta F(x^{\text{out}}))] \leq 2\epsilon$.*

While such algorithms using *exact* zeroth-order access already exist [46], extending them to the inexact gradient setting is non-trivial; we leverage recent ideas connecting online learning to nonsmooth nonconvex optimization by Cutkosky, Mehta, and Orabona [47] (cf. Section 4).

**(3) Linear inequality under assumptions on dual variable access.** For the inequality setting (i.e., Problem 1.1 with the lower-level constraint set $S(x) := \{y : Ax - By - b \leq 0\}$), we obtain dimension-free rates under an additional assumption (Assumption 2.5) on oracle access to the optimal dual variable $\lambda^*$ of the lower-level problem. We are not aware of a method to obtain this dual variable in a first-order fashion (though in practice, highly accurate approximations to $\lambda^*$ are readily available), hence the need for imposing this assumption. We believe that removing this assumption and obtaining dimension-free first-order rates in this setting would be an important direction for future work. Our guarantee for this setting is summarized below.

**Theorem 1.4** (Informal; Theorem 4.4 combined with Theorem 5.3). *Consider Problem 1.1 with linear inequality constraints $S(x) := \{y : Ax - By - b \leq 0\}$ and unconstrained upper-level variable. Under mild regularity assumptions on $f$ and $g$ (Assumption 2.2), on $y^*$ (Assumption 2.4), and assuming oracle access to the optimal dual variable $\lambda^*$ (Assumption 2.5), there exists an algorithm, which in $\widetilde{O}(\delta^{-1}\epsilon^{-4})$ oracle calls to $f$ and $g$ converges to a $(\delta, \epsilon)$-stationary point for $F$.*

We obtain this result by first reformulating Problem 4.1 via the penalty method and constructing an inexact gradient oracle for the hyperobjective $F$ (cf. Section 5). We then employ this inexact gradient oracle within an algorithm (Algorithm 3) designed to minimize Lipschitz nonsmooth nonconvex functions (in particular, $F$), with the following convergence guarantee.

**Theorem 1.5** (Informal; Theorem 4.4). *Given Lipschitz $F : \mathbb{R}^d \to \mathbb{R}$ and $\|\widetilde{\nabla}F(\cdot) - \nabla F(\cdot)\| \leq \epsilon$, there exists an algorithm that, in $T = O(\delta^{-1}\epsilon^{-3})$ calls to $\widetilde{\nabla}F$, outputs a $(\delta, 2\epsilon)$-stationary point of $F$.*

Our Algorithm 3 is essentially a "first-order" version of Algorithm 2. Similar to Algorithm 2, despite the existence of algorithms with these guarantees with access to exact gradients [48], their extensions to the *inexact* gradient setting are not trivial and also make use of the new framework of Cutkosky, Mehta, and Orabona [47]. We believe our analysis for this general task can be of independent interest to the broader optimization community. Lastly, we also use a more implementation-friendly variant of Algorithm 3 (with slightly worse theoretical guarantees) in numerical experiments.

## 1.2 Related work

The vast body of work on asymptotic results for bilevel programming, starting with classical works such as Anandalingam and White [49], Ishizuka and Aiyoshi [50], White and Anandalingam [51], Vicente, Savard, and Júdice [52], Zhu [53], and Ye and Zhu [54], typically fall into two categories: those based on approximate implicit differentiation: Amos and Kolter [17], Agrawal et al. [18], Domke [55], Pedregosa [56], Gould et al. [57], Liao et al. [58], Grazzi et al. [59], and Lorraine, Vicol, and Duvenaud [60] and those via iterative differentiation: Franceschi et al. [9], Shaban et al. [10], Domke [55], Grazzi et al. [59], Maclaurin, Duvenaud, and Adams [61], and Franceschi et al. [62]. Another recent line of work in this category includes Khanduri et al. [37], Liu et al. [63], Ye et al. [64], and Gao et al. [65], which use various smoothing techniques.

The first non-asymptotic result for bilevel programming was provided by Ghadimi and Wang [16], which was followed by a flurry of work: for example, algorithms that are single-loop stochastic: Chen, Sun, and Yin [66], Chen et al. [67], and Hong et al. [68], projection-free: Akhtar et al. [69], Jiang et al. [70], Abolfazli et al. [71], and Cao et al. [72], use variance-reduction and momentum: Khanduri et al. [73], Guo et al. [74], Yang, Ji, and Liang [75], and Dagréou et al. [76], those for single-variable bilevel programs: Jiang et al. [70], Sabach and Shtern [77], Amini and Yousefian [78, 79], and Merchav and Sabach [80], and for bilevel programs with special constraints: Khanduri et al. [37], Abolfazli et al. [71], Tsaknakis, Khanduri, and Hong [81], and Xu and Zhu [82].

The most direct predecessors of our work are those by Khanduri et al. [37], Yao et al. [38], Lu and Mei [39], Kwon et al. [40], and Liu et al. [63]. As alluded to earlier, Liu et al. [28] recently made a significant contribution by providing for bilevel programming a fully first-order algorithm with finite-time stationarity guarantees. This was extended to the stochastic setting by Kwon et al. [40]

(which we build upon), simplified and improved by Chen, Ma, and Zhang [83], and extended to the constrained setting by Khanduri et al. [37], Yao et al. [38], and Lu and Mei [39].

The works of Yao et al. [38] and Lu and Mei [39] study the more general problem of bilevel programming with general convex constraints. However, they use KKT stationarity as a proxy to the hypergradient stationarity. Our Theorem 1.4 is restricted to linear inequality constraints, we provide stationarity guarantees directly in terms of the objective of interest. Moreover, Yao et al. [38] assumes joint convexity of the lower-level constraints in upper and lower variables to allow for efficient projections, while we require convexity only in the lower-level variable.

The current best result for the linearly constrained setting is that of Khanduri et al. [37]. However, this work requires Hessian computations (and is therefore not fully first-order). Moreover, Khanduri et al. [37] imposes strong regularity assumptions on the hyperobjective $F$, which are, in general, impossible to verify. In contrast, Theorem 1.1 imposes *assumptions solely on the constituent functions $f$ and $g$*, none directly on $F$, thus making substantial progress on these two fronts.

## 2 Preliminaries

We follow standard notation (see Appendix A), with only the following crucial definition stated here.

**Definition 2.1.** *Consider a locally Lipschitz function $f : \mathbb{R}^d \to \mathbb{R}$, a point $x \in \mathbb{R}^d$, and a parameter $\delta > 0$. The Goldstein subdifferential [41] of $f$ at $x$ is the set $\partial_\delta f(x) := \text{conv}(\cup_{y \in \mathbb{B}_\delta(x)} \partial f(y))$, where $\partial f(x) = \text{conv}\{\lim_{n \to \infty} \nabla f(x_n) : x_n \to x, \ x_n \in \text{dom}(\nabla f)\}$ is the Clarke subdifferential [84] of $f$ and $\mathbb{B}_\delta(x)$ denotes the Euclidean ball of radius $\delta$ around $x$. A point $x$ is called $(\delta, \epsilon)$-stationary if $\text{dist}(0, \partial_\delta f(x)) \le \epsilon$, where $\text{dist}(x, S) := \inf_{y \in S} \|x - y\|$.*

### 2.1 Assumptions

We consider Problem 1.1 with linear equality constraints (Section 3) under Assumptions 2.2 and 2.3 and linear inequality constraints (Sections 4 and 5) under Assumptions 2.2, 2.4 and 2.5. We assume the upper-level (UL) variable $x \in \mathbb{R}^{d_x}$, lower-level (LL) variable $y \in \mathbb{R}^{d_y}$, and $A \in \mathbb{R}^{d_h \times d_x}$.

**Assumption 2.2.** *For Problem 1.1, we assume the following for both settings we study:*

 (i) *Upper-level: The objective $f$ is $C_f$-smooth and $L_f$-Lipschitz continuous in $(x, y)$.*
 (ii) *Lower-level: The objective $g$ is $C_g$-smooth. Fixing any $x \in \mathcal{X}$, $g(x, \cdot)$ is $\mu_g$-strongly convex.*
 (iii) *We assume that the linear independence constraint qualification (LICQ) condition holds for the LL problem at every $x$ and $y$, i.e., the constraint $h(x, y) := Ax - By - b$ has a full row rank $B$.*

**Assumption 2.3.** *For Problem 3.1 (with linear equality constraints), we additionally assume that the set $\mathcal{X}$ is convex and compact, and that the objective $g$ is $S_g$-Hessian smooth, that is, $\left\|\nabla^2 g(x, y) - \nabla^2 g(\bar{x}, \bar{y})\right\| \le S_g \left\|(x, y) - (\bar{x}, \bar{y})\right\| \forall x, \bar{x} \in \mathcal{X}$, and $y, \bar{y} \in \mathbb{R}^{d_y}$.*

**Assumption 2.4.** *For Problem 4.1 (with linear inequality constraints), we additionally assume that $y^*$ is $L_y$-Lipschitz in $x$, where $y^*$ is the LL primal solution $y^*(x), \lambda^*(x) = \arg\max_y \min_{\beta \ge 0} g(x, y) + \beta^\top h(x, y)$, where $h(x, y) := Ax - By - b$.*

**Assumption 2.5.** *We provide additional results for Problem 4.1 under additional stronger assumptions stated here: Denote the LL primal and dual solution $y^*(x), \lambda^*(x) = \arg\max_y \min_{\beta \ge 0} g(x, y) + \beta^\top h(x, y)$, where $h(x, y) := Ax - By - b$; then, we assume exact access to $\lambda^*$ and that $\|\lambda^*(x)\| \le R$.*

Assumptions 2.2(i) and 2.2(ii) are standard in bilevel optimization. Assumption 2.2(iii) is the same as the complete recourse assumption in stochastic programming [85], that is, the LL problem is feasible $y$ for every $x \in \mathbb{R}^{d_x}$. Assumption 2.3 is used only in the equality case and guarantees smoothness of $F$. Assumption 2.4 is used in the inequality case and implies Lipschitzness of $F$. We need the stronger assumption in Assumption 2.5 for our dimension-free result for the linear inequality case.

## 3 Lower-level problem with linear equality constraint

We first obtain improved results for the setting of bilevel programs with linear *equality* constraints in the lower-level problem. Our formal problem statement is:

$$\text{minimize}_{x \in \mathcal{X}} \ F(x) := f(x, y^*(x)) \ \text{subject to} \ y^*(x) \in \arg\min_{y:h(x,y)=0} g(x, y), \qquad (3.1)$$

where $f$, $g$, $h(x,y) := Ax - By - b$, and $\mathcal{X}$ satisfy Assumptions 2.2 and 2.3. The previous best result on Problem 3.1 providing finite-time $\epsilon$-stationarity guarantees, by Khanduri et al. [37], required certain regularity assumptions on $F$ as well as Hessian computations. In contrast, our finite-time guarantees *require assumptions only on $f$ and $g$, not on $F$*; indeed, in our work, these desirable properties of $F$ are naturally implied by our analysis. Specifically, our key insight is that the hypergradient $\nabla F(x) := \nabla_x f(x, y^*) + \left(\frac{dy^*(x)}{dx}\right)^\top \nabla_y f(x, y^*)$ for Problem 3.1 is Lipschitz-continuous and admits an easily computable — yet highly accurate — finite-difference approximation. Therefore, $O(\epsilon^{-2})$ iterations of gradient descent on $F$ with this finite-difference gradient proxy yield an $\epsilon$-stationary point.

Specifically, for any fixed $x \in \mathcal{X}$, our proposed finite-difference gradient proxy approximating the non-trivial-to-compute component $\left(\frac{dy^*(x)}{dx}\right)^\top \nabla_y f(x, y^*)$ of the hypergradient is given by

$$v_x := \frac{\nabla_x[g(x, y_\delta^*) + \langle \lambda_\delta^*, h(x, y^*)\rangle] - \nabla_x[g(x, y^*) + \langle \lambda^*, h(x, y^*)\rangle]}{\delta}, \tag{3.2}$$

where $(y_\delta^*, \lambda_\delta^*)$ are the primal and dual solutions to the perturbed lower-level problem:

$$y_\delta^* := \arg\min_{y:h(x,y)=0} g(x,y) + \delta f(x,y). \tag{3.3}$$

We show in Lemma 3.2 that $v$ in (3.2) approximates $\left(\frac{dy^*(x)}{dx}\right)^\top \nabla_y f(x, y^*)$ up to an $O(\delta)$-additive error, implying the gradient oracle construction outlined in the pseudocode presented in Algorithm 1. Our full implementable algorithm for solving Problem 3.1 is displayed in Algorithm 5.

---

**Algorithm 1** Inexact Gradient Oracle for Bilevel Program with Linear Equality Constraint

---

1: **Input:** Current $x$, accuracy $\epsilon$, perturbation $\delta = \epsilon^2$.
2: Compute $y^*$ (as in Problem 3.1) and corresponding optimal dual $\lambda^*$ (as in (B.1))
3: Compute $y_\delta^*$ (as in (3.3)) and and corresponding optimal dual $\lambda_\delta^*$ (as in (B.7))
4: Compute $v_x$ as in (3.2)          ▷ Approximates $(dy^*(x)/dx)^\top \nabla_y f(x, y^*)$
5: **Output:** $\widetilde{\nabla} F = v_x + \nabla_x f(x, y^*)$

---

Notice that the finite-difference term in (3.2) avoids differentiating through the implicit function $y^*$. Instead, all we need to evaluate it are the values of $(y^*, \lambda^*, y_\delta^*, \lambda_\delta^*)$ (and gradients of $g$ and $h$). Since $(y^*, \lambda^*)$ are solutions to a smooth strongly convex linearly constrained problem, they can be approximated at a linear rate. Similarly, since the approximation error in (3.2) is proportional to $\delta$ (cf. Lemma 3.2), a small enough $\delta$ in the perturbed objective $g + \delta f$ in (3.3) ensures that it is dominated by the strongly convex and smooth $g$, whereby accurate approximates to $(y_\delta^*, \lambda_\delta^*)$ can also be readily obtained. Putting it all together, the proposed finite-difference hypergradient proxy in (3.2) is efficiently computable, yielding the following guarantee.

**Theorem 3.1.** *Consider Problem 3.1 under Assumption 2.2, and let $\kappa = C_g/\mu_g$ be the condition number of $g$. Then Algorithm 5 finds an $\epsilon$-stationary point (in terms of gradient mapping, see (B.14)) after $T = \widetilde{O}(C_F(F(x_0) - \inf F)\sqrt{\kappa}\epsilon^{-2})$ oracle calls to $f$ and $g$, where $C_F := 2(L_f + C_f + C_g)C_H^3 S_g(L_g + \|A\|)^2$ is the smoothness constant of the hyperobjective $F$.*

We now sketch the proof of Theorem 3.1. The complete proofs may be found in Appendix B.

## 3.1 Main technical ideas

We briefly outline the two key technical building blocks alluded to above, that together give us Theorem 3.1: the approximation guarantee of our finite-difference gradient proxy ((3.2)) and the smoothness of hyperobjective $F$ (for Problem 3.1). The starting point for both these results is the following simple observation obtained by implicitly differentiating, with respect to $x$, the KKT system associated with $y^* = \arg\min_{h(x,y)=0} g(x,y)$ and optimal dual variable $\lambda^*$:

$$\begin{bmatrix} \frac{dy^*(x)}{dx} \\ \frac{d\lambda^*(x)}{dx} \end{bmatrix} = \begin{bmatrix} \nabla_{yy}^2 g(x, y^*) & \nabla_y h(x, y^*)^\top \\ \nabla_y h(x, y^*) & 0 \end{bmatrix}^{-1} \begin{bmatrix} -\nabla_{yx}^2 g(x, y^*) \\ -\nabla_x h(x, y^*) \end{bmatrix} \tag{3.4}$$

The invertibility of the matrix in the preceding equation is proved in Corollary B.3: essentially, this invertibility is implied by strong convexity of $g$ and $\nabla_y h(x, y^*) = B$ having full row rank. This in conjunction with the compactness of $\mathcal{X}$ implies that the inverse of the matrix is bounded by some constant $C_H$ (cf. Corollary B.3 for details). Our hypergradient approximation guarantee follows:

**Lemma 3.2.** *For Problem 3.1 under Assumption 2.2, with $v_x$ as in (3.2), the following holds:*

$$\left\| v_x - \left( \frac{dy^*(x)}{dx} \right)^\top \nabla_y f(x, y^*) \right\| \leq O(C_F \delta).$$

*Proof sketch; see Appendix B.* The main idea is that the two terms being compared are essentially the same by the implicit function theorem. First, we use the expression for $\frac{dy^*(x)}{dx}$ from (3.4):

$$\left( \frac{dy^*(x)}{dx} \right)^\top \nabla_y f(x, y^*) = \begin{bmatrix} -\nabla_{yx}^2 g(x, y^*) \\ -\nabla_x h(x, y^*) \end{bmatrix}^\top \begin{bmatrix} \nabla_{yy}^2 g(x, y^*) & \nabla_y h(x, y^*)^\top \\ \nabla_y h(x, y^*) & 0 \end{bmatrix}^{-1} \begin{bmatrix} \nabla_y f(x, y^*) \\ 0 \end{bmatrix}.$$

We now examine $v_x$. For simplicity of exposition, we instead consider $\lim_{\delta \to 0} \frac{\nabla_x[g(x, y_\delta^*) + \langle \lambda_\delta^*, h(x, y^*) \rangle] - \nabla_x[g(x, y^*) + \langle \lambda^*, h(x, y^*) \rangle]}{\delta}$, which, by the fundamental theorem of calculus and Assumption 2.3, equals $v_x$ up to an $O(\delta)$-additive error. Note that this expression is:

$$\nabla_{xy}^2 g(x, y^*) \frac{dy_\delta^*}{d\delta} + \nabla_x h(x, y^*)^\top \frac{d\lambda_\delta^*}{d\delta}. \tag{3.5}$$

Since $y_\delta^*$ is minimizes a strongly convex function over a linear equality constraint (3.3), the reasoning that yields (3.4) also gives the following, which, when combined with (3.5), finishes the proof:

$$\begin{bmatrix} \frac{dy_\delta^*(x)}{d\delta} \\ \frac{d\lambda_\delta^*(x)}{d\delta} \end{bmatrix} \Bigg|_{\delta=0} = \begin{bmatrix} \nabla_{yy}^2 g(x, y^*) & \nabla_y h(x, y^*)^\top \\ \nabla_y h(x, y^*) & 0 \end{bmatrix}^{-1} \begin{bmatrix} -\nabla_y f(x, y^*) \\ 0 \end{bmatrix}. \tag{3.6}$$

$\square$

Having shown the construction of the hypergradient approximation, we now state smoothness of the hyperobjective $F$ (proof in Appendix B), crucial to getting our claimed rate.

**Lemma 3.3.** *The solution $y^*$ (as defined in Problem 3.1) is $O(C_H \cdot (C_g + \|A\|))$-Lipschitz continuous and $O(C_H^3 \cdot S_g \cdot (C_g + \|A\|)^2)$-smooth as a function of $x$. Thus the hyper-objective $F$ is gradient-Lipschitz with a smoothness constant of $C_F := O\{(L_f + C_f + C_g)C_H^3 S_g(L_g + \|A\|)^2\}$.*

## 4 Nonsmooth nonconvex optimization with inexact oracle

We now shift gears from the case of linear *equality* constraints to that of linear *inequality* constraints. Specifically, defining $h(x, y) = Ax - By - b$, the problem we now consider is

$$\text{minimize}_x \ F(x) := f(x, y^*(x)) \quad \text{subject to } y^*(x) \in \arg\min_{y:h(x,y)\leq 0} g(x, y). \tag{4.1}$$

As noted earlier, for this larger problem class, the hyperobjective $F$ can be nonsmooth nonconvex, necessitating our measure of convergence to be the now popular notion of Goldstein stationarity [43].

Our first algorithm for solving Problem 4.1 is presented in Algorithm 2, with its convergence guarantee in Theorem 4.1. At a high level, this algorithm first obtains access to an inexact zeroth-order oracle to $F$ (we shortly explain how this is done) and uses this oracle to construct a (biased) gradient estimate of $F$. It then uses this gradient estimate to update the iterates with a rule motivated by recent works reducing nonconvex optimization to online optimization [47]. We explain this in Section 4.1.

**Theorem 4.1.** *Consider Problem 4.1 under Assumptions 2.2 and 2.4. Let $\kappa = C_g/\mu_g$ be the condition number of $g$. Then combining the procedure for Lemma 4.2 with Algorithm 2 run with $\rho = \min\left\{\frac{\delta}{2}, \frac{F(x_0) - \inf F}{L_f L_y}\right\}$, $\nu = \delta - \rho$, $D = \Theta\left(\frac{\nu \epsilon^2 \rho^2}{d_x \rho^2 L_f^2 L_y^2 + \alpha^2 d_x^2}\right)$, and $\eta = \Theta\left(\frac{\nu \epsilon^3 \rho^4}{(d_x \rho^2 L_f^2 L_y^2 + \alpha^2 d_x^2)^2}\right)$ outputs $x^{\mathrm{out}}$ such that $\mathbb{E}[\mathrm{dist}(0, \partial_\delta F(x^{\mathrm{out}}))] \leq \epsilon + \alpha$ with $T$ oracle calls to $f$ and $g$, where:*

$$T = O\left( \frac{\sqrt{\kappa} d_x (F(x_0) - \inf F)}{\delta \epsilon^3} \cdot \left( L_f^2 L_y^2 + \alpha^2 \left( \frac{d_x}{\delta^2} + \frac{d_x L_f^2 L_y^2}{(F(x_0) - \inf F)^2} \right) \right) \cdot \log(L_f/\alpha) \right).$$

---

**Algorithm 2** Nonsmooth Nonconvex Algorithm with Inexact Zero-Order oracle

---

1: **Input:** Initialization $x_0 \in \mathbb{R}^d$, clipping parameter $D > 0$, step size $\eta > 0$, smoothing parameter $\rho > 0$, accuracy parameter $\nu > 0$, iteration budget $T \in \mathbb{N}$, inexact zero-order oracle $\widetilde{F} : \mathbb{R}^d \to \mathbb{R}$.
2: **Initialize:** $\Delta_1 = \mathbf{0}$
3: **for** $t = 1, \ldots, T$ **do**
4:     Sample $s_t \sim \mathrm{Unif}[0, 1], \ w_t \sim \mathrm{Unif}(\mathbb{S}^{d-1})$
5:     $x_t = x_{t-1} + \Delta_t, \ \ z_t = x_{t-1} + s_t \Delta_t$
6:     $\widetilde{g}_t = \frac{d}{2\rho}(\widetilde{F}(z_t + \rho w_t) - \widetilde{F}(z_t - \rho w_t))w_t$
7:     $\Delta_{t+1} = \mathrm{clip}_D (\Delta_t - \eta \widetilde{g}_t)$              $\triangleright \mathrm{clip}_D(z) := \min\{1, \frac{D}{\|z\|}\} \cdot z$
8: $M = \lfloor \frac{\nu}{D} \rfloor, \ K = \lfloor \frac{T}{M} \rfloor$
9: **for** $k = 1, \ldots, K$ **do**
10:     $\overline{x}_k = \frac{1}{M} \sum_{m=1}^{M} z_{(k-1)M+m}$
11: Sample $x^{\mathrm{out}} \sim \mathrm{Unif}\{\overline{x}_1, \ldots, \overline{x}_K\}$
12: **Output:** $x^{\mathrm{out}}$.

---

Algorithm 2 is a variant of gradient descent with momentum and clipping, with $\widetilde{g}_t$ the inexact gradient, $\Delta_t$ a clipped accumulated gradient (hence accounts for past gradients, which serve as a momentum), and the clipping ensuring that consecutive iterates of the algorithm reside within a $\delta$-ball of each other. While similar algorithms have appeared in prior work on nonsmooth nonconvex optimization (e.g. [47]), none of them account for inexactness in the gradient, crucial in our setting.

### 4.1 Nonsmooth nonconvex optimization with inexact zeroth-order oracle

We can obtain inexact zeroth-order oracle access to $F$ because (as formalized in Lemma 4.2) despite potential nonsmoothness and nonconvexity of $F$ in Problem 4.1, estimating its *value* $F(x)$ at any point $x$ amounts to solving a single smooth and strongly convex optimization problem, which can be done can be done in $\widetilde{O}(1)$ oracle calls to $f$ and $g$ by appealing to a result by Zhang and Lan [45].

**Lemma 4.2** (Proof in Appendix C.1). *Given any $x$, we can return $\widetilde{F}(x)$ such that $|F(x) - \widetilde{F}(x)| \leq \alpha$ using $O(\sqrt{C_g/\mu_g} \log(L_f/\alpha))$ first-order oracle calls to $f$ and $g$.*

Having computed an inexact value of the hyperobjective $F$, we now show how to use it to develop an algorithm for solving Problem 4.1. To this end, we first note that $F$, despite being possibly nonsmooth and nonconvex, is Lipschitz and hence amenable to the use of recent algorithmic developments in nonsmooth nonconvex optimization pertaining to Goldstein stationarity.

**Lemma 4.3.** *Under Assumption 2.2 and 2.5, $F$ in Problem 4.1 is $O(L_f L_y)$-Lipschitz in $x$.*

With this guarantee on the Lipschitzness of $F$, we prove Theorem C.1 for attaining Goldstein stationarity using the inexact zeroth-order oracle of a Lipschitz function. Our proof of Theorem C.1 crucially uses the recent online-to-nonconvex framework of Cutkosky, Mehta, and Orabona [47]. Combining Lemma 4.2 and Theorem C.1 then immediately implies Theorem 4.1.

### 4.2 Nonsmooth nonconvex optimization with inexact *gradient* oracle

In Section 5, we provide a way to generate approximate gradients of $F$. Here, we present an algorithm that attains Goldstein stationarity of Problem 4.1 using this inexact gradient oracle. While there has been a long line of recent work on algorithms for nonsmooth nonconvex optimization with convergence to Goldstein stationarity [43, 48, 86–88], these results necessarily require *exact* gradients. This brittleness to any error in gradients renders them ineffective in our setting, where our computed (hyper)gradient necessarily suffers from an additive error. While inexact oracles are known to be effective for smooth or convex objectives [89], utilizing inexact gradients in the nonsmooth nonconvex regime presents a nontrivial challenge. Indeed, without any local bound on gradient variation due to smoothness, or convexity that ensures that gradients are everywhere correlated with the direction pointing at the minimum, it is not clear a priori how to control the accumulating price of inexactness throughout the run of an algorithm. To derive such results, we use the recently proposed connection between online learning and nonsmooth nonconvex optimization by Cutkosky, Mehta,

and Orabona [47]. By controlling the accumulated error suffered by online gradient descent for *linear* losses (cf. Lemma C.3), we derive guarantees for our setting of interest, providing Lipschitz optimization algorithms that converge to Goldstein stationary points even with inexact gradients.

This algorithm matches the best known complexity in first-order nonsmooth nonconvex optimization [43, 47, 48], merely replacing the convergence to a $(\delta, \epsilon)$-stationary point by $(\delta, \epsilon + \alpha)$-stationarity, where $\alpha$ is the inexactness of the gradient oracle.

---

**Algorithm 3** Nonsmooth Nonconvex Algorithm with Inexact Gradient Oracle

---

1: **Input:** Initialization $x_0 \in \mathbb{R}^d$, clipping parameter $D > 0$, step size $\eta > 0$, accuracy parameter $\delta > 0$, iteration budget $T \in \mathbb{N}$, inexact gradient oracle $\widetilde{\nabla} F : \mathbb{R}^d \to \mathbb{R}^d$.
2: **Initialize:** $\Delta_1 = \mathbf{0}$
3: **for** $t = 1, \ldots, T$ **do**
4:     Sample $s_t \sim \text{Unif}[0,1]$
5:     $x_t = x_{t-1} + \Delta_t, \quad z_t = x_{t-1} + s_t \Delta_t$
6:     $\widetilde{g}_t = \widetilde{\nabla} F(z_t)$
7:     $\Delta_{t+1} = \text{clip}_D (\Delta_t - \eta \widetilde{g}_t)$                $\triangleright \text{clip}_D(z) := \min\{1, \frac{D}{\|z\|}\} \cdot z$
8: $M = \lfloor \frac{\delta}{D} \rfloor, \ K = \lfloor \frac{T}{M} \rfloor$
9: **for** $k = 1, \ldots, K$ **do**
10:     $\overline{x}_k = \frac{1}{M} \sum_{m=1}^{M} z_{(k-1)M+m}$
11: Sample $x^{\text{out}} \sim \text{Unif}\{\overline{x}_1, \ldots, \overline{x}_K\}$
12: **Output:** $x^{\text{out}}$.

---

**Theorem 4.4.** *Suppose $F : \mathbb{R}^d \to \mathbb{R}$ is $L$-Lipschitz and that $\|\widetilde{\nabla} F(\cdot) - \nabla F(\cdot)\| \leq \alpha$. Then running Algorithm 3 with $D = \Theta(\frac{\delta \epsilon^2}{L^2}), \eta = \Theta(\frac{\delta \epsilon^3}{L^4})$, outputs a point $x^{\text{out}}$ such that $\mathbb{E}[\text{dist}(0, \partial_\delta F(x^{\text{out}}))] \leq \epsilon + \alpha$, with $T = O\left(\frac{(F(x_0) - \inf F)L^2}{\delta \epsilon^3}\right)$ calls to $\widetilde{\nabla} F(\cdot)$.*

We defer the proof of Theorem 4.4 to Appendix C.1. Plugging the complexity of computing inexact gradients, as given by Theorem 5.3, into the result above, we immediately obtain convergence to a $(\delta, \epsilon)$-stationary point of Problem 1.1 with $\widetilde{O}(\delta^{-1} \epsilon^{-4})$ gradient calls overall.

**Implementation-friendly algorithm.** While Algorithm 3 matches the best-known results in nonsmooth nonconvex optimization, it could be impractical due to several hyperparameters which need tuning. Arguably, a more natural application of the hypergradient estimates would be simply plugging them into gradient descent, which requires tuning only the stepsize. Since $F$ is neither smooth nor convex, perturbations are required to guarantee differentiability along the trajectory. We therefore complement Theorem 4.4 by analyzing perturbed (inexact) gradient descent in the nonsmooth nonconvex setting (Algorithm 7) and state its theoretical guarantee in Theorem C.4. Despite its suboptimal worst-case theoretical guarantees, we find this algorithm easier to implement in practice.

## 5 Inequality constraints: constructing the inexact gradient oracle

Computing a stationary point of $F$ of Problem 4.1 via any first-order method would require:

$$\nabla F(x) = \nabla_x f(x, y^*) + \left(\frac{dy^*(x)}{dx}\right)^\top \nabla_y f(x, y^*), \tag{5.1}$$

for which the key challenge lies in computing $dy^*(x)/dx$. This requires differentiating through an argmin operator, which typically requires second-order derivatives. Instead, here we differentiate (using the implicit function theorem) through the KKT conditions describing $y^*(x)$ and get:

$$\begin{bmatrix} \nabla_{yy}^2 g + (\lambda^*)^\top \nabla_{yy}^2 h & \nabla_y h_{\mathcal{I}}^\top \\ \text{diag}(\lambda_{\mathcal{I}}^*) \nabla_y h_{\mathcal{I}} & 0 \end{bmatrix} \begin{bmatrix} \frac{dy^*(x)}{dx} \\ \frac{d\lambda_{\mathcal{I}}^*(x)}{dx} \end{bmatrix} = - \begin{bmatrix} \nabla_{yx}^2 g + (\lambda^*)^\top \nabla_{yx}^2 h \\ \text{diag}(\lambda_{\mathcal{I}}^*) \nabla_x h_{\mathcal{I}} \end{bmatrix} \tag{5.2}$$

where given $x$, we assume efficient access to the optimal dual solution $\lambda^*(x)$ of the LL problem in Problem 4.1. In (5.2), we use $\mathcal{I} := \{i \in [d_h] : h_i(x, y) = 0, \lambda_i^* > 0\}$ to denote the set of active

constraints with non-zero dual solution, with $h_{\mathcal{I}} := [h_i]_{i \in \mathcal{I}}$ and $\lambda_{\mathcal{I}}^* := [\lambda_i^*]_{i \in \mathcal{I}}$ being the constraints and dual variables, respectively, corresponding to $\mathcal{I}$.

Observe that as currently stated, (5.2) leads to a second-order computation of $dy^*(x)/dx$. In the rest of the section, we provide a fully first-order approximate hypergradient oracle by constructing an equivalent reformulation of Problem 4.1 using a penalty function.

## 5.1 Reformulation via the penalty method

We begin by reformulating Problem 4.1 into a single level constrained optimization problem:

$$\text{minimize}_{x,y} \ f(x,y) \text{ subject to } \begin{cases} g(x,y) + (\lambda^*(x))^\top h(x,y) \leq g^*(x) \\ h(x,y) \leq 0 \end{cases}, \quad (5.3)$$

where $g^*(x) := \min_{y:h(x,y) \leq 0} g(x,y) = g(x, y^*(x))$ and $\lambda^*(x)$ is the optimal dual solution. The equivalence of this reformulation to Problem 4.1 is spelled out in Appendix D. From (5.3), we define the following penalty function, crucial to our analysis:

$$\mathcal{L}_{\lambda^*,\boldsymbol{\alpha}}(x,y) = f(x,y) + \alpha_1 \left( g(x,y) + (\lambda^*)^\top h(x,y) - g^*(x) \right) + \frac{\alpha_2}{2} \left\| h_{\mathcal{I}}(x,y) \right\|^2, \quad (5.4)$$

where $\boldsymbol{\alpha} = [\alpha_1, \alpha_2] \geq 0$ are the penalty parameters. Notably, we can compute its derivative with respect to $x$ of (5.4) in a fully first-order fashion by the following expression:

$$\nabla_x \mathcal{L}_{\lambda^*,\boldsymbol{\alpha}}(x,y) = \nabla_x f(x,y) + \alpha_1 (\nabla_x g(x,y) + \nabla_x h(x,y)^\top \lambda^* - \nabla_x g^*(x)) + \alpha_2 \nabla_x h_{\mathcal{I}}(x,y)^\top h_{\mathcal{I}}(x,y).$$

To give some intuition for our choice of penalties in (5.4), we note that the two constraints in (5.3) behave quite differently. The first constraint $g(x,y) + \lambda^*(x)^\top h(x,y) \leq g^*(x)$ is one-sided, i.e., can only be violated or met, and hence just needs a penalty parameter $\alpha_1$ to weight the "violation". As to the second constraint $h(x,y) \leq 0$, it can be arbitrary. To allow for such a "two-sided" constraint, we penalize only the active constraints $\mathcal{I}$, i.e., we use $\|h_{\mathcal{I}}(x,y)\|^2$ to penalize deviation.

Next, we define the optimal solutions to the penalty function optimization by:

$$y_{\lambda^*,\boldsymbol{\alpha}}^*(x) := \arg\min_y \mathcal{L}_{\lambda^*,\boldsymbol{\alpha}}(x,y). \quad (5.5)$$

We now show that this minimizer is close to the optimal solution of the LL problem, while suffering only a small constraint violation.

**Lemma 5.1.** *Given any $x$, the corresponding dual solution $\lambda^*(x)$, primal solution $y^*(x)$ of the lower optimization problem in Problem 4.1, and $y_{\lambda^*,\boldsymbol{\alpha}}^*(x)$ as in (5.5), satisfy:*

$$\left\| y_{\lambda^*,\boldsymbol{\alpha}}^*(x) - y^*(x) \right\| \leq O(\alpha_1^{-1}) \quad \text{and} \quad \left\| h_{\mathcal{I}}(x, y_{\lambda^*,\boldsymbol{\alpha}}^*(x)) \right\| \leq O(\alpha_1^{-1/2} \alpha_2^{-1/2}). \quad (5.6)$$

The proof of Lemma 5.1 is based on the Lipschitzness of $f$ and strong convexity of $g$ for sufficiently large $\alpha_1$. The aforementioned constraint violation bound on $h_{\mathcal{I}}(x,y)$ is later used in Lemma 5.2 to bound the inexactness of our proposed gradient oracle.

## 5.2 Main result: approximating the hypergradient

The main export of this section is the following bound on the approximation of the hypergradient. This, together with the bounds in Lemma 5.1, validate our use of the penalty function in (5.4).

**Lemma 5.2.** *Consider $F$ as in Problem 4.1, $\mathcal{L}$ as in (5.4), a fixed $x$, and $y_{\lambda^*,\boldsymbol{\alpha}}^*$ as in (5.5). Then under Assumptions 2.2 and 2.5, we have:*

$$\left\| \nabla F(x) - \nabla_x \mathcal{L}_{\lambda^*,\boldsymbol{\alpha}}(x, y_{\lambda^*,\boldsymbol{\alpha}}^*) \right\| \leq O(\alpha_1^{-1}) + O(\alpha_1^{-1/2} \alpha_2^{-1/2}) + O(\alpha_1^{1/2} \alpha_2^{-1/2}) + O(\alpha_1^{-3/2} \alpha_2^{1/2}).$$

The proof can be found in Appendix G. With this hypergradient approximation guarantee, we design Algorithm 4 to compute an inexact gradient oracle for the hyperobjective $F$.

**Theorem 5.3.** *Given any accuracy parameter $\alpha > 0$, Algorithm 4 outputs $\widetilde{\nabla}_x F(x)$ such that $\|\widetilde{\nabla} F(x) - \nabla F(x)\| \leq \alpha$ within $\widetilde{O}(\alpha^{-1})$ gradient oracle evaluations.*

The full proof of this result may be found in Appendix H.

---

**Algorithm 4** Inexact Gradient Oracle for General Inequality Constraints

---

1: **Input:** Upper level variable $x$, accuracy $\alpha$, penalty parameters $\alpha_1 = \alpha^{-2}, \alpha_2 = \alpha^{-4}$.
2: Compute $y^*$, $\lambda^*$, and active constraints $\mathcal{I}$ of the constrained LL problem $\min_{y:h(x,y)\leq 0} g(x,y)$.
3: Define penalty function $\mathcal{L}_{\lambda^*,\boldsymbol{\alpha}}(x,y)$ by (5.4)
4: Compute the minimizer $y^*_{\lambda^*,\boldsymbol{\alpha}} = \arg\min_y \mathcal{L}_{\lambda^*,\boldsymbol{\alpha}}(x,y)$ (as in (5.5)).
5: **Output:** $\widetilde{\nabla} F := \nabla_x \mathcal{L}_{\lambda^*,\boldsymbol{\alpha}}(x, y^*_{\lambda^*,\boldsymbol{\alpha}})$.

---

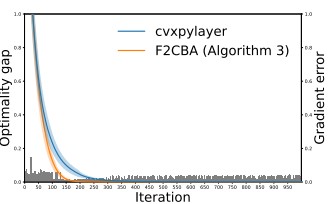 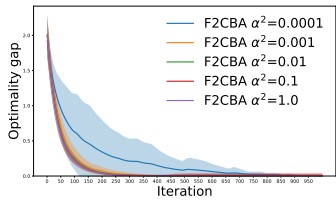 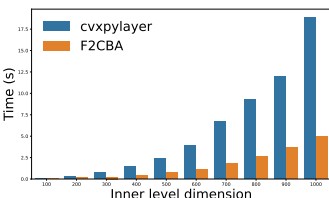

**(a)** Convergence and gradient error of Fully First-order Constrained Bilevel Algorithm (F2CBA) compared to cvxpylayer [18].

**(b)** Convergence analysis with varying gradient inexactness $\alpha$ to measure the tradeoff of accuracy and convergence.

**(c)** Computation cost per gradient step of varying problem size $d_y$. We vary $d_y$ from 100 to 1000 to measure the computation cost.

**Figure 1:** We run Algorithm 3 using Algorithm 4 on the bilevel optimization in the toy example in Problem L.1 with $d_x = 100$, $d_y = 200$, $n_{\mathrm{const}} = d_y/5$, and accuracy $\alpha = 0.1$. Figure 2a, Figure 2b, Figure 2c vary # of iterations, gradient exactness $\alpha$, and $d_y$, respectively, to compare the performance under different settings.

## 6 Experiments

We generate instances of the following constrained bilevel optimization problem:

$$\min_x c^\top y^* + 0.01 \|x\|^2 + 0.01 \|y^*\|^2 \quad \text{s.t.} \quad y^* = \arg\min_{y:h(x,y)\leq 0} \frac{1}{2} y^\top Q y + x^\top P y, \quad (6.1)$$

where $h(x,y) = Ay - b$ is a $d_h$-dim linear constraint. The PSD matrix $Q \in \mathbb{R}^{d_y \times d_y}$, $c \in \mathbb{R}^{d_y}$, $P \in \mathbb{R}^{d_x \times d_y}$, and constraints $A \in \mathbb{R}^{d_h \times d_y}$, $b \in \mathbb{R}^{d_h}$ are randomly generated from normal distributions (cf. Appendix K). We compare Algorithm 3 with a non-fully first-order method using cvxpyLayer [18]. Both algorithms use Adam [90] to control the learning rate, and are averaged over 10 random seeds.

Figure 2a shows that both the algorithms converge to the same optimal solution at the same rate. Simultaneously, the colorful bars represent the gradient differences between two methods, showing the inexactness of our gradients. Figure 2b additionally varies this inexactness to demonstrate its impact on convergence with standard deviation plotted. Figure 2c compares the computation costs for different lower-level problem sizes. Our fully first-order method significantly outperforms, in computation cost, the non-fully first-order method implemented using differentiable optimization method. The implementation can be found in https://github.com/guaguakai/constrained-bilevel-optimization.

## 7 Limitations and future directions

One limitation to our approach is that the inexact gradient oracle we constructed in Section 5 requires access to the exact dual multiplier $\lambda^*$. For a first-order algorithm, the closest proxy one could get to this would be a highly accurate approximation (which could be computed up to $\epsilon$ error in $O(\log(1/\epsilon))$ gradient oracle evaluations). Removing this "exact dual access" assumption (Assumption 2.5) would be an important result.

Another important direction for future work would be to extend the hypergradient stationarity guarantee of Theorem 1.4 to bilevel programs with *general convex* constraints. To this end, we conjecture that the use of a primal-only gradient approximation oracle could be potentially effective.

Finally, our current rate of $\widetilde{O}(\delta^{-1}\epsilon^{-4})$ oracle calls for reaching $(\delta, \epsilon)$-Goldstein stationarity is not necessarily inherent to the problem; indeed, it might be the case that an alternate approach could improve it to the best known rate of $O(\delta^{-1}\epsilon^{-3})$ for nonsmooth nonconvex optimization.

## Acknowledgments

GK is supported by an Azrieli Foundation graduate fellowship. KW is supported by Schmidt Sciences AI2050 Fellowship and NSF IIS-2403240. SS acknowledges generous support from the Alexander von Humboldt Foundation. SP is supported by NSF CCF-2112665 (TILOS AI Research Institute) and gratefully acknowledges Ali Jadbabaie for many useful discussions. ZZ is supported by NSF FODSI Fellowship.

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

# Appendix

## A  Notation

We use $\langle \cdot, \cdot \rangle$ to denote inner products and $\| \cdot \|$ for the Euclidean norm. Unless transposed, all vectors are column vectors. For $f : \mathbb{R}^{d_2} \to \mathbb{R}^{d_1}$ its Jacobian with respect to $x \in \mathbb{R}^{d_2}$ is $\nabla f \in \mathbb{R}^{d_1 \times d_2}$. For $f : \mathbb{R}^d \to \mathbb{R}$, we overload $\nabla f$ to refer to its gradient (the transposed Jacobian), a column vector. We use $\nabla_x$ to denote partial derivatives with respect to $x$.

A function $f : \mathbb{R}^n \to \mathbb{R}^m$ is $L$-Lipschitz if for any $x, y$, we have $\|f(x) - f(y)\| \leq L\|x - y\|$. A differentiable function $f : \mathbb{R}^n \to \mathbb{R}$ is convex if for any $x, y \in \mathbb{R}^n$ we have $f(y) \geq f(x) + \nabla f(x)^\top (y - x)$; it is $\mu$-strongly convex if $f - \frac{\mu}{2}\| \cdot \|^2$ is convex; it is $\beta$-smooth if $\nabla f$ is $\beta$-Lipschitz.

For a Lipschitz function $f$, a point $x$ is $(\delta, \epsilon)$-stationary if within a $\delta$-ball around $x$, there exists a convex combination of subgradients of $f$ with norm at most $\epsilon$. For a differentiable function $f$, we say that $x$ is $\epsilon$-stationary if $\|\nabla f(x)\| \leq \epsilon$.

## B  Proofs from [Section 3](#)

In this section, we provide the full proofs of claims for bilevel programs with linear equality constraints, as stated in Section 3. We first state a few technical results using the implicit function theorem that we repeatedly invoke in our results for this setting.

**Lemma B.1.** *Fix a point $x$. Given $y^* = \arg\min_{y:h(x,y)=0} g(x, y)$ where $g$ is strongly convex in $y$ and $\lambda^*$ is the dual optimal variable for this problem, define $\mathcal{L}_{eq}(x, y, \lambda) = g(x, y) + \langle \lambda, h(x, y) \rangle$. Then, we have*

$$\underbrace{\begin{bmatrix} \nabla_{yy}^2 \mathcal{L}_{eq}(x, y^*, \lambda^*) & \nabla_y h(x, y^*)^\top \\ \nabla_y h(x, y^*) & 0 \end{bmatrix}}_{H \ for \ linear \ equality \ constraints} \begin{bmatrix} \frac{dy^*}{dx} \\ \frac{d\lambda^*}{dx} \end{bmatrix} = \begin{bmatrix} -\nabla_{yx}^2 g(x, y^*) - \nabla_{yx}^2 \langle \lambda^*, h(x, y^*) \rangle \\ -\nabla_x h(x, y^*) \end{bmatrix}.$$

*Proof.* Since $g$ is strongly convex, by linear constraint qualification, the KKT condition is both sufficient and necessary condition for optimality. Hence, consider the following KKT system obtained via first order optimality of $y^*$, with dual optimal variable $\lambda^*$:

$$\nabla_y g(x, y^*) + \nabla_y \langle \lambda^*, h(x, y^*) \rangle = 0, \ \text{and} \ h(x, y^*) = 0. \tag{B.1}$$

Differentiating the system of equations in (B.1) with respect to $x$ and rearranging terms in a matrix-vector format yields:

$$\begin{bmatrix} \nabla_{yy}^2 g(x, y^*) + \nabla_{yy}^2 \langle \lambda^*, h(x, y^*) \rangle & \nabla_y h(x, y^*)^\top \\ \nabla_y h(x, y^*) & 0 \end{bmatrix} \begin{bmatrix} \frac{dy^*}{dx} \\ \frac{d\lambda^*}{dx} \end{bmatrix} = \begin{bmatrix} -\nabla_{yx}^2 g(x, y^*) - \nabla_{yx}^2 \langle \lambda^*, h(x, y^*) \rangle \\ -\nabla_x h(x, y^*) \end{bmatrix}$$

$$\tag{B.2}$$

Noting that $\nabla_{yy}^2 \mathcal{L}_{eq}(x, y, \lambda) = \nabla_{yy}^2 g(x, y) + \nabla_{yy}^2 \langle \lambda, h(x, y) \rangle$, we can write (B.2) in the form shown in the lemma. $\square$

**Lemma B.2.** *Consider the setup in [Lemma B.1](#). The matrix $H$ defined in (3.4) is invertible if the Hessian $\nabla_{yy}^2 \mathcal{L}_{eq}(x, y^*, \lambda^*) := \nabla_{yy}^2 g(x, y^*) + \nabla_{yy}^2 \langle \lambda^*, h(x, y^*) \rangle$ satisfies $\nabla_{yy}^2 \mathcal{L}_{eq}(x, y^*, \lambda^*) \succ 0$ over the tangent plane $T := \{y : \nabla_y h(x, y^*)y = 0\}$ and $\nabla_y h$ has full rank.*

*Proof.* Let $u = [y, \lambda]$. We show that $Hu = 0$ implies $u = 0$, which in turn implies invertibility of $H$. If $\nabla_y h(x, y^*)y \neq 0$, then by construction of $u$ and $H$, we must also have $Hu \neq 0$. Otherwise if $\nabla_y h(x, y^*)y = 0$ and $y \neq 0$, the quadratic form $u^\top Hu$ is positive, as seen by

$$u^\top Hu = y^\top \nabla_{yy}^2 \mathcal{L}_{eq}(x, y^*, \lambda^*)y > 0,$$

where the final step is by the assumption of $\mathcal{L}_{eq}$ being positive definite over the defined tangent plane $T = \{y : \nabla_y h(x, y^*)y = 0\}$. If $y = 0$ while $Hu = 0$, then $\nabla_y h$ having full rank implies $\lambda = 0$. Combined with $y = 0$, this means $u = 0$, as required when $Hu = 0$. This concludes the proof. $\square$

**Corollary B.3.** *For [Problem 3.1] under [Assumption 2.2] and [Assumption 2.3], the matrix $H$ (as defined in (3.4)) is non-singular. Further, there exists a finite $C_H$ such that $\|H^{-1}\| \leq C_H$.*

*Proof.* Since we are assuming strong convexity of $g$, [Lemma B.2] applies, yielding the claimed invertibility of $H$. Combined with the boundedness of variables $x$ (per [Assumption 2.3]) and continuity of the inverse implies a bound on $\|H^{-1}\|$. □

## B.1 Construction of the inexact gradient oracle

We now show how to construct the inexact gradient oracle for the objective $F$ in [Problem 3.1]. As sketched in [Section 3], we then use this oracle in a projected gradient descent algorithm to get the claimed guarantee.

**Lemma B.4.** *Consider [Problem 3.1] under [Assumption 2.2] and [Assumption 2.3]. Let $y_\delta^*$ be as defined in (3.3). Then, for any $\delta \in [0, \Delta]$ with $\Delta \leq \mu_g / 2C_f$, the following relation is valid:*

$$\|y_\delta^* - y^*\| \leq M(x)\delta, \text{ with } M(x) := \frac{2}{\mu_g}\|\nabla_y f(x, y^*)\| \leq \frac{2L_f}{\mu_g}.$$

*Proof.* The first-order optimality condition applied to $g(x, y) + \delta f(x, y)$ at $y^*$ and $y_\delta^*$ gives

$$\langle \nabla_y g(x, y_\delta^*) + \delta \nabla_y f(x, y_\delta^*), y^* - y_\delta^* \rangle \geq 0,$$

which upon adding and subtracting $\nabla_y f(x, y^*)$ transforms into

$$\langle \nabla_y g(x, y_\delta^*) + \delta[\nabla_y f(x, y_\delta^*) - \nabla_y f(x, y^*)] + \delta \nabla_y f(x, y^*), y^* - y_\delta^* \rangle \geq 0. \tag{B.3}$$

Similarly, the first-order optimality condition applied to $g$ at $y^*$ and $y_\delta^*$ gives

$$\langle \nabla_y g(x, y^*), y_\delta^* - y^* \rangle \geq 0. \tag{B.4}$$

Adding [Inequality (B.3)] and [Inequality (B.4)] and rearranging yields

$$\langle \nabla_y g(x, y_\delta^*) - \nabla_y g(x, y^*) + \delta[\nabla_y f(x, y_\delta^*) - \nabla_y f(x, y^*)], y_\delta^* - y^* \rangle \leq \langle \delta \nabla_y f(x, y^*), y^* - y_\delta^* \rangle.$$

Applying to the left side above a lower bound via strong convexity of $g + \delta f$ and to the right hand side an upper bound via Cauchy-Schwarz inequality, we have

$$s\|y_\delta^* - y^*\| \leq \delta \|\nabla_y f(x, y^*)\|, \tag{B.5}$$

where $s$ is the strong convexity of $g + \delta f$. Since $f$ is $C_f$-smooth, the worst case value of this is $s = \mu_g - \delta C_f = \mu_g - \frac{\mu_g}{2C_f}C_f = \mu_g/2$, which when plugged in [Inequality (B.5)] then gives the claimed bound. □

**Lemma B.5.** *Consider [Problem 3.1] under [Assumption 2.2] and [Assumption 2.3]. Then the following relation is valid.*

$$\lim_{\delta \to 0} \frac{\nabla_x[g(x, y_\delta^*(x)) + \lambda_\delta^* h(x, y^*)] - \nabla_x[g(x, y^*(x)) + \lambda^* h(x, y^*)]}{\delta} = \left(\frac{dy^*(x)}{dx}\right)^\top \nabla_y f(x, y^*(x)).$$

*Proof.* Recall that by definition, $g$ is strongly convex and $y^* = \arg\min_{y:h(x,y)=0} g(x, y)$. Hence, we can apply [Lemma B.1]. Combining this with [Lemma B.2] and further applying that linearity of $h$ implies $\nabla_{yy}^2 h = 0$ and $\nabla_{xy}^2 h = 0$, we obtain the following:

$$\begin{bmatrix} \frac{dy^*}{dx} \\ \frac{d\lambda^*}{dx} \end{bmatrix} = \begin{bmatrix} \nabla_{yy}^2 g(x, y^*) & \nabla_y h(x, y^*)^\top \\ \nabla_y h(x, y^*) & 0 \end{bmatrix}^{-1} \begin{bmatrix} -\nabla_{yx}^2 g(x, y^*) \\ -\nabla_x h(x, y^*) \end{bmatrix}.$$

So we can express the right-hand side of the claimed equation in the lemma statement by

$$\left(\frac{dy^*(x)}{dx}\right)^\top \nabla_y f(x, y^*(x)) = \left[ \left(\frac{dy^*}{dx}\right)^\top \quad \left(\frac{d\lambda^*}{dx}\right)^\top \right] \begin{bmatrix} \nabla_y f(x, y^*(x)) \\ 0 \end{bmatrix},$$

which can be further simplified to

$$\begin{bmatrix} -\nabla_{yx}^2 g(x, y^*)^\top & -\nabla_x h(x, y^*)^\top \end{bmatrix} \begin{bmatrix} \nabla_{yy}^2 g(x, y^*) & \nabla_y h(x, y^*)^\top \\ \nabla_y h(x, y^*) & 0 \end{bmatrix}^{-1} \begin{bmatrix} \nabla_y f(x, y^*(x)) \\ 0 \end{bmatrix}. \tag{B.6}$$

We now apply Lemma B.1 to the perturbed problem defined in (3.3). We know from Lemma B.4 that $\lim_{\delta\to 0} y^*_\delta = y^*$. The associated KKT system is given by

$$\delta\nabla_y f(x, y^*_\delta) + \nabla_y g(x, y^*_\delta) + \nabla_y\langle\lambda^*_\delta, h(x, y^*_\delta)\rangle = 0 \text{ and } h(x, y^*_\delta) = 0. \tag{B.7}$$

Taking the derivative with respect of (B.7) gives the following implicit system, where we used the fact that $h$ is linear and hence $\nabla^2_{yy} h = 0$:

$$\underbrace{\begin{bmatrix} \delta\nabla^2_{yy} f(x, y^*_\delta) + \nabla^2_{yy} g(x, y^*_\delta) & \nabla_y h(x, y^*_\delta)^\top \\ \nabla_y h(x, y^*_\delta) & 0 \end{bmatrix}}_{H_\delta} \begin{bmatrix} \frac{dy^*_\delta}{d\delta} \\ \frac{d\lambda^*_\delta}{d\delta} \end{bmatrix} = \begin{bmatrix} -\nabla_y f(x, y^*_\delta)^\top \\ 0 \end{bmatrix}. \tag{B.8}$$

For a sufficiently small $\delta$, we have $\nabla^2_{yy} g(x, y^*_\delta) + \delta\nabla^2_{yy} f(x, y^*_\delta) \succeq \frac{\mu_g}{2} I$, which implies invertibility of $H_\delta$ by an application of Lemma B.2. Since Lemma B.4 implies $\lim_{\delta\to 0} y^*_\delta = y^*$, we get

$$\begin{bmatrix} \frac{dy^*_\delta}{d\delta} \\ \frac{d\lambda^*_\delta}{d\delta} \end{bmatrix}\Big|_{\delta=0} = \begin{bmatrix} \nabla^2_{yy} g(x, y^*) & \nabla_y h(x, y^*)^\top \\ \nabla_y h(x, y^*) & 0 \end{bmatrix}^{-1} \begin{bmatrix} -\nabla_y f(x, y^*) \\ 0 \end{bmatrix}.$$

So we can express the left-hand side of the expression in the lemma statement by

$$\lim_{\delta\to 0} \frac{\nabla_x[g(x, y^*_\delta(x)) + \langle\lambda^*_\delta, h(x, y^*)\rangle] - \nabla_x[g(x, y^*(x)) + \langle\lambda^*, h(x, y^*)\rangle]}{\delta}$$

$$= \nabla^2_{xy} g(x, y^*)\frac{dy^*_\delta}{d\delta} + \nabla_x h(x, y^*)^\top\frac{d\lambda^*_\delta}{d\delta}$$

$$= \begin{bmatrix} \nabla^2_{xy} g(x, y^*) & \nabla_x h(x, y^*)^\top \end{bmatrix} \begin{bmatrix} \nabla^2_{yy} g(x, y^*) & \nabla_y h(x, y^*)^\top \\ \nabla_y h(x, y^*) & 0 \end{bmatrix}^{-1} \begin{bmatrix} -\nabla_y f(x, y^*) \\ 0 \end{bmatrix},$$

which matches (B.6) (since $(\nabla^2_{yx} g)^\top = \nabla^2_{xy} g$), thus concluding the proof. $\qquad\square$

**Lemma 3.3.** *The solution $y^*$ (as defined in Problem 3.1) is $O(C_H \cdot (C_g + \|A\|))$-Lipschitz continuous and $O(C_H^3 \cdot S_g \cdot (C_g + \|A\|)^2)$-smooth as a function of $x$. Thus the hyper-objective $F$ is gradient-Lipschitz with a smoothness constant of $C_F := O\{(L_f + C_f + C_g)C_H^3 S_g(L_g + \|A\|)^2\}$.*

*Proof.* Rearranging (3.4) and applying Corollary B.3, we have

$$\begin{bmatrix} \frac{dy^*}{dx} \\ \frac{d\lambda^*}{dx} \end{bmatrix} = \begin{bmatrix} \nabla^2_{yy} g(x, y^*) & B^\top \\ B & 0 \end{bmatrix}^{-1} \begin{bmatrix} -\nabla^2_{yx} g(x, y^*) \\ -\nabla_x h(x, y^*) \end{bmatrix}.$$

This implies a Lipschitz bound of $C_H \cdot (C_g + \|A\|)$. Next, note that in the case with linear equality constraints, the terms in (B.2) involving second-order derivatives of $h$ are all zero; differentiating (B.2) with respect to $x$, we notice that the linear system we get again has the same matrix $H$ from before. We can therefore again perform the same inversion and apply the bound on $\|H^{-1}\|$ and on the third-order derivatives of $g$ (Assumption 2.3) to observe that $\|\frac{d^2 y^*}{dx^2}\| \leq O(C_H \cdot S_g\|\frac{dy^*}{dx}\|^2) = O(C_H^3 \cdot S_g \cdot (C_g + \|A\|)^2)$, where we are hiding numerical constants in the Big-Oh notation.

As a result, we can calculate the Lipschitz smoothness constant associated with the hyper-objective $F$ by

$$\|\nabla F(x) - \nabla F(\bar{x})\|$$

$$\leq \|\frac{dy^*(x)}{dx}\nabla_y f(x, y^*(x)) - \frac{dy^*(\bar{x})}{dx}\nabla_y f(\bar{x}, y^*(\bar{x}))\| + \|\nabla_x f(x, y^*(x)) - \nabla_x f(\bar{x}, y^*(\bar{x}))\|$$

$$\leq [C_f C_H(L_g + \|A\|) + C_f C_H^2(L_g + \|A\|)^2 + L_f C_H^3 S_g(L_g + \|A\|)^2]\|x - \bar{x}\|$$

$$+ [C_f + C_f C_H(L_g + \|A\|)]\|x - \bar{x}\|$$

$$\leq \underbrace{2(L_f + C_f + C_g)C_H^3 S_g(L_g + \|A\|)^2}_{C_F}\|x - \bar{x}\|.$$

$\qquad\square$

**Lemma 3.2.** *For [Problem 3.1](#) under [Assumption 2.2](#), with $v_x$ as in [(3.2)](#), the following holds:*

$$\left\| v_x - \left( \frac{dy^*(x)}{dx} \right)^\top \nabla_y f(x, y^*) \right\| \leq O(C_F \delta).$$

*Proof.* For simplicity, we adopt the following notation throughout this proof: $g_{xy}(x, y) = \nabla^2_{xy} g$, and $g_{xyy}$ denotes the tensor such that its $ijk$ entry is given by $\frac{\partial^3 g}{\partial x_i \partial y_j \partial y_k}$. We first consider the terms involving $g$. By the fundamental theorem of calculus, we have

$$\nabla_x g(x, y^*_\delta(x)) - \nabla_x g(x, y^*(x)) = \int_{t=0}^{\delta} g_{xy}(x, y^*_t(x)) \frac{dy^*_t(x)}{dt} dt.$$

As a result, we have

$$\frac{\nabla_x g(x, y^*_\delta(x)) - \nabla_x g(x, y^*(x))}{\delta} - g_{xy}(x, y^*(x)) \frac{dy^*_t(x)}{dt}|_{t=0}$$

$$= \frac{1}{\delta} \int_{t=0}^{\delta} \left( g_{xy}(x, y^*_t(x)) \frac{dy^*_t(x)}{dt} - g_{xy}(x, y^*(x)) \frac{dy^*_t(x)}{dt}|_{t=0} \right) dt$$

$$= \frac{1}{\delta} \int_{t=0}^{\delta} \left( g_{xy}(x, y^*_t(x)) \frac{dy^*_t(x)}{dt} - g_{xy}(x, y^*(x)) \frac{dy^*_t(x)}{dt}|_{t=0} \right) dt$$

$$= \frac{1}{\delta} \int_{t=0}^{\delta} \left( g_{xy}(x, y^*_t(x)) - g_{xy}(x, y^*(x)) \right) \frac{dy^*_t(x)}{dt} dt + \frac{1}{\delta} \int_{t=0}^{\delta} g_{xy}(x, y^*(x)) \cdot \left( \frac{dy^*_t(x)}{dt} - \frac{dy^*_t(x)}{dt}|_{t=0} \right) dt.$$

$$(B.9)$$

We now bound each of the terms on the right-hand side of [(B.9)](#). For the first term, we have

$$\| \frac{1}{\delta} \int_{t=0}^{\delta} \left( g_{xy}(x, y^*_t(x)) - g_{xy}(x, y^*(x)) dt \right) \frac{dy^*_t(x)}{dt} \|$$

$$\leq \frac{1}{\delta} \int_{t=0}^{\delta} \| \frac{dy^*_t(x)}{dt} \| \cdot \int_{s=0}^{t} \| g_{xyy}(x, y^*_s(x)) \| \| \frac{dy^*_s(x)}{ds} \| ds \cdot dt$$

$$\leq \frac{1}{\delta} \int_{t=0}^{\delta} \| \frac{dy^*_t(x)}{dt} \| \cdot \max_{s \in [0,\delta]} \| g_{xyy}(x, y^*_s(x)) \| \cdot \| \frac{dy^*_s(x)}{ds} \| t dt$$

$$\leq \frac{1}{\delta} \cdot \max_{u \in [0,\delta]} \| g_{xyy}(x, y^*_u(x)) \| \cdot \delta^2 \cdot \max_{t \in [0,\delta]} \| \frac{dy^*_t(x)}{dt} \|^2$$

$$\leq \delta \cdot \max_{u \in [0,\delta]} \| g_{xyy}(x, y^*_u(x)) \| \cdot \max_{t \in [0,\delta]} \| \frac{dy^*_t(x)}{dt} \|^2$$

$$= \delta \cdot S_g \cdot M_y^2,$$

$$(B.10)$$

where $M_y$ is the Lipschitz bound on $y^*$ as shown in [Lemma 3.3](#), and $S_g$ is the smoothness of $g$ from [Assumption 2.3](#). For the second term on the right-hand side of [(B.9)](#), we have

$$\| \frac{1}{\delta} \int_{t=0}^{\delta} g_{xy}(x, y^*(x)) \cdot \left( \frac{dy^*_t(x)}{dt} - \frac{dy^*(x)}{dt} \right) \| \leq \frac{1}{\delta} \cdot \| g_{xy}(x, y^*(x)) \| \cdot \int_{t=0}^{\delta} \left( \int_{s=0}^{t} \| \frac{d^2}{ds^2} y^*_s(x) \| ds \right) dt$$

$$\leq \frac{1}{\delta} \cdot \| g_{xy}(x, y^*(x)) \| \cdot \max_{s \in [0,\delta]} \| \frac{d^2}{ds^2} y^*_s(x) \| \cdot \delta^2$$

$$\leq \delta \cdot \| g_{xy}(x, y^*(x)) \| \cdot \max_{s \in [0,\delta]} \| \frac{d^2}{ds^2} y^*_s(x) \|$$

$$= \delta \cdot C_g \cdot C_y,$$

$$(B.11)$$

where $C_g$ is the bound on smoothness of $g$ as in Assumption 2.3, and $C_y$ is the bound on $\|\frac{d^2 y^*}{dx^2}\|$ from Lemma 3.3. For the terms involving the function $h$, we have

$$
\begin{aligned}
\|\frac{\lambda_\delta^* - \lambda^*}{\delta} - \frac{d\lambda_\delta^*}{d\delta}|_{\delta=0}\| &= \frac{1}{\delta} \int_{t=0}^\delta \|\frac{d\lambda_t^*}{dt} - \frac{d\lambda_\delta^*}{d\delta}|_{\delta=0}\| dt \\
&= \frac{1}{\delta} \int_{t=0}^\delta \int_{s=0}^t \|\frac{d^2}{ds^2}\lambda_s^*\| ds \cdot dt \\
&\leq \frac{1}{\delta} \max_{s\in[0,\delta]} \|\frac{d^2}{ds^2}\lambda_s^*\| \cdot \delta^2 \leq \delta \cdot \max_{s\in[0,\delta]} \|\frac{d^2}{ds^2}\lambda_s^*\| \\
&= \delta \cdot C_\ell,
\end{aligned}
\tag{B.12}
$$

where $C_\ell$ is the bound on $\|\frac{d^2\lambda^*}{ds^2}\|$ from Lemma 3.3. Combining (B.9), Inequality (B.10), Inequality (B.11), and Inequality (B.12), along with Lemma B.5, Corollary B.3, and Lemma 3.3, we have that overall bound is

$$
\delta \cdot (S_g M_y^2 + C_g C_y + C_\ell) \leq O(\delta \cdot (S_g \cdot C_H^3 \cdot (C_g + \|A\|)^2 \cdot (C_g + C_f + L_f))).
$$

$\square$

## B.2  Cost of linear equality constrained bilevel program

---
**Algorithm 5** The Fully First-Order Method for Bilevel Equality Constrained Problem
---
1: **Input:** Current $x_0$, accuracy $\epsilon$, perturbation $\delta = \epsilon^2/8C_F^2 R_\mathcal{X}$ with $C_F = 2(L_f + C_f + C_g)C_H^3 S_g(L_g + \|A\|)^2$, accuracy for the lower level problem $\tilde{\delta} = 2(C_g + \|A\|)\delta^2$.

2: **for** t=0,1,2,... **do**

3:  Run Algorithm 6 to generate $\tilde{\delta}$-accurate primal and dual solutions $(\hat{y}^*, \hat{\lambda}^*)$ for

$$
\min_{y:Ax_t+By=b} g(x_t, y)
$$

4:  Run Algorithm 6 to generate $\tilde{\delta}$-accurate primal and dual solutions $(\hat{y}_\delta^*, \hat{\lambda}_\delta^*)$ for

$$
\min_{y:Ax_t+By=b} g(x_t, y) + \delta f(x_t, y)
$$

5:  Compute $\hat{v}_t := \frac{\nabla_x[g(x_t,\hat{y}_\delta^*)+\hat{\lambda}_\delta^* h(x,\hat{y}^*)] - \nabla_x[g(x_t,\hat{y}^*)+\hat{\lambda}^* h(x,\hat{y}^*)]}{\delta}$, set

$$
\widetilde{\nabla}F(x_t) := \hat{v}^t + \nabla_x f(x, \hat{y}^*(x)).
$$

6:  Set $x_{t+1} \leftarrow \arg\min_{z\in\mathcal{X}} \|z - (x_t - \frac{1}{C_F}\widetilde{\nabla}F(x_t))\|^2$.

---

**Theorem 3.1.** *Consider Problem 3.1 under Assumption 2.2, and let $\kappa = C_g/\mu_g$ be the condition number of $g$. Then Algorithm 5 finds an $\epsilon$-stationary point (in terms of gradient mapping, see (B.14)) after $T = \widetilde{O}(C_F(F(x_0) - \inf F)\sqrt{\kappa}\epsilon^{-2})$ oracle calls to $f$ and $g$, where $C_F := 2(L_f + C_f + C_g)C_H^3 S_g(L_g + \|A\|)^2$ is the smoothness constant of the hyperobjective $F$.*

*Proof.* We first show the inexact gradient $\widetilde{\nabla}F(x_t)$ generated in Algorithm 5 is an $\delta$-accurate approximation to the hyper-gradient $\nabla F(x_t)$. Consider the inexact gradient defined in (3.2)

$$
\begin{aligned}
\|v_t - \hat{v}_t\| &\leq \frac{1}{\delta}\{\|[\nabla_x g(x_t, \hat{y}_\delta^*) - \nabla_x[g(x_t, \hat{y}^*)] - [\nabla_x g(x_t, y_\delta^*) - \nabla_x[g(x_t, y^*)]\| \\
&\quad + \|\hat{\lambda}_\delta^* - \hat{\lambda}^* - [\lambda_\delta^* - \lambda^*]\|\|A\|\} \\
&\leq \frac{2}{\delta}[C_g + \|A\|]\tilde{\delta}.
\end{aligned}
$$

Thus we get

$$\|\widetilde{\nabla}F(x_t) - \nabla F(x_t)\| \leq \|\nabla_x f(x_t, y^*) - \nabla_x f(x_t, \hat{y}^*)\| + \|\hat{v}^t - v^t\| + \|v^t - \frac{dy^*(x^t)}{dx}\nabla_y f(x_t, y^*(x_t))\|$$

$$\leq C_f\tilde{\delta} + \frac{2}{\delta}[C_g + \|A\|]\tilde{\delta} + C_F\delta$$

$$\leq \frac{2\tilde{\delta}}{\delta}[C_f + C_g + \|A\|] + C_F\delta$$

$$\leq \frac{\epsilon^2}{4C_F R_{\mathcal{X}}}.$$

Applied to the $C_F$-smooth hyper-objective $F$, such an inexact gradient oracle satisfies the requirement for Proposition B.6. Thus an $\epsilon$-stationary point with $\|\mathcal{G}_F(x^t)\| \leq \epsilon$ (see Eq. (B.14) for the definition of gradient mapping) must be found in $N = O(\frac{C_F(F(x^0)-F^*)}{\epsilon^2})$ iterations. Noting the evaluation of inexact solutions $(\hat{y}^*, \hat{\lambda}^*, \hat{y}^*_\delta, \hat{\lambda}^*_\delta)$ requires $\tilde{O}(\sqrt{C_g/\mu_g})$ first order oracle evaluations, we arrive at the total oracle complexity of $\tilde{O}(\sqrt{C_g/\mu_g}\frac{C_F(F(x^0)-F^*)}{\epsilon^2})$ for finding an $\epsilon$-stationary point. $\qquad\square$

### B.3 The cost of inexact projected gradient descent method

In this subsection, we state the number of iterations required by projected gradient descent method to find an $\epsilon$-stationary point using inexact gradient oracles. Specifically, we consider the following non-convex smooth problem where the objective $F$ is assumed to be $C_F$-Lipschitz smooth:

$$\text{minimize}_{x\in\mathcal{X}} F(x). \tag{B.13}$$

Since the feasible region $\mathcal{X}$ is compact, we use the norm of the following gradient mapping $\mathcal{G}_F(x)$ as the stationarity criterion

$$\mathcal{G}_F(x) := C_F(x - x^+) \text{ where } x^+ = \arg\min_{z\in\mathcal{X}}\left\|z - \left(x - \frac{1}{C_F}\nabla F(x)\right)\right\|^2. \tag{B.14}$$

Initialized to some $x_0$ and the inexact gradient oracle $\widetilde{\nabla}F$, the updates of the inexact projected gradient descent method is given by

**For** t=1,2,..., N **do**:

$$\text{Set } x_t \leftarrow \arg\min_{z\in\mathcal{X}}\left\|z - \left(x_{t-1} - \frac{1}{C_F}\widetilde{\nabla}F(x_{t-1})\right)\right\|^2. \tag{B.15}$$

The next proposition calculates the complexity result.

**Proposition B.6.** *Consider the constrained optimization problem in* (B.13) *with $F$ being $C_F$-Lipschitz smooth and $\mathcal{X}$ having a radius of $R$. When supplied with a $\delta = \epsilon^2/4C_F R$ -inexact gradient oracle $\widetilde{\nabla}F$, that is, $\|\nabla F(x) - \widetilde{\nabla}F(x)\| \leq \delta$, the solution generated by the projected gradient descent method* (B.15) *satisfies*

$$\min_{t\in[N]}\|\mathcal{G}_F(x_t)\|^2 \leq \frac{C_F(F(x_0) - F^*)}{N} + \delta C_F R,$$

*that is, it takes at most $O(\frac{C_F(F(x^0)-F^*)}{\epsilon^2})$ iterations to generate some $\bar{x}$ with $\|\mathcal{G}_F(x)\| \leq \epsilon$.*

*Proof.* By $C_F$-smoothness of $F$, we have

$$f(x_{t+1}) = f(x_t - \frac{1}{C_F}\widetilde{\mathcal{G}}_F(x_t)) \leq f(x_t) - \frac{1}{C_F}\widetilde{\mathcal{G}}_F(x_t)^\top\nabla f(x_t) + \frac{1}{2C_F}\|\widetilde{\mathcal{G}}_F(x_t)\|^2$$

$$= f(x_t) - \frac{1}{2C_F}\|\widetilde{\mathcal{G}}_F(x_t)(x_t)\|^2 + \frac{1}{C_F}\widetilde{\mathcal{G}}_F(x_t)^\top(\widetilde{\mathcal{G}}_F(x_t) - \nabla f(x_t)).$$
$$\tag{B.16}$$

We now show that $\frac{1}{\beta}\widetilde{\mathcal{G}}_F(x_t)^\top(\widetilde{\mathcal{G}}_F(x_t) - \nabla f(x_t)) \leq 0$. Let $\widetilde{y}_t = x_t - \frac{1}{C_F}\widetilde{\nabla}F(x_t)$, and let $y_t = x_t - \frac{1}{C_F}\nabla f(x_t)$. Then have that

$$
\begin{aligned}
\frac{1}{C_F}\widetilde{\mathcal{G}}_F(x_t)^\top(\frac{1}{C_F}\widetilde{\mathcal{G}}_F(x_t) - \nabla f(x_t)) &= C_F(x_t - \mathrm{proj}_{\mathcal{X}}(\widetilde{y}_t))^\top(y_t - \mathrm{proj}_{\mathcal{X}}(\widetilde{y}_t)) \\
&= C_F(x_t - \mathrm{proj}_{\mathcal{X}}(\widetilde{y}_t))^\top(\widetilde{y}_t - \mathrm{proj}_{\mathcal{X}}(\widetilde{y}_t)) \\
&\quad + C_F(x_t - \mathrm{proj}_{\mathcal{X}}(\widetilde{y}_t))^\top(y_t - \widetilde{y}_t) \\
&\leq C_F(x_t - \mathrm{proj}_{\mathcal{X}}(\widetilde{y}_t))^\top(y_t - \widetilde{y}_t) \\
&\leq \delta C_F R,
\end{aligned}
$$

where the penultimate inequality uses the fact that $\mathcal{X}$ is a convex set, and $R$ is the diameter of the set $X$. Combining this with Inequality (B.16), we have that the function decrease per iteration is

$$
F(x_{t+1}) \leq F(x_t) - \frac{1}{2C_F}\|\widetilde{\mathcal{G}}_F(x_t)\|^2 + \delta C_F R.
$$

Summing over $N$ iterations telescopes the terms, we get

$$
\min_{t\in[N]}\|\widetilde{\mathcal{G}}_F(x_t)\|^2 \leq \frac{1}{N}C_F(F(x^0) - F^*) + \delta C_F R.
$$

Substituting in $N = \frac{4}{\epsilon^2}C_F(F(x^0) - F^*)$ and the choice of $\delta = \epsilon^2/4C_F R$, we get

$$
\min_{t\in[N]}\|\widetilde{\mathcal{G}}_F(x_t)\|^2 \leq \frac{\epsilon^2}{2}.
$$

Taking into account the fact that $\|\widetilde{\mathcal{G}}_F(x_t) - \mathcal{G}_F(x_t)\| \leq \|\nabla F(x^t) - \widetilde{\nabla}F(x^t)\| \leq \delta$, we obtain the desired result.

$\square$

## B.4 The cost of generating approximate solutions to the linearly constrained LL problem

In this subsection, we address the issue of generating approximations to the primal and dual solutions $(y^*, \lambda^*)$ associated with the lower-level problem in Problem 3.1. These approximations are required for computing the approximate hypergradient in Algorithm 1. For notational simplicity, we are going to consider the following constrained strongly convex problem:

$$
\begin{aligned}
\text{minimize}_{y\in\mathbb{R}^d} \quad & g(y) \\
\text{subject to} \quad & By = b.
\end{aligned} \tag{B.17}
$$

We propose the following simple scheme to generate approximate solutions to Problem B.17.

> Compute a feasible $\hat{y}$ such that $\|\hat{y} - y^*\| \leq \delta$. Then solve
> $$
> \hat{\lambda} = \arg\min_{\lambda\in\mathbb{R}^m} \|\nabla_y g(\hat{y}) - B^\top\lambda\|^2. \tag{B.18}
> $$

The following lemma tells us that $\hat{\lambda}$ is close to $\lambda^*$ if $B$ has full row rank.

**Lemma B.7.** *Suppose $g$ in Problem B.17 is a $C_g$-Lipschitz smooth, and the matrix $B$ has full row rank such that the following matrix $M_B$ is invertible*

$$
M_B = \begin{bmatrix} I & B^\top \\ B & 0 \end{bmatrix}.
$$

*Then the approximate solution $(\hat{\lambda}, \hat{y})$ from (B.18) satisfies $\|\hat{\lambda} - \lambda^*\| \leq \|M_B^{-1}\|(1 + C_g)\delta$.*

*Proof.* Since $(\lambda^*, y^*)$ satisfy the KKT conditions, they are the solution to the following linear system

$$
\underbrace{\begin{bmatrix} I & B^\top \\ B & 0 \end{bmatrix}}_{=M_B}\begin{bmatrix} y^* \\ \lambda^* \end{bmatrix} = \begin{bmatrix} -\nabla_y g(y^*) + Iy^* \\ b \end{bmatrix}. \tag{B.19}
$$

That is

$$\begin{bmatrix} y^* \\ \lambda^* \end{bmatrix} = M_B^{-1} \begin{bmatrix} -\nabla_y g(y^*) + Iy^* \\ b \end{bmatrix}.$$

On the other hand, the approximate solutions $(\hat{y}, \hat{\lambda})$ in (B.18) satisfies

$$\begin{bmatrix} I & B^\top \\ B & 0 \end{bmatrix} \begin{bmatrix} \hat{y} \\ \hat{\lambda} \end{bmatrix} = \begin{bmatrix} B^\top \hat{\lambda} + I\hat{y} \\ b \end{bmatrix}.$$

We show the right hand side (r.h.s) of the above equation to be close to the r.h.s of (B.19). Let $S := \{B^\top \lambda : \lambda \in \mathbb{R}^m\}$ denote the subspace spanned by the rows of $B$. We can rewrite $B^\top \hat{\lambda}$ as the projection of $\nabla g(\hat{y})$ onto $S$, that is,

$$B^\top \hat{\lambda} = \arg\min_{s \in S} \|\nabla_y g(\hat{y}) - s\|^2$$
$$-\nabla_y g(y^*) = B^\top \lambda^* = \arg\min_{s \in S} \|\nabla_y g(y^*) - s\|^2,$$

where the second relation follows from the KKT conditon associated with $(\lambda^*, y^*)$. Since the projection is an non-expansive operation, we have

$$\|B^\top \hat{\lambda} - (-\nabla_y g(y^*))\| = \|B^\top \hat{\lambda} - B^\top \lambda^*\| \leq \|\nabla_y g(\hat{y}) - \nabla g(y^*)\| \leq C_g \|\hat{y} - y^*\| \leq C_g \delta.$$

We can rewrite $(\hat{y}, \hat{\lambda})$ as solutions to the following linear system with some $\|\tau\| \leq (1 + C_g)\delta$,

$$\begin{bmatrix} \hat{y} \\ \hat{\lambda} \end{bmatrix} = M_B^{-1} \begin{bmatrix} -\nabla_y g(y^*) + Iy^* + \tau \\ b \end{bmatrix}.$$

Thus we get

$$\| \begin{bmatrix} \hat{y} \\ \hat{\lambda} \end{bmatrix} - \begin{bmatrix} y^* \\ \lambda^* \end{bmatrix} \| = \|M_B^{-1}\| \| \begin{bmatrix} \tau \\ 0 \end{bmatrix} \| \leq \|M_B^{-1}\|(1 + C_g)\delta.$$

$\square$

Now we can just use the AGD method to generate a close enough approximate solution $\hat{y}$ and call up the Subroutine in (B.18) to generate the approximate dual solution $\hat{\lambda}$.

---

**Algorithm 6** The Projected Gradient Method to Generate Primal and Dual Solutions for a Linearly Constrained Problem

1: **Input**: accuracy requirement $\epsilon > 0$ and linearly constrained problem $\min_{y:By=b} g(y)$.
2: Starting from $y^0 = 0$ and using $Y := \{y \in \mathbb{R}^d : By = b\}$ as the simple feasible region.
3: Run the Accelerated Gradient Descent (AGD) Method (Section 3.3 in [91]) for $N = \lceil 4\sqrt{C_g/\mu_g} \log \frac{\|y^*\| \|M_B^{-1}\|(C_g+1)}{\mu_g \epsilon} \rceil$ iterations.
4: Use the $y^N$ as the approximate solution $\hat{y}$ to generate $\hat{\lambda}$ according to (B.18).
5: **return** $(\hat{y}, \hat{\lambda})$

---

**Proposition B.8.** *Suppose the objective function $g$ is both $L_g$-smooth and $\mu_g$-strongly convex, and that the constraint satisfies the assumption in Lemma B.7. Fix an $\epsilon > 0$, the solution $(\hat{y}, \hat{\lambda})$ returned by the above procedure satisfies $\|y^* - \hat{y}\| \leq \epsilon$ and $\|\hat{\lambda} - \lambda^*\| \leq \epsilon$. In another words, the cost of generating $\epsilon$-close primal and dual solutions are bounded by $O(\sqrt{\frac{C_g}{\mu_g}} \log \frac{1}{\epsilon})$.*

*Proof.* With $N := \lceil 4\sqrt{C_g/\mu_g} \log \frac{\|y^*\| \|M_B^{-1}\|(L_g+1)}{\mu_g \epsilon} \rceil$, Theorem 3.7 in [91] shows that $\|y^N - \hat{y}\| \leq \epsilon/\|M_B^{-1}\|(1 + L_g)$. Then we can apply Lemma B.7 to obtain the desired bound. $\square$

# C  Proofs for Section 4

Our algorithms are based on the Lipschitzness of $F$, which we prove below.

**Lemma 4.3.** *Under Assumption 2.2 and 2.5, $F$ in Problem 4.1 is $O(L_f L_y)$-Lipschitz in $x$.*

*Proof.* By Lemma 2.1 of [16], the hypergradient of $F$ computed with respect to the variable $x$ may be expressed as $\nabla_x F(x) = \nabla_x f(x, y^*(x)) + \left(\frac{dy^*(x)}{dx}\right)^\top \cdot \nabla_y f(x, y^*(x))$. Since we impose Lipschitzness on $f$ and $y^*$, we can bound each of the terms of $\nabla_x F(x)$ by the claimed bound. $\quad\square$

## C.1  Faster algorithm for low upper-level dimensions

In this section we analyze Algorithm 2, which as stated in Section 4, requires evaluating only the hyperobjective $F$ (as opposed to estimating the hypergradient in Algorithm 3).

The motivation for designing such an algorithm, is that while evaluating $\nabla F$ up to $\alpha$ accuracy requires $O(\alpha^{-1})$ gradient evaluations, the hyperobjective value can be estimated at a linear rate:

**Lemma 4.2** (Proof in Appendix C.1). *Given any $x$, we can return $\widetilde{F}(x)$ such that $|F(x) - \widetilde{F}(x)| \leq \alpha$ using $O(\sqrt{C_g/\mu_g} \log(L_f/\alpha))$ first-order oracle calls to $f$ and $g$.*

*Proof of Lemma 4.2.* We note that it suffices to find $\tilde{y}^*$ such that $\|\tilde{y}^* - y^*(x)\| \leq \alpha/L_f$, since setting $\widetilde{F}(x) := f(x, \tilde{y}^*)$ will then satisfy $|\widetilde{F}(x) - F(x)| = |f(x, \tilde{y}^*) - f(x, y^*(x))| \leq L_f \cdot \frac{\alpha}{L_f} = \alpha$ by Lispchitzness of $f$, as required. Noting that $y^*(x) = \arg\min_{h(x,y)\leq 0} g(x, y)$ is the solution to a constrained smooth, strongly-convex problem with condition number $C_g/\mu_g$, it is possible to approximate it up to $\alpha/L_f$ with $O(\sqrt{C_g/\mu_g} \log(L_f/\alpha))$ first-order oracle calls using the result of Zhang and Lan [45]. $\quad\square$

Accordingly, we consider Algorithm 2, which is a zero-order variant of Algorithm 3, whose guarantee is summarized is the theorem below.

**Theorem C.1.** *Suppose $F : \mathbb{R}^d \to \mathbb{R}$ is $L$-Lipschitz, and that $|\widetilde{F}(\cdot) - F(\cdot)| \leq \alpha$. Then running Algorithm 3 with $\rho = \min\left\{\frac{\delta}{2}, \frac{F(x_0) - \inf F}{L}\right\}, \nu = \delta - \rho, D = \Theta\left(\frac{\nu\epsilon^2\rho^2}{d\rho^2 L^2 + \alpha^2 d^2}\right), \eta = \Theta\left(\frac{\nu\epsilon^3\rho^4}{(d\rho^2 L^2 + \alpha^2 d^2)^2}\right)$, outputs a point $x^{\mathrm{out}}$ such that $\mathbb{E}[\mathrm{dist}(0, \partial_\delta F(x^{\mathrm{out}}))] \leq \epsilon + \alpha$ with*

$$T = O\left(\frac{d(F(x_0) - \inf F)}{\delta\epsilon^3} \cdot \left(L^2 + \alpha^2\left(\frac{d}{\delta^2} + \frac{dL^2}{(F(x_0) - \inf F)^2}\right)\right)\right) \text{ calls to } \widetilde{F}(\cdot).$$

Combining the result of Theorem C.1 with the complexity of hyperobjective estimation, as given by Lemma 4.2, we obtain convergence to a $(\delta, \epsilon)$-stationary point of Problem 4.1 with $\widetilde{O}(d_x\delta^{-1}\epsilon^{-3})$ gradient calls overall.

### C.1.1  Proof of Theorem C.1

Denoting the uniform randomized smoothing $F_\rho(x) := \mathbb{E}_{\|z\|\leq 1}[F(x + \rho \cdot z)]$ where the expectation, here and in what follows, is taken with respect to the uniform measure, it is well known [92, Lemma 10] that

$$\mathbb{E}_{\|w\|=1}\left[\frac{d}{2\rho}(F(x + \rho w) - F(x - \rho w))w\right] = \nabla F_\rho(x),$$

$$\mathbb{E}_{\|w\|=1}\left\|\nabla F_\rho(x) - \frac{d}{2\rho}(F(x + \rho w) - F(x - \rho w))w\right\|^2 \lesssim dL^2. \tag{C.1}$$

We first show that replacing the gradient estimator with the inexact evaluations $\widetilde{F}(\cdot)$ leads to a biased gradient estimator of $F$.

**Lemma C.2.** *Suppose $|F(\cdot) - \widetilde{F}(\cdot)| \leq \alpha$. Denoting*

$$g_x = \frac{d}{2\rho}(F(x + \rho w) - F(x - \rho w))w,$$

$$\tilde{g}_x = \frac{d}{2\rho}(\widetilde{F}(x + \rho w) - \widetilde{F}(x - \rho w))w,$$

*it holds that*

$$\mathbb{E}_{\|w\|=1} \|g_x - \widetilde{g}_x\| \le \frac{\alpha d}{\rho} , \quad \textit{and} \quad \mathbb{E}_{\|w\|=1} \|\widetilde{g}_x\|^2 \lesssim \frac{\alpha^2 d^2}{\rho^2} + dL^2 .$$

*Proof.* For the first bound, we have

$$\mathbb{E}_{\|w\|=1} \|g_x - \widetilde{g}_x\| \le \frac{d}{2\rho}(2\alpha)\mathbb{E}_{\|w\|=1} \|w\| = \frac{\alpha d}{\rho} ,$$

while for the second bound

$$\mathbb{E}_{\|w\|=1} \|\widetilde{g}_x\|^2 = \mathbb{E}_{\|w\|=1} \|\widetilde{g}_x - g_x + g_x\|^2 \le 2\mathbb{E}_{\|w\|=1} \|\widetilde{g}_x - g_x\|^2 + 2\mathbb{E}_{\|w\|=1} \|g_x\|^2 \lesssim \frac{d^2}{\rho^2}\cdot\alpha^2 + dL^2 ,$$

where the last step invoked (C.1). $\qquad\qquad\square$

We are now ready to analyze Algorithm 2. We denote $\alpha' = \frac{\alpha d}{\rho}$, $\widetilde{G} = \sqrt{\frac{\alpha^2 d^2}{\rho^2} + dL^2}$. Since $x_t = x_{t-1} + \Delta_t$, we have

$$
\begin{aligned}
F_\rho(x_t) - F_\rho(x_{t-1}) &= \int_0^1 \langle \nabla F_\rho(x_{t-1} + s\Delta_t), \Delta_t \rangle \, ds \\
&= \mathbb{E}_{s_t \sim \mathrm{Unif}[0,1]} \left[ \nabla F_\rho(x_{t-1} + s_t\Delta_t), \Delta_t \right] \\
&= \mathbb{E}\left[ \langle \nabla F_\rho(z_t), \Delta_t \rangle \right] .
\end{aligned}
$$

By summing over $t \in [T] = [K \times M]$, we get for any fixed sequence $u_1, \ldots, u_K \in \mathbb{R}^d$ :

$$
\begin{aligned}
\inf F_\rho \le F_\rho(x_T) &\le F_\rho(x_0) + \sum_{t=1}^{T} \mathbb{E}\left[ \langle \nabla F_\rho(z_t), \Delta_t \rangle \right] \\
&= F_\rho(x_0) + \sum_{k=1}^{K} \sum_{m=1}^{M} \mathbb{E}\left[ \langle \nabla F_\rho(z_{(k-1)M+m}), \Delta_{(k-1)M+m} - u_k \rangle \right] \\
&\quad + \sum_{k=1}^{K} \sum_{m=1}^{M} \mathbb{E}\left[ \langle \nabla F_\rho(z_{(k-1)M+m}), u_k \rangle \right] \\
&\le F_\rho(x_0) + \sum_{k=1}^{K} \mathrm{Reg}_M(u_k) + \sum_{k=1}^{K} \sum_{m=1}^{M} \mathbb{E}\left[ \langle \nabla F_\rho(z_{(k-1)M+m}), u_k \rangle \right] \\
&\le F_\rho(x_0) + KD\widetilde{G}\sqrt{M} + K\alpha'DM + \sum_{k=1}^{K} \sum_{m=1}^{M} \mathbb{E}\left[ \langle \nabla F_\rho(z_{(k-1)M+m}), u_k \rangle \right]
\end{aligned}
$$

where the last inequality follows by combining Lemma C.2 and Lemma C.3. By setting $u_k := -D \frac{\sum_{m=1}^{M} \nabla F_\rho(z_{(k-1)M+m})}{\left\| \sum_{m=1}^{M} \nabla F_\rho(z_{(k-1)M+m}) \right\|}$, rearranging and dividing by $DT = DKM$ we obtain

$$
\begin{aligned}
\frac{1}{K} \sum_{k=1}^{K} \mathbb{E} \left\| \frac{1}{M} \sum_{m=1}^{M} \nabla F_\rho(z_{(k-1)M+m}) \right\| &\le \frac{F_\rho(x_0) - \inf F_\rho}{DT} + \frac{\widetilde{G}}{\sqrt{M}} + \alpha' \\
&= \frac{F_\rho(x_0) - \inf F_\rho}{K\nu} + \frac{\sqrt{\frac{\alpha^2 d^2}{\rho^2} + L^2 d}}{\sqrt{M}} + \frac{\alpha d}{\rho} \\
&\le \frac{F_\rho(x_0) - \inf F_\rho}{K\nu} + \frac{\frac{\alpha d}{\rho}}{\sqrt{M}} + \frac{L\sqrt{d}}{\sqrt{M}} + \frac{\alpha d}{\rho} . \qquad \text{(C.2)}
\end{aligned}
$$

Finally, note that for all $m \in [M]$ : $\left\| z_{(k-1)M+m} - \overline{x}_k \right\| \le MD \le \nu$, therefore $\nabla F_\rho(z_{(k-1)M+m}) \in \partial_\nu F_\rho(\overline{x}_k) \subset \partial_\delta F(\overline{x}_k)$, where the last containment is due to [46, Lemma 4] by using our assignment $\rho + \nu = \delta$. Invoking the convexity of the Goldstein subdifferential, this implies that

$$\frac{1}{M} \sum_{m=1}^{M} \nabla F_\rho(z_{(k-1)M+m}) \in \partial_\delta F(\overline{x}_k) ,$$

thus it suffices to bound the first three summands in (C.2) by $\epsilon$ in order to finish the proof. This happens as long as $\frac{F_\rho(x_0) - \inf F_\rho}{K\nu} \le \frac{\epsilon}{3}$, $\frac{\frac{\alpha d}{\rho}}{\sqrt{M}} \le \frac{\epsilon}{3}$, and $\frac{L\sqrt{d}}{\sqrt{M}} \le \frac{\epsilon}{3}$, which imply $K \gtrsim \frac{F_\rho(x_0) - \inf F_\rho}{\nu\epsilon}$, $M \gtrsim \frac{\alpha^2 d^2}{\rho^2\epsilon^2}$, and $M \gtrsim \frac{L^2 d}{\epsilon^2}$. By our assignments of $\rho$ and $\nu$, these result in

$$
\begin{aligned}
T = KM &= O\left(\frac{F_\rho(x_0) - \inf F_\rho}{\nu\epsilon} \cdot \left(\frac{\alpha^2 d^2}{\rho^2\epsilon^2} + \frac{L^2 d}{\epsilon^2}\right)\right) \\
&= O\left(\frac{(F(x_0) - \inf F)d}{\delta\epsilon^3} \cdot \left(\frac{\alpha^2 d}{\rho^2} + L^2\right)\right) \\
&= O\left(\frac{(F(x_0) - \inf F)d}{\delta\epsilon^3} \cdot \left(\alpha^2 d \cdot \max\left\{\frac{1}{\delta^2}, \frac{L^2}{(F(x_0) - \inf F)^2}\right\} + L^2\right)\right),
\end{aligned}
$$

completing the proof.

## C.2  Proof of Theorem 4.4

We recall Theorem 4.4 below to keep this section self-contained.

**Theorem 4.4.** *Suppose $F : \mathbb{R}^d \to \mathbb{R}$ is L-Lipschitz and that $\|\widetilde{\nabla} F(\cdot) - \nabla F(\cdot)\| \le \alpha$. Then running Algorithm 3 with $D = \Theta(\frac{\delta\epsilon^2}{L^2})$, $\eta = \Theta(\frac{\delta\epsilon^3}{L^4})$, outputs a point $x^{\text{out}}$ such that $\mathbb{E}[\text{dist}(0, \partial_\delta F(x^{\text{out}}))] \le \epsilon + \alpha$, with $T = O\left(\frac{(F(x_0) - \inf F)L^2}{\delta\epsilon^3}\right)$ calls to $\widetilde{\nabla} F(\cdot)$.*

Our analysis is inspired by the reduction from online learning to nonconvex optimization given by [47]. To that end, we start by proving a seemingly unrelated result, asserting that online gradient descent minimizes the regret with respect to inexact evaluations. Recalling standard definitions from online learning, given a sequence of linear losses $\ell_m(\cdot) = \langle g_m, \cdot \rangle$, if an algorithm chooses $\Delta_1, \ldots, \Delta_M$ we denote the regret with respect to $u$ as

$$
\text{Reg}_M(u) := \sum_{m=1}^M \langle g_m, \Delta_m - u \rangle.
$$

Consider an update rule according to online projected *inexact* gradient descent:

$$
\Delta_{m+1} := \text{clip}_D(\Delta_m - \eta_m \widetilde{g}_m).
$$

**Lemma C.3** (Inexact Online Gradient Descent). *In the setting above, suppose that $(\widetilde{g}_m)_{m=1}^M$ are possibly randomized vectors, such that $\mathbb{E}\|\widetilde{g}_m - g_m\| \le \alpha$ and $\mathbb{E}\|\widetilde{g}_m\|^2 \le \widetilde{G}^2$ for all $m \in [M]$. Then for any $\|u\| \le D$ it holds that*

$$
\mathbb{E}[\text{Reg}_M(u)] \le \frac{D^2}{\eta_M} + \widetilde{G}^2 \sum_{m=1}^M \eta_m + \alpha DM,
$$

*where the expectation is with respect to the (possible) randomness of $(\widetilde{g}_m)_{m=1}^M$. In particular, setting $\eta_m \equiv \frac{D}{\widetilde{G}\sqrt{M}}$ yields*

$$
\mathbb{E}[\text{Reg}_M(u)] \lesssim D\widetilde{G}\sqrt{M} + \alpha DM.
$$

*Proof.* For any $m \in [M]$:

$$
\begin{aligned}
\|\Delta_{m+1} - u\|^2 &= \|\text{clip}_D(\Delta_m - \eta_m \widetilde{g}_m) - u\|^2 \\
&\le \|\Delta_m - \eta_m \widetilde{g}_m - u\|^2 = \|\Delta_m - u\|^2 + \eta_m^2 \|\widetilde{g}_m\|^2 - 2\eta_m \langle \Delta_m - u, \widetilde{g}_m \rangle,
\end{aligned}
$$

thus

$$
\langle \widetilde{g}_m, \Delta_m - u \rangle \le \frac{\|\Delta_m - u\|^2 - \|\Delta_{m+1} - u\|^2}{2\eta_m} + \frac{\eta_m}{2}\|\widetilde{g}_m\|^2,
$$

from which we get that

$$
\begin{aligned}
\mathbb{E}\langle g_m, \Delta_m - u \rangle &= \mathbb{E}\langle \widetilde{g}_m, \Delta_m - u \rangle + \mathbb{E}\langle g_m - \widetilde{g}_m, \Delta_m - u \rangle \\
&\le \frac{\|\Delta_m - u\|^2 - \|\Delta_{m+1} - u\|^2}{2\eta_m} + \frac{\eta_m}{2}\mathbb{E}\|\widetilde{g}_m\|^2 + \mathbb{E}\|g_m - \widetilde{g}_m\| \cdot \|\Delta_m - u\| \\
&\le \frac{\|\Delta_m - u\|^2 - \|\Delta_{m+1} - u\|^2}{2\eta_m} + \frac{\eta_m}{2}\widetilde{G}^2 + \alpha D.
\end{aligned}
$$

Summing over $m \in [M]$, we see that

$$\mathbb{E}\left[\operatorname{Reg}_M(u)\right] \le \sum_{m=1}^{M} \|\Delta_m - u\|^2 \left(\frac{1}{\eta_m} - \frac{1}{\eta_{m-1}}\right) + \frac{\widetilde{G}^2}{2} \sum_{m=1}^{M} \eta_m + M\alpha D$$

$$\le \frac{D^2}{\eta_M} + \widetilde{G}^2 \sum_{m=1}^{M} \eta_m + \alpha DM \ .$$

The simplification for $\eta_m \equiv \frac{D}{\widetilde{G}\sqrt{M}}$ readily follows. $\qquad \square$

We are now ready to analyze Algorithm 3 in the inexact gradient setting.

*Proof of Theorem 4.4.* Since Algorithm 3 has $x_t = x_{t-1} + \Delta_t$, we have

$$F(x_t) - F(x_{t-1}) = \int_0^1 \langle \nabla F(x_{t-1} + s\Delta_t), \Delta_t \rangle \, ds$$

$$= \mathbb{E}_{s_t \sim \mathrm{Unif}[0,1]}\left[\langle \nabla F(x_{t-1} + s_t\Delta_t), \Delta_t \rangle\right]$$

$$= \mathbb{E}\left[\langle \nabla F(z_t), \Delta_t \rangle\right] \ .$$

By summing over $t \in [T] = [K \times M]$, we get for any fixed sequence $u_1, \ldots, u_K \in \mathbb{R}^d$ :

$$\inf F \le F(x_T) \le F(x_0) + \sum_{t=1}^{T} \mathbb{E}\left[\langle \nabla F(z_t), \Delta_t \rangle\right]$$

$$= F(x_0) + \sum_{k=1}^{K} \sum_{m=1}^{M} \mathbb{E}\left[\langle \nabla F(z_{(k-1)M+m}), \Delta_{(k-1)M+m} - u_k \rangle\right]$$

$$+ \sum_{k=1}^{K} \sum_{m=1}^{M} \mathbb{E}\left[\langle \nabla F(z_{(k-1)M+m}), u_k \rangle\right]$$

$$\le F(x_0) + \sum_{k=1}^{K} \operatorname{Reg}_M(u_k) + \sum_{k=1}^{K} \sum_{m=1}^{M} \mathbb{E}\left[\langle \nabla F(z_{(k-1)M+m}), u_k \rangle\right]$$

$$\le F(x_0) + KD\widetilde{G}\sqrt{M} + K\alpha DM + \sum_{k=1}^{K} \sum_{m=1}^{M} \mathbb{E}\left[\langle \nabla F(z_{(k-1)M+m}), u_k \rangle\right]$$

where the last inequality follows from Lemma C.3 for $\widetilde{G} = \sqrt{L^2 + \alpha^2}$, $\eta = \frac{D}{\widetilde{G}\sqrt{M}}$, since $\|\widetilde{g}_t - \nabla F(z_t)\| \le \alpha$ (deterministically) for all $t \in [T]$ by assumption. Letting $u_k := -D\frac{\sum_{m=1}^{M} \nabla F(z_{(k-1)M+m})}{\left\|\sum_{m=1}^{M} \nabla F(z_{(k-1)M+m})\right\|}$, rearranging and dividing by $DT = DKM$, we obtain

$$\frac{1}{K} \sum_{k=1}^{K} \mathbb{E}\left\|\frac{1}{M} \sum_{m=1}^{M} \nabla F(z_{(k-1)M+m})\right\| \le \frac{F(x_0) - \inf F}{DT} + \frac{\widetilde{G}}{\sqrt{M}} + \alpha$$

$$= \frac{F(x_0) - \inf F}{K\delta} + \frac{\widetilde{G}}{\sqrt{M}} + \alpha \ . \qquad (\mathrm{C.3})$$

Finally, note that for all $k \in [K], m \in [M]$ : $\left\|z_{(k-1)M+m} - \overline{x}_k\right\| \le MD \le \delta$, therefore $\nabla F(z_{(k-1)M+m}) \in \partial_\delta F(\overline{x}_k)$. Invoking the convexity of the Goldstein subdifferential, we see that

$$\frac{1}{M} \sum_{m=1}^{M} \nabla F(z_{(k-1)M+m}) \in \partial_\delta F(\overline{x}_k) \ ,$$

thus it suffices to bound the first two summands on the right-hand side in (C.3) by $\epsilon$ in order to finish the proof. This happens as long as $\frac{F(x_0) - \inf F}{K\delta} \le \frac{\epsilon}{2}$ and $\frac{\widetilde{G}}{\sqrt{M}} \le \frac{\epsilon}{2}$. These are equivalent to $K \ge \frac{2(F(x_0) - \inf F)}{\delta\epsilon}$ and $M \ge \frac{4\widetilde{G}^2}{\epsilon^2}$, which results in

$$T = KM = O\left(\frac{F(x_0) - \inf F}{\delta\epsilon} \cdot \frac{L^2 + \alpha^2}{\epsilon^2}\right) = O\left(\frac{(F(x_0) - \inf F)L^2}{\delta\epsilon^3}\right),$$

completing the proof. □

## C.3 An implementation-friendly algorithm and its analysis

---

**Algorithm 7** Perturbed Inexact GD

---

1: **Input:** Inexact gradient oracle $\widetilde{\nabla} F : \mathbb{R}^d \to \mathbb{R}^d$, initialization $x_0 \in \mathbb{R}^d$, spatial parameter $\delta > 0$, step size $\eta > 0$, iteration budget $T \in \mathbb{N}$.
2: **for** $t = 0, \ldots, T - 1$ **do**
3:      Sample $w_t \sim \mathrm{Unif}(\mathbb{S}^{d-1})$
4:      $\widetilde{g}_t = \widetilde{\nabla} F(x_t + \delta \cdot w_t)$
5:      $x_{t+1} = x_t - \eta \widetilde{g}_t$
6: **Output:** $x^{\mathrm{out}} \sim \mathrm{Unif}\{x_0, \ldots, x_{T-1}\}$.

---

**Theorem C.4.** *Suppose $F : \mathbb{R}^d \to \mathbb{R}$ is $L$-Lipschitz, and that $\|\widetilde{\nabla} F(\cdot) - \nabla F(\cdot)\| \leq \alpha$. Then running Algorithm 7 with $\eta = \Theta\left( \frac{((F(x_0) - \inf F) + \delta L)^{1/2} \delta^{1/2}}{T^{1/2} L^{1/2} d^{1/4} (\alpha + L)} \right)$ outputs a point $x^{\mathrm{out}}$ such that $\mathbb{E}[\mathrm{dist}(0, \partial_\delta F(x^{\mathrm{out}}))] \leq \epsilon + \sqrt{\alpha L}$, with*

$$T = O\left( \frac{(F(x_0) - \inf F + \delta L) L^3 \sqrt{d}}{\delta \epsilon^4} \right) \text{ calls to } \widetilde{\nabla} F(\cdot).$$

*Proof.* Throughout the proof we denote $z_t = x_t + \delta \cdot w_t$. Since $F$ is $L$-Lipschitz, $F_\delta(x) := \mathbb{E}_{w \sim \mathrm{Unif}(\mathbb{S}^{d-1})}[F(x + \delta \cdot w)]$ is $L$-Lipschitz and $O(L\sqrt{d}/\delta)$-smooth. By smoothness we get

$$F_\delta(x_{t+1}) - F_\delta(x_t) \leq \langle \nabla F_\delta(x_t), x_{t+1} - x_t \rangle + O\left( \frac{L\sqrt{d}}{\delta} \right) \cdot \|x_{t+1} - x_t\|^2$$

$$= -\eta \langle \nabla F_\delta(x_t), \widetilde{g}_t \rangle + O\left( \frac{\eta^2 L\sqrt{d}}{\delta} \right) \cdot \|\widetilde{g}_t\|^2$$

$$= -\eta \langle \nabla F_\delta(x_t), \nabla F(z_t) \rangle - \eta \langle \nabla F_\delta(x_t), \widetilde{g}_t - \nabla F(z_t) \rangle + O\left( \frac{\eta^2 L\sqrt{d}}{\delta} \right) \cdot \|\widetilde{g}_t\|^2 .$$

Noting that $\mathbb{E}[\nabla F(z_t)] = \nabla F_\delta(x_t)$ and that $\|\widetilde{g}_t\| \leq \|\widetilde{g}_t - \nabla F(z_t)\| + \|\nabla F(z_t)\| \leq \alpha + L$, we see that

$$\mathbb{E}[F_\delta(x_{t+1}) - F_\delta(x_t)] \leq -\eta \mathbb{E} \|\nabla F_\delta(x_t)\|^2 + \eta L \alpha + O\left( \frac{\eta^2 L\sqrt{d}}{\delta} (\alpha + L)^2 \right) ,$$

which implies

$$\mathbb{E} \|\nabla F_\delta(x_t)\|^2 \leq \frac{\mathbb{E}[F_\delta(x_t)] - \mathbb{E}[F_\delta(x_{t+1})]}{\eta} + L\alpha + O\left( \frac{\eta L\sqrt{d}(\alpha + L)^2}{\delta} \right) .$$

Averaging over $t = 0, \ldots, T - 1$ and noting that $F_\delta(x_0) - \inf F_\delta \leq (F(x_0) - \inf F) + \delta L$ results in

$$\mathbb{E} \|\nabla F_\delta(x^{\mathrm{out}})\|^2 = \frac{1}{T} \sum_{t=0}^{T-1} E \|\nabla F_\delta(x_t)\|^2 \leq \frac{(F(x_0) - \inf F) + \delta L}{\eta T} + L\alpha + O\left( \frac{\eta L\sqrt{d}(\alpha + L)^2}{\delta} \right) .$$

By Jensen's inequality and the sub-additivity of the square root,

$$\mathbb{E} \|\nabla F_\delta(x^{\mathrm{out}})\| \leq \sqrt{\frac{(F(x_0) - \inf F) + \delta L}{\eta T}} + \sqrt{L\alpha} + O\left( \sqrt{\frac{\eta L\sqrt{d}(\alpha + L)^2}{\delta}} \right) .$$

Setting $\eta = \frac{\sqrt{((F(x_0)-\inf F)+\delta L)\delta}}{\sqrt{TL\sqrt{d}(\alpha+L)^2}}$ yields the final bound

$$\mathbb{E}\left\|\nabla F_\delta(x^{\text{out}})\right\| \lesssim \frac{((F(x_0)-\inf F)+\delta L)^{1/4}L^{1/4}d^{1/8}(\alpha+L)^{1/2}}{\delta^{1/4}T^{1/4}} + \sqrt{L\alpha}\,,$$

and the first summand is bounded by $\epsilon$ for $T = O\left(\frac{((F(x_0)-\inf F)+\delta L)L\sqrt{d}(L+\alpha)^2}{\delta\epsilon^4}\right)$.

$\square$

## D  Reformulation equivalence

**Theorem D.1** (Reformulation equivalence)**.** *When $\lambda^*$ matches to an optimal dual solution to the lower level problem $y^* = \arg\min_y g(x,y)$ s.t. $h(x,y) \leq 0$, we show that for each $x$, the reformulation has the same feasible region of $y$.*

*Proof.* We first show that lower-level feasibility implies feasibility of the reformulated problem. Let $y^*, \lambda^* = \min_y \max_{\beta \geq 0} g(x,y) + \beta^\top h(x,y)$ be the primal and the dual solution to the lower level problem with parameter $x$. We can verify that $y^*$ satisfies all the constraints in the reformulation problem. The feasibility condition $h(x,y^*)$ is automatically satisfied. We just need to check:

$$
\begin{aligned}
g^*(x) &:= \min_\theta g(x,\theta) + (\lambda^*)^\top h(x,\theta) \\
&= g(x,y^*) + (\lambda^*)^\top h(x,y^*).
\end{aligned}
\tag{D.1}
$$

Therefore, $x, y^*$ is a feasible point to the reformulation problem.

We now show the other direction, i.e., that feasibility of the reformulaed problem implies that of the lower-level problem. Given $\lambda^*$, let us assume $y$ satisfies $g(x,y) \leq g^*_{\lambda^*}(x)$ and $h(x,y) \leq 0$. On the other hand, assume $y^*, \lambda^* = \min_y \max_{\beta \geq 0} g(x,y) + \beta^\top h(x,y)$ be the primal and the dual solution. We can show that:

$$g(x,y) + (\lambda^*)^\top h(x,y) \leq g^*(x) := \min_\theta g(x,\theta) + (\lambda^*)^\top h(x,\theta). \tag{D.2}$$

By the strong convexity of $g + (\lambda^*)^\top h$, we know that $y$ matches to the unique minimum $y^*$, which implies that $y = y^*$ is also a feasible point to the original bilevel problem. $\square$

## E  Active constraints in differentiable optimization

By computing the derivative of the KKT conditions in Section 2.1, we get:

$$(\nabla^2_{yx} g + (\lambda^*)^\top \nabla^2_{yx} h) + (\nabla^2_{yy} g + (\lambda^*)^\top \nabla^2_{yy} h)\frac{dy^*}{dx} + (\nabla_y h)^\top \frac{d\lambda^*}{dx} = 0 \tag{E.1}$$

$$\text{diag}(\lambda^*)\nabla_x h + \text{diag}(\lambda^*)\nabla_y h \frac{dy^*}{dx} + \text{diag}(h)\frac{d\lambda^*}{dx} = 0. \tag{E.2}$$

Let $\mathcal{I} = \{i \in [d_h] | h(x,y^*)_i = 0, \lambda^*_i > 0\}$ be the set of active constraints with positive dual solution, and $\mathcal{I}_1 = \{i | h(x,y^*)_i \neq 0\}$ be the set of inactive constraints and $\mathcal{I}_2 = \{i | h(x,y^*)_i = 0, \lambda^*_i = 0\}$. We know that $\mathcal{I} = \mathcal{I}_1 \cup \mathcal{I}_2$. For each $i \in \mathcal{I}_1$, due to complementary slackness, we know that $\lambda^*_i = 0$.

For $i \in \mathcal{I}_1$ in (E.1), we have $\lambda^*_i \nabla_x h(x,y^*)_i + \lambda^*_i \nabla_y h(x,y^*)_i \frac{dy^*}{dx} + h(x,y^*)_i \frac{d\lambda^*_i}{dx} = 0$, which implies $h(x,y^*)_i \frac{d\lambda^*_i}{dx} = 0$ because $\lambda^*_i = 0$. This in turn implies $\frac{d\lambda^*_i}{dx} = 0$ because $h(x,y^*)_i < 0$. That means the dual variable $\lambda^*_i = 0$ and has zero gradient $\frac{d\lambda^*_i}{dx} = 0$ for any index $i \in \mathcal{I}_1$. Therefore, we can remove row $i \in \mathcal{I}_1$ in (E.2) and obtain $\lambda^*_i = 0$ and $\frac{d\lambda^*_i}{dx} = 0$.

For $i \in \mathcal{I}_2$, the KKT condition in (E.2) is degenerate. Therefore, $\frac{d\lambda^*_i}{dx}$ can be arbitrary, i.e., non-differentiable. As a subgradient choice, we can set $\frac{d\lambda^*_i}{dx} = 0$ for such $i$. This choice will also eliminate

its impact on the KKT condition in (E.1) because $\frac{d\lambda_i^*}{dx}$ is set to be 0. By this choice of subgradient, we can also remove row $i \in \mathcal{I}_2$ (E.2).

Thus (E.2) can be written as the following set of equations, for $h_{\mathcal{I}} = [h_i]_{i \in \mathcal{I}}$ and $\lambda_{\mathcal{I}}^* = [\lambda_i^*]_{i \in \mathcal{I}}$:

$$\text{diag}(\lambda^*)\nabla_x h_{\mathcal{I}} + \text{diag}(\lambda_{\mathcal{I}}^*)\nabla_y h_{\mathcal{I}}\frac{dy^*}{dx} + \text{diag}(h_{\mathcal{I}})\frac{d\lambda_{\mathcal{I}}^*}{dx} = 0$$

$$\implies \text{diag}(\lambda^*)\nabla_x h_{\mathcal{I}} + \text{diag}(\lambda_{\mathcal{I}}^*)\nabla_y h_{\mathcal{I}}\frac{dy^*}{dx} = 0 \quad \text{(due to } h_{\mathcal{I}}(x, y^*) = 0\text{).} \tag{E.3}$$

In (E.1), due to $\frac{d\lambda_i^*}{dx} = 0$ for all $i \in \bar{\mathcal{I}}$, we can remove $\frac{d\lambda_i^*}{dx} \ \forall i \in \bar{\mathcal{I}}$ in (E.1) by:

$$0 = (\nabla_{yx}^2 g + (\lambda^*)^\top \nabla_{yx}^2 h) + (\nabla_{yy}^2 g + (\lambda^*)^\top \nabla_{yy}^2 h)\frac{dy^*}{dx} + (\nabla_y h)^\top \frac{d\lambda^*}{dx}$$

$$= (\nabla_{yx}^2 g + (\lambda^*)^\top \nabla_{yx}^2 h) + (\nabla_{yy}^2 g + (\lambda^*)^\top \nabla_{yy}^2 h)\frac{dy^*}{dx} + (\nabla_y h_{\mathcal{I}})^\top \frac{d\lambda_{\mathcal{I}}^*}{dx}. \tag{E.4}$$

Combining (E.4) and (E.3), we get:

$$(\nabla_{yx}^2 g + (\lambda^*)^\top \nabla_{yx}^2 h) + (\nabla_{yy}^2 g + (\lambda^*)^\top \nabla_{yy}^2 h)\frac{dy^*}{dx} + (\nabla_y h_{\mathcal{I}})^\top \frac{d\lambda_{\mathcal{I}}^*}{dx} = 0$$

$$\text{diag}(\lambda^*)\nabla_x h_{\mathcal{I}} + \text{diag}(\lambda_{\mathcal{I}}^*)\nabla_y h_{\mathcal{I}}\frac{dy^*}{dx} = 0,$$

which can be written in its matrix form:

$$\begin{bmatrix} \nabla_{yy}^2 g + (\lambda^*)^\top \nabla_{yy}^2 h & \nabla_y h_{\mathcal{I}}^\top \\ \text{diag}(\lambda_{\mathcal{I}}^*)\nabla_y h_{\mathcal{I}} & 0 \end{bmatrix} \begin{bmatrix} \frac{dy^*}{dx} \\ \frac{d\lambda_{\mathcal{I}}^*}{dx} \end{bmatrix} = -\begin{bmatrix} \nabla_{yx}^2 g + (\lambda^*)^\top \nabla_{yx}^2 h \\ \text{diag}(\lambda_{\mathcal{I}}^*)\nabla_x h_{\mathcal{I}} \end{bmatrix} \tag{E.5}$$

This concludes the derivation of the derivative of constrained optimization in (5.2).

# F  Inequality case: bounds on primal solution error and constraint violation

**Lemma 5.1.** *Given any $x$, the corresponding dual solution $\lambda^*(x)$, primal solution $y^*(x)$ of the lower optimization problem in Problem 4.1, and $y_{\lambda^*,\boldsymbol{\alpha}}^*(x)$ as in (5.5), satisfy:*

$$\left\| y_{\lambda^*,\boldsymbol{\alpha}}^*(x) - y^*(x) \right\| \leq O(\alpha_1^{-1}) \ \text{ and } \ \left\| h_{\mathcal{I}}(x, y_{\lambda^*,\boldsymbol{\alpha}}^*(x)) \right\| \leq O(\alpha_1^{-1/2}\alpha_2^{-1/2}). \tag{5.6}$$

*Proof.* We first provide the claimed bound on $\|y_{\alpha_1,\alpha_2}^* - y^*(x)\|$.

**Part 1: Bound on the convergence of $y$.**

Since $y_{\lambda^*,\boldsymbol{\alpha}}^*$ minimizes $\mathcal{L}_{\boldsymbol{\alpha},\lambda^*}(x, y)$, the first-order condition gives us:

$$0 = \nabla_y \mathcal{L}_{\boldsymbol{\alpha},\lambda^*}(x, y_{\lambda^*,\boldsymbol{\alpha}}^*).$$

Similarly, we can compute the gradient of $\mathcal{L}_{\boldsymbol{\alpha},\lambda^*}(x, y)$ at $y^*$:

$$\nabla_y \mathcal{L}_{\boldsymbol{\alpha}}(x, y^*) = \nabla_y f(x, y^*) + \alpha_1(\nabla_y g(x, y^*) + (\lambda^*)^\top \nabla_y h(x, y^*)) + \alpha_2 \nabla_y h_{\mathcal{I}}(x, y^*)^\top h_{\mathcal{I}}(x, y^*)$$
$$= \nabla_y f(x, y^*),$$

where the second step is due to the property of the primal and dual solution: $\nabla_y g(x, y^*) + (\lambda^*)^\top \nabla_y h(x, y^*) = 0$ by the stationarity condition in the KKT conditions, and by definition of the active constraints $h_{\mathcal{I}}$ where the optimal $y^*$ must have $h_{\mathcal{I}}(x, y^*) = 0$.

Since, for a sufficiently large $\alpha_1$, the penalty function is $\alpha_1 \mu_g - L_f \geq \frac{\alpha_1 \mu_g}{2}$ strongly convex in $y$, we have:

$$\frac{\alpha_1 \mu_g}{2} \left\| y^* - y_{\lambda^*,\boldsymbol{\alpha}}^* \right\| \leq \left\| \nabla_y \mathcal{L}_{\boldsymbol{\alpha},\lambda^*}(x, y^*) - \nabla_y \mathcal{L}_{\boldsymbol{\alpha},\lambda^*}(x, y_{\lambda^*,\boldsymbol{\alpha}}^*) \right\| = \left\| \nabla_y f(x, y^*) \right\| \leq L_f.$$

Therefore, upon rearranging the terms, we obtain the claimed bound:

$$\left\| y^* - y_{\boldsymbol{\alpha},\lambda^*}^* \right\| \leq \frac{2L_f}{\alpha_1 \mu_g}.$$

**Part 2: bound on the constraint violation.**

When we plug $y^*$ into (5.4), we get:

$$\mathcal{L}_{\boldsymbol{\alpha},\lambda^*}(x,y^*) = f(x,y^*) + \alpha_1(g(x,y^*) + (\lambda^*)^\top h(x,y^*) - g^*_{\lambda^*}(x)) + \frac{\alpha_2}{2}\left\|h_\mathcal{I}(x,y^*)\right\|^2 = f(x,y^*).$$

Plugging in $y^*_{\boldsymbol{\alpha},\lambda^*}$, we may obtain:

$$\begin{aligned}
\mathcal{L}_{\boldsymbol{\alpha},\lambda^*}(x,y^*_{\lambda^*,\boldsymbol{\alpha}}) &= f(x,y^*_{\lambda^*,\boldsymbol{\alpha}}) + \alpha_1(g(x,y^*_{\lambda^*,\boldsymbol{\alpha}}) + (\lambda^*)^\top h(x,y^*_{\lambda^*,\boldsymbol{\alpha}}) - g^*(x)) + \frac{\alpha_2}{2}\left\|h_\mathcal{I}(x,y^*_{\lambda^*,\boldsymbol{\alpha}})\right\|^2 \\
&= f(x,y^*_{\lambda^*,\boldsymbol{\alpha}}) + \alpha_1(g(x,y^*_{\lambda^*,\boldsymbol{\alpha}}) + (\lambda^*)^\top h(x,y^*_{\lambda^*,\boldsymbol{\alpha}}) - g(x,y^*) - (\lambda^*)^\top h(x,y^*)) \\
&\quad + \frac{\alpha_2}{2}\left\|h_\mathcal{I}(x,y^*_{\lambda^*,\boldsymbol{\alpha}})\right\|^2 \\
&\geq f(x,y^*_{\lambda^*,\boldsymbol{\alpha}}) + \alpha_1\frac{\mu_g}{2}\left\|y^* - y^*_{\lambda^*,\boldsymbol{\alpha}}\right\|^2 + \frac{\alpha_2}{2}\left\|h_\mathcal{I}(x,y^*_{\lambda^*,\boldsymbol{\alpha}})\right\|^2,
\end{aligned}$$

where we used the strong convexity (with respect to $y$) of $g(x,y) + (\lambda^*)^\top h(x,y)$ and the optimality of $y^*$ for $g(x,y) + (\lambda^*)^\top h(x,y)$. By the optimality of $y^*_{\lambda^*,\boldsymbol{\alpha}}$ for $\mathcal{L}_{\boldsymbol{\alpha},\lambda^*}$, we know that

$$f(x,y^*) = \mathcal{L}_{\boldsymbol{\alpha},\lambda^*}(x,y^*) \geq \mathcal{L}_{\boldsymbol{\alpha},\lambda^*}(x,y^*_{\lambda^*,\boldsymbol{\alpha}}) \geq f(x,y^*_{\lambda^*,\boldsymbol{\alpha}}) + \alpha_1\frac{\mu_g}{2}\left\|y^* - y^*_{\lambda^*,\boldsymbol{\alpha}}\right\|^2 + \frac{\alpha_2}{2}\left\|h_\mathcal{I}(x,y^*_{\lambda^*,\boldsymbol{\alpha}})\right\|^2.$$

Therefore, by the Lipschitzness of the function $f$ in terms of $y$, and the bound $\|y^* - y^*_{\lambda^*,\boldsymbol{\alpha}}\| \leq \frac{2L_f}{\alpha_1\mu_g}$, we know that:

$$\begin{aligned}
\frac{\alpha_2}{2}\left\|h_\mathcal{I}(x,y^*_{\lambda^*,\boldsymbol{\alpha}})\right\|^2 &\leq f(x,y^*) - f(x,y^*_{\lambda^*,\boldsymbol{\alpha}}) - \alpha_1\frac{\mu_g}{2}\left\|y^* - y^*_{\lambda^*,\boldsymbol{\alpha}}\right\|^2 \\
&\leq L_f\left\|y^* - y^*_{\lambda^*,\boldsymbol{\alpha}}\right\| - \alpha_1\frac{\mu_g}{2}\left\|y^* - y^*_{\lambda^*,\boldsymbol{\alpha}}\right\|^2 \\
&\leq L_f\left\|y^* - y^*_{\lambda^*,\boldsymbol{\alpha}}\right\| \\
&= O(\alpha_1^{-1}).
\end{aligned}$$

Rearranging terms then gives the claimed bound. $\qquad\square$

The bound on the constraint violation in Lemma 5.1 is an important step in the following theorem.

# G   Proof of Lemma 5.2: gradient approximation for inequality constraints

**Lemma 5.2.** *Consider $F$ as in Problem 4.1, $\mathcal{L}$ as in (5.4), a fixed $x$, and $y^*_{\lambda^*,\boldsymbol{\alpha}}$ as in (5.5). Then under Assumptions 2.2 and 2.5, we have:*

$$\left\|\nabla F(x) - \nabla_x \mathcal{L}_{\lambda^*,\boldsymbol{\alpha}}(x,y^*_{\lambda^*,\boldsymbol{\alpha}})\right\| \leq O(\alpha_1^{-1}) + O(\alpha_1^{-1/2}\alpha_2^{-1/2}) + O(\alpha_1^{1/2}\alpha_2^{-1/2}) + O(\alpha_1^{-3/2}\alpha_2^{1/2}).$$

*Proof.* First, we recall (5.4) here:

$$\mathcal{L}_{\lambda^*,\boldsymbol{\alpha}}(x,y) = f(x,y) + \alpha_1\left(g(x,y) + (\lambda^*)^\top h(x,y) - g^*(x)\right) + \frac{\alpha_2}{2}\left\|h_\mathcal{I}(x,y)\right\|^2.$$

Next, recall from Equation D.1, we can express $g^*(x) = g(x,y^*) + (\lambda^*)^\top h(x,y^*)$, which we use in the first step below:

$$\nabla_x F(x) - \frac{d}{dx}\mathcal{L}_{\lambda^*,\boldsymbol{\alpha}}(x, y^*_{\lambda^*,\alpha})$$

$$= \left(\nabla_x f(x,y^*) + \frac{dy^*}{dx}^\top \nabla_y f(x,y^*)\right) - \left(\nabla_x f(x, y^*_{\lambda^*,\boldsymbol{\alpha}}) + \alpha_1(\nabla_x g(x, y^*_{\lambda^*,\boldsymbol{\alpha}}) + \nabla_x h(x, y^*_{\lambda^*,\boldsymbol{\alpha}})^\top \lambda^*\right.$$

$$\left. - \alpha_1(\nabla_x g(x,y^*) + \nabla_x h(x,y^*)^\top \lambda^*) + \alpha_2 \nabla_x h_{\mathcal{I}}(x, y^*_{\lambda^*,\boldsymbol{\alpha}})^\top h_{\mathcal{I}}(x, y^*_{\lambda^*,\boldsymbol{\alpha}})\right)$$

$$= \nabla_x f(x,y^*) - \nabla_x f(x, y^*_{\lambda^*,\boldsymbol{\alpha}}) \tag{G.1}$$

$$+ \frac{dy^*}{dx}^\top \nabla_y f(x,y^*) - \frac{dy^*}{dx}^\top \nabla_y f(x, y^*_{\lambda^*,\boldsymbol{\alpha}}) \tag{G.2}$$

$$+ \frac{dy^*}{dx}^\top \nabla_y f(x, y^*_{\lambda^*,\boldsymbol{\alpha}}) - \alpha_1 \underbrace{\begin{bmatrix} \nabla^2_{yx}g + (\lambda^*)^\top \nabla^2_{yx}h \\ \mathrm{diag}(\lambda^*_{\mathcal{I}})\nabla_x h_{\mathcal{I}} \end{bmatrix}^\top \begin{bmatrix} y^*_{\lambda^*,\boldsymbol{\alpha}} - y^* \\ 0 \end{bmatrix}}_{\text{added term 1}}$$

$$- \alpha_2 \underbrace{\begin{bmatrix} \nabla^2_{yx}g + (\lambda^*)^\top \nabla^2_{yx}h\lambda^* \\ \mathrm{diag}(\lambda^*_{\mathcal{I}})\nabla_x h_{\mathcal{I}} \end{bmatrix}^\top \begin{bmatrix} 0 \\ \mathrm{diag}(1/\lambda^*_{\mathcal{I}})h_{\mathcal{I}}(x, y^*_{\lambda^*,\boldsymbol{\alpha}}) \end{bmatrix}}_{\text{added term 2}} \tag{G.3}$$

$$+ \alpha_1 \left(\nabla_x g(x,y^*) - \nabla_x g(x, y^*_{\lambda^*,\boldsymbol{\alpha}}) + \nabla_x h(x,y^*)^\top \lambda^* - \nabla_x h(x, y^*_{\lambda^*,\boldsymbol{\alpha}})^\top \lambda^*\right.$$

$$\left. + \underbrace{\begin{bmatrix} \nabla^2_{yx}g + (\lambda^*)^\top \nabla^2_{yx}h \\ \mathrm{diag}(\lambda^*_{\mathcal{I}})\nabla_x h_{\mathcal{I}} \end{bmatrix}^\top \begin{bmatrix} y^*_{\lambda^*,\boldsymbol{\alpha}} - y^* \\ 0 \end{bmatrix}}_{\text{added term 1}}\right) \tag{G.4}$$

$$- \alpha_2 \nabla_x h_{\mathcal{I}}(x, y^*_{\lambda^*,\boldsymbol{\alpha}})^\top h_{\mathcal{I}}(x, y^*_{\lambda^*,\boldsymbol{\alpha}}) + \alpha_2 \underbrace{\begin{bmatrix} \nabla^2_{yx}g + (\lambda^*)^\top \nabla^2_{yx}h\lambda^* \\ \mathrm{diag}(\lambda^*_{\mathcal{I}})\nabla_x h_{\mathcal{I}} \end{bmatrix}^\top \begin{bmatrix} 0 \\ \mathrm{diag}(1/\lambda^*_{\mathcal{I}})h_{\mathcal{I}}(x, y^*_{\lambda^*,\boldsymbol{\alpha}}) \end{bmatrix}}_{\text{added term 2}}.$$

$$\tag{G.5}$$

According to (5.2) and (E.5), we let

$$H = \begin{bmatrix} \nabla^2_{yy}g + (\lambda^*)^\top \nabla^2_{yy}h & \nabla_y h_{\mathcal{I}}^\top \\ \mathrm{diag}((\lambda^*)^*_{\mathcal{I}})\nabla_y h_{\mathcal{I}} & 0 \end{bmatrix},$$

which is invertible by Assumption 2.2(iii) and by the fact that we remove all the inactive constraints. We now bound the terms in (G.1), (G.2), (G.3), (G.4), and (G.5).

**Bounding (G.1) and (G.2):** (G.1) can be easily bounded by the smoothness of $f$ in terms of $x$ and $y$, and the bound on $\|y^* - y^*_{(\lambda^*)^*,\boldsymbol{\alpha}}\| \le O(\alpha_1^{-1})$ from Lemma 5.1. Therefore, we know:

$$\left\|\nabla_x f(x,y^*) - \nabla_x f(x, y^*_{(\lambda^*)^*,\boldsymbol{\alpha}})\right\| \le C_f \left\|y^* - y^*_{(\lambda^*)^*,\boldsymbol{\alpha}}\right\| \le C_f \cdot O(\alpha_1^{-1}).$$

Similarly, given Assumption 2.5 by which $y^*(x)$ is $L_y$-Lipschitz in $x$, we have the bound $\left\|\frac{dy^*}{dx}\right\| \le L_y$. Therefore, (G.2) can be bounded by:

$$\left\|\frac{dy^*}{dx}^\top \nabla_y f(x,y^*) - \frac{dy^*}{dx}^\top \nabla_y f(x, y^*_{(\lambda^*)^*,\boldsymbol{\alpha}})\right\| \le C_f \left\|\frac{dy^*}{dx}\right\| \left\|y^* - y^*_{(\lambda^*)^*,\boldsymbol{\alpha}}\right\| \le C_f L_y \cdot O(\alpha_1^{-1}).$$

**Bounding (G.3):**

Using (5.2) to solve $\left[\begin{smallmatrix} \frac{dy^*}{dx} \\ \frac{d(\lambda^*)^*}{dx} \end{smallmatrix}\right] = -H^{-1} \left[\begin{smallmatrix} \nabla^2_{yx}g + ((\lambda^*)^*)^\top \nabla^2_{yx}h \\ \mathrm{diag}((\lambda^*)^*_\mathcal{I})\nabla_x h_\mathcal{I} \end{smallmatrix}\right]$, we can write:

$$\frac{dy^*}{dx}^\top \nabla_y f(x, y^*_{(\lambda^*)^*, \alpha}) = \begin{bmatrix} \nabla^2_{yx}g + ((\lambda^*)^*)^\top \nabla^2_{yx}h \\ \mathrm{diag}((\lambda^*)^*_\mathcal{I})\nabla_x h_\mathcal{I} \end{bmatrix}^\top (H^{-1})^\top \begin{bmatrix} -\nabla_y f(x, y^*_{(\lambda^*)^*, \alpha}) \\ 0 \end{bmatrix}$$

$$= -\frac{dy^*}{dx}^\top \left( \alpha_1 \begin{bmatrix} \nabla_y g(x, y^*_{(\lambda^*)^*, \alpha}) + \nabla_y h(x, y^*_{(\lambda^*)^*, \alpha})^\top (\lambda^*)^* \\ 0 \end{bmatrix} \right.$$

$$\left. + \alpha_2 \begin{bmatrix} \nabla_y h_\mathcal{I}(x, y^*_{(\lambda^*)^*, \alpha})^\top h_\mathcal{I}(x, y^*_{(\lambda^*)^*, \alpha}) \\ 0 \end{bmatrix} \right), \tag{G.6}$$

where we use the optimality of $y^*_{(\lambda^*)^*, \alpha}$ from (5.5):

$$\nabla_y f(x, y^*_{(\lambda^*)^*, \alpha}) + \alpha_1 \left( \nabla_y g(x, y^*_{(\lambda^*)^*, \alpha}) + \nabla_y h(x, y^*_{(\lambda^*)^*, \alpha})^\top (\lambda^*)^* \right) \tag{G.7}$$

$$+ \alpha_2 \nabla_y h_\mathcal{I}(x, y^*_{(\lambda^*)^*, \alpha})^\top h_\mathcal{I}(x, y^*_{(\lambda^*)^*, \alpha}) = 0.$$

Further, recall that $H$ is non-degenerate by Assumption 2.2, as a result of which, the added term 1 in (G.3) can be modified as follows:

$$\begin{bmatrix} \nabla^2_{yx}g + ((\lambda^*)^*)^\top \nabla^2_{yx}h \\ \mathrm{diag}((\lambda^*)^*_\mathcal{I})\nabla_x h_\mathcal{I} \end{bmatrix}^\top \begin{bmatrix} \alpha_1(y^*_{(\lambda^*)^*, \alpha} - y^*) \\ 0 \end{bmatrix}$$

$$= \begin{bmatrix} \nabla^2_{yx}g + ((\lambda^*)^*)^\top \nabla^2_{yx}h \\ \mathrm{diag}((\lambda^*)^*_\mathcal{I})\nabla_x h_\mathcal{I} \end{bmatrix}^\top (H^{-1})^\top H^\top \begin{bmatrix} \alpha_1(y^*_{(\lambda^*)^*, \alpha} - y^*) \\ 0 \end{bmatrix}$$

$$= \alpha_1 \begin{bmatrix} \nabla^2_{yx}g + ((\lambda^*)^*)^\top \nabla^2_{yx}h \\ \mathrm{diag}((\lambda^*)^*_\mathcal{I})\nabla_x h_\mathcal{I} \end{bmatrix}^\top (H^{-1})^\top \begin{bmatrix} (\nabla^2_{yy}g + ((\lambda^*)^*)^\top \nabla^2_{yy}h)^\top (y^*_{(\lambda^*)^*, \alpha} - y^*) \\ \nabla_y h_\mathcal{I}(x, y^*)(y^*_{(\lambda^*)^*, \alpha} - y^*) \end{bmatrix}. \tag{G.8}$$

The added term 2 in (G.3) can be expanded to:

$$\alpha_2 \begin{bmatrix} \nabla^2_{yx}g + ((\lambda^*)^*)^\top \nabla^2_{yx}h(\lambda^*)^* \\ \mathrm{diag}((\lambda^*)^*_\mathcal{I})\nabla_x h_\mathcal{I} \end{bmatrix}^\top \begin{bmatrix} 0 \\ \mathrm{diag}(1/(\lambda^*)^*_\mathcal{I})h_\mathcal{I}(x, y^*_{(\lambda^*)^*, \alpha}) \end{bmatrix}$$

$$= \alpha_2 \begin{bmatrix} \nabla^2_{yx}g + ((\lambda^*)^*)^\top \nabla^2_{yx}h(\lambda^*)^* \\ \mathrm{diag}((\lambda^*)^*_\mathcal{I})\nabla_x h_\mathcal{I} \end{bmatrix}^\top (H^{-1})^\top H^\top \begin{bmatrix} 0 \\ \mathrm{diag}(1/(\lambda^*)^*_\mathcal{I})h_\mathcal{I}(x, y^*_{(\lambda^*)^*, \alpha}) \end{bmatrix}$$

$$= \alpha_2 \begin{bmatrix} \nabla^2_{yx}g + ((\lambda^*)^*)^\top \nabla^2_{yx}h(\lambda^*)^* \\ \mathrm{diag}((\lambda^*)^*_\mathcal{I})\nabla_x h_\mathcal{I} \end{bmatrix}^\top (H^{-1})^\top \begin{bmatrix} \nabla_y h_\mathcal{I}(x, y^*)^\top h_\mathcal{I}(x, y^*_{(\lambda^*)^*, \alpha}) \\ 0 \end{bmatrix} \tag{G.9}$$

Therefore, we can compute the difference between (G.6), (G.8), and (G.9) to bound (G.3), and use the fact that $\nabla_y g(x, y^*) + (\lambda^*)^\top \nabla_y h(x, y^*) = 0$:

$$\frac{dy^*}{dx}^\top \nabla_y f(x, y^*_{\lambda^*, \alpha}) - \text{added term 1} - \text{added term 2}$$

$$= \begin{bmatrix} \nabla^2_{yx}g + (\lambda^*)^\top \nabla^2_{yx}h \\ \mathrm{diag}(\lambda^*_\mathcal{I})\nabla_x h_\mathcal{I} \end{bmatrix}^\top (H^{-1})^\top \left( \alpha_1 \begin{bmatrix} \nabla_y g(x, y^*_{\lambda^*, \alpha}) - \nabla_y g(x, y^*) - \nabla^2_{yy}g(x, y^*)(y^*_{\lambda^*, \alpha} - y^*) \\ 0 \end{bmatrix} \right.$$

$$\tag{G.10}$$

$$+ \alpha_1 \begin{bmatrix} \nabla_y h(x, y^*_{\lambda^*, \alpha})^\top \lambda^* - \nabla_y h(x, y^*)^\top \lambda^* - \nabla^2_{yy}h(x, y^*)^\top \lambda^*(y^*_{\lambda^*, \alpha} - y^*) \\ 0 \end{bmatrix} \tag{G.11}$$

$$- \alpha_1 \begin{bmatrix} 0 \\ \nabla_y h_\mathcal{I}(x, y^*)(y^*_{\lambda^*, \alpha} - y^*) \end{bmatrix} \tag{G.12}$$

$$+ \alpha_2 \left( \begin{bmatrix} \nabla_y h_\mathcal{I}(x, y^*_{\lambda^*, \alpha})^\top h_\mathcal{I}(x, y^*_{\lambda^*, \alpha}) \\ 0 \end{bmatrix} - \begin{bmatrix} \nabla_y h_\mathcal{I}(x, y^*)^\top h_\mathcal{I}(x, y^*_{\lambda^*, \alpha}) \\ 0 \end{bmatrix} \right) \tag{G.13}$$

The terms in (G.10) and (G.11) can both be bounded by $\alpha_1 C_g L_y \|y^*_{\lambda^*, \alpha} - y^*\|^2$ and $\alpha_1 R C_h L_y \|y^*_{\lambda^*, \alpha} - y^*\|^2$ by the smoothness of $g$ and $h^\top \lambda^*$. Further, plugging in $\|y^* - y^*_{\lambda^*, \alpha}\| \le O(\alpha_1^{-1})$ from Lemma 5.1 bounds both these terms by $O(\alpha_1^{-1})$.

To bound the term in (G.12), we use:
$$\left\| h_{\mathcal{I}}(x, y^*_{\lambda^*, \boldsymbol{\alpha}}) - h_{\mathcal{I}}(x, y^*) - \nabla_y h_{\mathcal{I}}(x, y^*)(y^*_{\lambda^*, \boldsymbol{\alpha}} - y^*) \right\| \le C_h \left\| y^*_{\lambda^*, \boldsymbol{\alpha}} - y^* \right\|^2 .$$
Therefore, we have:
$$\left\| \nabla_y h_{\mathcal{I}}(x, y^*)(y^*_{\lambda^*, \boldsymbol{\alpha}} - y^*) \right\| \le \left\| h_{\mathcal{I}}(x, y^*_{\lambda^*, \boldsymbol{\alpha}}) \right\| + \left\| h_{\mathcal{I}}(x, y^*) \right\| + C_h O(\left\| y^*_{\lambda^*, \boldsymbol{\alpha}} - y^* \right\|^2)$$
$$\le O(\alpha_1^{-1/2} \alpha_2^{-1/2}) + 0 + O(\alpha_1^{-2})$$
$$= O(\alpha_1^{-1/2} \alpha_2^{-1/2} + \alpha_1^{-2}),$$
which upon scaling by $\alpha_1$ gives us the following bound on the term in (G.12):
$$\alpha_1 \left\| \nabla_y h_{\mathcal{I}}(x, y^*)(y^*_{\lambda^*, \boldsymbol{\alpha}} - y^*) \right\| \le O(\alpha_1^{1/2} \alpha_2^{-1/2} + \alpha_1^{-1}) .$$

The term in (G.13) can be bounded by:
$$\alpha_2 \left\| \nabla_x h_{\mathcal{I}}(x, y^*_{\lambda^*, \boldsymbol{\alpha}})^\top h_{\mathcal{I}}(x, y^*_{\lambda^*, \boldsymbol{\alpha}}) - \nabla_x h_{\mathcal{I}}(x, y^*)^\top h_{\mathcal{I}}(x, y^*_{\lambda^*, \boldsymbol{\alpha}}) \right\|$$
$$= \alpha_2 \left\| \nabla_x h_{\mathcal{I}}(x, y^*_{\lambda^*, \boldsymbol{\alpha}}) - \nabla_x h_{\mathcal{I}}(x, y^*) \right\| O(\left\| h_{\mathcal{I}}(x, y^*_{\boldsymbol{\alpha}, \lambda^*}) \right\|)$$
$$= \alpha_2 \cdot O(\alpha_1^{-1}) O(\alpha_1^{-1/2} \alpha_2^{-1/2})$$
$$= O(\alpha_1^{-3/2} \alpha_2^{1/2}) \tag{G.14}$$

**Bounding (G.4):** This can be easily bounded by the smoothness of $g$ and $h$, and the bound on the dual solution $\|\lambda^*\| \le R$. Thus (G.4) can be bounded by $R \cdot O(\alpha_1^{-1}) = O(\alpha_1^{-1})$.

**Bounding (G.5):** By the same argument in (G.14), we get:
$$\alpha_2 \left\| \nabla_y h_{\mathcal{I}}(x, y^*_{\lambda^*, \boldsymbol{\alpha}})^\top h_{\mathcal{I}}(x, y^*_{\lambda^*, \boldsymbol{\alpha}}) - \nabla_y h_{\mathcal{I}}(x, y^*)^\top h_{\mathcal{I}}(x, y^*_{\lambda^*, \boldsymbol{\alpha}}) \right\|$$
$$\le \alpha_2 \left\| \nabla_y h_{\mathcal{I}}(x, y^*_{\lambda^*, \boldsymbol{\alpha}}) - \nabla_y h_{\mathcal{I}}(x, y^*) \right\| \left\| h_{\mathcal{I}}(x, y^*_{\lambda^*, \boldsymbol{\alpha}}) \right\|$$
$$= \alpha_2 \cdot O(\alpha_1^{-1}) O(\alpha_1^{-1/2} \alpha_2^{-1/2})$$
$$= O(\alpha_1^{-3/2} \alpha_2^{1/2}) .$$

Combining all upper bounds gives the claimed bound. $\qquad\square$

## H  Proof of the main result (Theorem 5.3): convergence and computation cost

**Theorem 5.3.** *Given any accuracy parameter $\alpha > 0$, Algorithm 4 outputs $\widetilde{\nabla}_x F(x)$ such that $\|\widetilde{\nabla} F(x) - \nabla F(x)\| \le \alpha$ within $\widetilde{O}(\alpha^{-1})$ gradient oracle evaluations.*

*Proof.* First, given the bound in Lemma 5.2, we choose $\alpha_1 = \alpha^{-2}$ and $\alpha_2 = \alpha^{-4}$ to ensure the inexactness of the gradient oracle is bounded by $\alpha$. In the later analysis, we will still use $\alpha_1$ and $\alpha_2$ in the penalty function for clarity.

Now we estimate the computation cost of the inexact gradient oracle:

**Lower-level problem.** Given the oracle access to the optimal dual solution $\lambda^*(x)$, we can recover the primal solution $y^*(x)$ efficiently (e.g, by [45]). Therefore, we can use the primal and dual solutions to construct the penalty function $\mathcal{L}_{\lambda^*, \alpha}(x, y)$ in (5.4).

**Penalty function minimization problem.** The second main optimization problem is the penalty minimization problem in Line 4 of Algorithm 4. Recall from (5.4) that
$$\mathcal{L}_{\lambda, \boldsymbol{\alpha}}(x, y) = f(x, y) + \alpha_1 \left( g(x, y) + \lambda^\top h(x, y) - g^*(x) \right) + \frac{\alpha_2}{2} \left\| h_{\mathcal{I}}(x, y) \right\|^2 , \tag{H.1}$$
where we use the approximate dual solution $\lambda$ as opposed to the optimal dual solution $\lambda^*$.

Given (H.1), we solve the penalty minimization problem:
$$y'_{\lambda, \alpha}(x) := \arg\min_y \mathcal{L}_{\lambda, \boldsymbol{\alpha}}(x, y).$$

The penalty minimization is a unconstrained strongly convex optimization problem, which is known to have linear convergence rate. We further analyze its convexity and smoothness below to precisely estimate the computation cost:

- The strong convexity of $\mathcal{L}_{\lambda,\boldsymbol{\alpha}}(x,y)$ is lower bounded by $\frac{\alpha_1\mu_g}{2} = O(\alpha_1)$.

- The smoothness of $\mathcal{L}_{\lambda,\boldsymbol{\alpha}}(x,y)$ is dominated by the smoothness of $\alpha_2 \|h_{\mathcal{I}}(x,y)\|^2$ since $\alpha_2 \gg \alpha_1$. By Lemma 5.2, we know that the optimal solution must lie in an open ball $B(y^*, O(1/\alpha_1))$ with center $y^*$ (inner optimization primal solution) and a radius of the order of $O(\frac{1}{\alpha_1})$. This implies that we just need to search over a bounded feasible set of $y$, which we can bound $\|\nabla_y h(x,y)\| \le L_h$ and $h(x,y) \le H$ within the bounded region $y \in B(y^*, O(1/\alpha_1))$. We can show that $h^2$ is smooth (gradient Lipschitz) within the bounded region by the following:

$$\left\|\nabla_{yy}^2 h^2\right\| = \left\|h\nabla_{yy}^2 h + \nabla_y h^\top \nabla_y h\right\| \le \left\|h\nabla_{yy}^2 h\right\| + \left\|\nabla_y h^\top \nabla_y h\right\| \le HC_h + L_h^2$$

which also implies $h_{\mathcal{I}}^2$ is also smooth (gradient Lipschitz). Therefore, $\alpha h_{\mathcal{I}}^2$ is $(HC_h + L_h^2)\alpha_2 = O(\alpha_2)$ smooth.

Choosing $\alpha_1 = \frac{1}{\alpha^2}$ and $\alpha_2 = \frac{1}{\alpha^4}$, the condition number of $\mathcal{L}_{\boldsymbol{\alpha},\lambda}(x,y)$ becomes $\kappa = O(\alpha_2/\alpha_1) = O(\frac{1}{\alpha^2})$. Therefore, by the linear convergence of gradient descent in strongly convex smooth optimization, the number of iterations needed to get to $\alpha$ accuracy is $O(\sqrt{\alpha^{-2}} \times \log(\frac{1}{\alpha})) = O(\frac{1}{\alpha}\log(\frac{1}{\alpha}))$. Therefore, we can get a near optimal solution $y'_{\lambda,\alpha}$ with inexactness $\alpha$ in $O(\frac{1}{\alpha})$ oracle calls.

**Computation cost and results.** Overall, for the inner optimization, we can invoke the efficient optimal dual solution oracle to get the optimal dual solution $\lambda^*(x)$ and recover the optimal primal solution $y^*(x)$ from there. For the penalty minimization, we need $O(\frac{1}{\alpha})$ oracle calls to solve an unconstrained strongly convex smooth optimization problem to get to $\alpha$ accuracy. In conclusion, combining everything in Appendix H, we run $O(\frac{1}{\alpha})$ oracle calls to obtain an $\alpha$ accurate gradient oracle to approximate the hyperobjective gradient $\nabla_x F(x)$. This concludes the proof of Theorem 5.3. □

**Remark H.1.** *The following analysis quantifies how the error in the optimal dual solution propagates to the inexact gradient estimate. This is not needed if such a dual solution oracle exists. But in practice, the oracle may come with some error, for which we bound the error.*

**Bounding the error propagation in error in dual solution and the penalty minimization.** First, if we do not get an exact optimal dual solution, the error in the dual solution $\lambda$ with $\|\lambda - \lambda^*\| \le \alpha$ will slightly impact the analysis in Lemma 5.2. Specifically, in Appendix G, the approximate $\lambda$ will impact the inexact gradient $\nabla_x \mathcal{L}_{\lambda,\alpha}(x, y'_{\lambda,\alpha})$ computation and the analysis in (G.4) and (G.7). In (G.4), to change $\lambda$ to $\lambda^*$, we get an additional error:

$$\alpha_1\left(\nabla_x h(x,y')^\top(\lambda - \lambda^*) - \nabla_x h(x, y'_{\lambda,\alpha})^\top(\lambda - \lambda^*)\right) \tag{H.2}$$

$$=\alpha_1(\nabla_x h(x,y') - \nabla_x h(x, y'_{\lambda,\alpha}))^\top(\lambda - \lambda^*)$$

$$\le \alpha_1 C_h \left\|y' - y'_{\lambda,\alpha}\right\|(\lambda - \lambda^*)$$

$$\le O(\alpha_1\alpha_1^{-1}\alpha) = O(\alpha),$$

where the last inequality is due to $\left\|y' - y'_{\lambda,\alpha}\right\| \le O(\alpha_1^{-1})$ that is based on a similar analysis in Lemma 5.1 with a near-optimal $y'_{\lambda,\alpha}$ under $\alpha^2 = \alpha_1$ accuracy.

Therefore, the error incurred by inexact $\lambda$ in (G.4) is at most $O(\alpha)$, which is of the same rate as the current gradient inexactness $O(\alpha)$.

In (G.7), the optimality holds approximately for the approximate $\lambda$. Therefore, by the near optimality of $y'_{\lambda,\boldsymbol{\alpha}}$ (strongly convex optimization), we know that the following gradient is also $\alpha$-close to 0, i.e.,

$$\|\nabla_y f(x, y'_{\lambda,\boldsymbol{\alpha}}) + \alpha_1\left(\nabla_y g(x, y'_{\lambda,\boldsymbol{\alpha}}) + \nabla_y h(x, y'_{\lambda,\boldsymbol{\alpha}})^\top\lambda\right) \tag{H.3}$$

$$+ \alpha_2\nabla_y h_{\mathcal{I}}(x, y'_{\lambda,\boldsymbol{\alpha}})^\top h_{\mathcal{I}}(x, y'_{\lambda,\boldsymbol{\alpha}})\| \le \alpha,$$

whose inexactness matches the inexactness of the gradient oracle $\alpha$, and thus we do not incur additional order of inexactness here.

Moreover, there is an additional error because we need $\lambda^*$ as opposed to a near-optimal $\lambda$ to make the analysis in Appendix G work. The error between using $\lambda$ and $\lambda^*$ in (H.3) can be bounded by:

$$\left\| \nabla_y h(x, y'_{\lambda,\alpha})^\top (\lambda - \lambda^*) \right\| \leq L_h \alpha, \tag{H.4}$$

where we use the local Lipschitzness of the function $h$ in an open ball near $y^*$. Therefore, the additional error is also $O(\alpha)$, which matches the inexactness of the inexact gradient oracle.

Therefore, we conclude that in order to bound the inexactness of the gradient oracle, we just need an efficient inexact dual solution with $\alpha$ accuracy.

# I  Practical oracle to optimal (approximate) dual solution

Here we discuss how practical the assumption on the oracle access to the optimal dual solution is.

For linear inequality constraint $h(x, y) = Ax - By - b$, the LL problem is a constrained strongly convex smooth optimization problem. To show that we can compute an approximate solution to the optimal dual solution for linear inequality constraints, we apply the result from [93]:

**Corollary I.1** (Application of Corollary 3.1 in [93])**.** *When $h(x, y) = Ax - By + b$ is linear in $y$, the primal and dual solutions can be written as:*

$$y^*, \lambda^* = \arg\min_y \max_\lambda g(x, y) + (\lambda^*)^\top h(x, y) = g(x, y) - (\lambda^*)^\top By + R(x)$$

$$\Longleftrightarrow y^*, \lambda^* = \arg\min_y \max_\lambda g(x, y) - (\lambda^*)^\top By \tag{I.1}$$

*where $g$ is strongly convex in $y$ and $B$ is of full rank by Assumption 2.2. According to Corollary 3.1 from [93], the primal-dual gradient method guarantees a linear convergence. More precisely, in $t = O(\log \frac{1}{\alpha})$ iterations, we get:*

$$\left\| y^t - y^* \right\| \leq \alpha \text{ and } \left\| \lambda^t - \lambda^* \right\| \leq \alpha. \tag{I.2}$$

Given Corollary I.1, we can efficiently approximate the primal and dual solutions up to high accuracy with $O(\log \frac{1}{\alpha})$ oracle calls when the inequality constraints are linear. This gives us an efficient approximate oracle access to the dual solution.

**Remark I.2.** *Under the assumption of an optimal dual solution oracle, all the analyses mentioned in Section 5 hold for the general convex inequality constraints. However, the main technical challenge is that the dual solution oracle for general convex inequality cannot be guaranteed in practice. In fact, to the best of our knowledge, there is no iterate convergence in the dual solution $\lambda$ for general convex inequality constraints. Most of the literature in strongly-convex-concave saddle point convergence only guarantees dual solution convergence in terms of its duality gap or some other merit functions. We are not aware of any successful bound on the dual solution iterate convergence, which is an important research question to answer by itself. This is the main technical bottleneck for general convex inequality constraints as well.*

**Remark I.3.** *On the other hand, we need the dual solution iterate convergence with rate $O(1/\alpha)$ to ensure the error to be bounded. But this is not a necessary condition. To ensure a bound on the error propagation, we just need to bound some forms of merit functions ((H.2) and (H.4)) of the dual solutions, which we believe that this is much more tractable than the actual iterate dual solution convergence. We leave this as a future direction and this will generalize the analysis from linear inequality constraints to general convex inequality constraints.*

# J  The role of $\lambda^*(x)$ in the derivative of Equation (5.4)

Notice that Equation (5.4), we treat the dual solution $\lambda^*(x)$ as a constant to define the penalty function derivative. Yet, the dual solution $\lambda^*(x)$ is in fact also a function of $x$. Therefore, in theory, we should also compute its derivative with respect to $x$.

However, notice that the following:

$$\nabla_x (\lambda^*(x))^\top h(x, y) = \nabla_x h(x, y)^\top \lambda^* + \frac{d\lambda(x)}{dx}^\top h(x, y) \tag{J.1}$$

The later term in Equation (J.1) can be divided into two cases:

- For active constraint $i \in \mathcal{I}$ with $h(x, y^*) = 0$, we know that $y^*_{\lambda, \alpha}$ is close to $y^*$ by Lemma 5.1. Therefore, the derivative $\left\| \frac{d\lambda(x)}{dx}^\top h(x, y^*_{\lambda, \alpha}) \right\| \leq L_h L_\lambda \alpha_1 = O(\alpha_1) = O(\alpha^2)$ by the local smoothness of $h$ near $y^*$ and the Lipschitzness assumption of $\lambda^*$ in Assumption 2.5.

- For inactive constraint $i \in \bar{\mathcal{I}}$ and $\lambda_i^* > 0$, we can solve the KKT conditions and get $\frac{d\lambda(x)}{dx} = 0$. Therefore, the second term becomes 0.

- For inactive constraint $i \in \bar{\mathcal{I}}$ and $\lambda_i^* = 0$, the KKT system degenerates and we need to use subgradient. By solving the KKT system, we find that $\frac{d\lambda(x)}{dx} = 0$ is a valid subgradient. Therefore, by choosing this subgradient, the second term also vanishes.

Therefore, we do not need to compute the derivative of $\lambda^*$ as the terms involved its derivative is negligible compared to other major terms.

## K   Experimental setup

All experiments were run on a computing cluster with Dual Intel Xeon Gold 6226 CPUs @ 2.7 GHz and DDR4-2933 MHz DRAM. No GPU was used, and we used 1 core with 8GB RAM per instance of the experiment. The cutoff time for running the algorithms is set to be 6 hours. All experiments were run and averaged over 10 different random seeds. All parameters in the constrained bilevel optimization in Section 6, including the objective parameters and the constrain parameters, are randomly generated from a normal distribution with 0-mean and standard deviation 1.

For our fully first-order algorithm, we implement Algorithm 3, where the inexact gradient oracle subroutine is provided by implementing Algorithm 4. All algorithms are implemented in PyTorch [94] to compute gradients, and using Cvxpy [95] to solve the LL problem and the penalty minimization problem. We implement our fully first-order method based on the solutions returned by Cvxpy with certain accuracy requirement, and use PyTorch to compute the inexact gradient discussed in Section 5. We implement the non-fully first-order method using the CvxpyLayer [18], which is a Cvxpy compatible library that can differentiate through the LL convex optimization problem.

## L   Additional Experimental Results

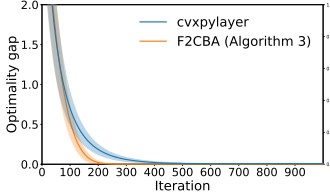
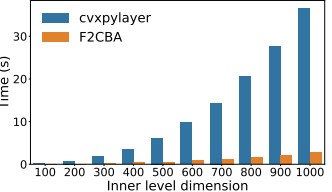

**(a)** Convergence and gradient error of Fully First-order Constrained Bilevel Algorithms (F2CBA). We set $d_x = 20$

**(b)** Convergence analysis with varying gradient inexactness $\alpha$. We set $d_x = 100$ to measure the tradeoff of accuracy and convergence.

**(c)** Computation cost per gradient step of varying problem size $d_y$. We set $d_x = 100$ to measure large-scale computation cost.

**Figure 2:** We run Algorithm 3 using Algorithm 4 on the bilevel optimization in the toy example in Problem L.1 with varying upper-level variable dimensions $d_x$, a fixed lower-level variable dimension $d_y = 200$, and the number of constraints $n_{\text{const}} = d_y/5 = 40$, and accuracy $\alpha = 0.1$. Figure 2a, Figure 2b, Figure 2c vary # of iterations, gradient exactness $\alpha$, and $d_y$, respectively, to compare the performance under different settings.

We generate instances of the following constrained bilevel optimization problem:

$$\text{minimize}_x \ c^\top y^* + 0.01 \|x\|^2 + 0.01 \|y^*\|^2 \quad \text{subject to} \quad y^* = \arg\min_{y:h(x,y)\leq 0} \frac{1}{2} y^\top Q y + x^\top P y, \quad \text{(L.1)}$$

where $h_i(x, y) = x^\top A_i y - b_i^\top x \ \forall i \in [d_h]$ is a $d_h$-dim bilinear constraint, where the constraint bilinear matrix $A_i \in \mathbb{R}^{d_x \times d_y}$, $b_i \in \mathbb{R}^{d_x}$ for all $i \in [d_h]$ are randomly generated from normal

distributions. The bilinear (nonlinear) constraint of the lower-level problem is the major difference compared to the experiment in Section 6. We are interested in whether our algorithms work beyond the linear constraints where our theory guarantees.

The rest of the parameters are the same as in Section 6. The PSD matrix $Q \in \mathbb{R}^{d_y \times d_y}$, $c \in \mathbb{R}^{d_y}$, $P \in \mathbb{R}^{d_x \times d_y}$. We compare our Algorithm 3 with a non-fully first-order method implemented using `cvxpyLayer` [18]. Both algorithms use Adam [90] to control the learning rate in gradient descent. All the experiments are averaged over ten random seeds.

