# OpenReview forum: "First-Order Methods for Linearly Constrained Bilevel Optimization"
_NeurIPS.cc/2024/Conference — NeurIPS 2024 poster_

### Official Review · Reviewer_c7kZ · 2024-06-29

**Soundness:** 3
**Presentation:** 2
**Contribution:** 4
**Rating:** 6
**Confidence:** 4

**Summary:**

This paper studies fully first-order methods for bilevel optimization with strongly convex lower-level objectives and linear lower-level constraints, including both inequality constraints and equality constraints. For linear inequality constraints, the paper used a penalty methods to construct hypergradient estimators and demonstrate the approximation error. Then combined with a gradient method for nonsmooth nonconvex optimization to obtain a Goldstein stationarity point. While for linear inequality constraints, the paper proposed to use zeroth-order approximation of the hyper-gradient and demonstrate complexity.

**Strengths:**

1. The paper considered both linear inequality and equality constraints.
2. The hypergradient construction in the linear inequality constraint setting is interesting and the paper provides approximation guarantees. It would be useful for follow-up papers.
3. Obviously, the complexity bounds are not yet optimal. But the paper has contributed several interesting directions that worth further exploration.
3. Algorithm 5 uses zeroth-order oracle is very smart as in many bilevel problems, the dimension of the upper-level is small such as hyperparameter optimization.
3. The finite-difference gradient proxy for the linear equality constrained case is also interesting.

**Weaknesses:**

1. How to solve linear inequality constrained case with simple gradient methods to achieve optimal complexity bounds still remain unknown. The current algorithm is a bit complicated and hard to tune in practice.
2. It is unclear if the complexity bounds are optimal for both linearly inequality and equality constrained settings. Following the conventional notation for stationarity such that $||\nabla F(x)||\leq \epsilon^2$, the complexity bound for Alg 1 is $O(\epsilon^{-8})$, for Alg 2 is $O(\epsilon^{-10})$ if further taking into account the complexity to get $tilde \nabla F$, and for Alg 3 is $O(\epsilon^{-4})$. Note that the paper consider deterministic bilevel optimization, it implies that there are a lot of room for improvements. The paper should mention comparison to unconstrained lower-level bilevel optimization so that readers know the results can be improved. Despite the complexity bounds, I still believe that this paper makes good contribution to the community of bilevel optimization.
3. The methods proposed in the paper are not aligning with each other well. Despite the constrained lower-level bilevel optimization setting, the paper lacks a coherent narrative in terms of algorithmic design.

**Questions:**

1. In the Alg 3 perturbed GD, what is the intuition to use perturbed GD instead of just inexact GD $\tilde \nabla F(x_t)$. In other words, why one needs to perturb using samples from sphere distribution?
2. In Sec 3, although the problem is nonsmooth and nonconvex, given Thm 3.3, I would assume that the $\nabla F(x)$ exists but it is not Lipschitz continuous?
3. Please be specific about Lipschitz, i.e., does it mean Lipschitz continuous or Lipschitz smooth.
4. What is the reason to consider zeroth-order methods in the linear equality constrained setting? Is it possible to derive fully first-order gradient estimator for the linear equality constrained setting? If possible, why not do it?
5. In Sec 3.1, assuming knowing $\lambda^*$ means that one also knows $y^*$ and $g^*$.
6. The definition of complexity is not very clear. In line 229, it says solving a single smooth and strongly convex optimization problems amounts to $\tilde O(1)$ oracle calls. But line 226 implies that one of such oracle call requires $\epsilon^{-1}$ gradient evaluations. Then in line 322, the complexity seems to become $O(log(1/\epsilon))$. Please specify how expensive to obtain $y^*$ and $\lambda^*$ in an $\epsilon$ error and clean oracle calls and complexity bounds in a consistent manner.
7. What is the motivation of (5.3) the perturbed lower-level problem?
8. Alg 1 requires computing $y^*_{\lambda^*,\alpha}$, this requires either $f$ to be convex in $y$ or requires $\alpha$ to be large enough to ensure $\mathcal{L}$ is convex in $y$. Please specify explicitly the requirements. Is $\mathcal{L}$ strongly convex in $y$?

I am happy to raise the score if these issues are well-addressed.

**Limitations:**

The paper includes several settings. Yet why adapting a specific strategy to solve each setting is not sufficiently motivated. Such as why one needs to use Alg 2 and Alg 3 instead of directly using the hypergradient estimator with gradient descent. Why one needs to use zeroth-order oracle for the linearly equality constrained setting instead of first-order oracle?

---

> ### Author Rebuttal · Authors · 2024-08-06
>
> We are extremely grateful to the reviewer for their thoughtful questions and are encouraged that they found numerous aspects of our contribution interesting and useful for follow-up papers.
>
> ---
> # Response to Weaknesses
> ---
> 1. ## Complexity of Algorithms.
>
>     We agree with the reviewer that some of our results are probably suboptimal in terms of convergence rates. Nonetheless, we wish to point out that our rate for the linear equality case is in fact optimal. Moreover, despite this suboptimality, ours is the first result with dimension-free rates for these settings for our choice of stationarity and oracle. We will add an explicit comparison with the unconstrained setting.
>
> ---
> 2. ## Coherence of narrative.
>
>     We thank the reviewer for raising this valid point. Our algorithm for the linear *equality* case crucially relies on our novel observation that the hyperobjective $F$ is smooth; in contrast, this does not necessarily hold in the linear *inequality* case. Therefore, the two cases require different approaches, and unifying them would be an interesting future direction. We will clarify this better.
>
> ---
> # Response to Questions
> ---
> 1. ## Need for perturbation.
>
>       Because we do not assume a bound on the smoothness of the hyperobjective (a strong assumption by several prior works), it is known [1] to be impossible to optimize $F$ directly in general – in particular by inexact GD, as suggested.
>
>    Since the hyperobjective is Lipschitz (cf. Lemma 4.1), it is differentiable almost everywhere by Rademacher's theorem; we can therefore implicitly optimize its uniform smoothing via random perturbations. This method of using perturbations to avoid non-differentiable points is widely used in the literature on nonsmooth optimization. We will clarify this.
>
> ---
> 2. ## Meaning of ${\nabla} F$.
>
>    We thank the reviewer for pointing out this subtlety, which we will better explain: As correctly noted, $F$ can be nonsmooth nonconvex; in this setting, algorithms generally require a subgradient, i.e., an element of the Clarke subdifferential (see Definition 2.1), which is what we mean by $\nabla F$ (which, as the reviewer notes, is not Lipschitz continuous). Furthermore, the $\tilde{\nabla} F$ we construct is such that its distance to some subgradient of $\nabla F$ is at most $\alpha$.
>
> ---
> 3. ## Lipschitz notation.
>
>     We acknowledge the confusion caused by our notation and will switch all notations to either $L$-Lipschitz or $S$-smooth.
>
> ---
> 4. ## Zeroth-order approach for linear equality setting.
>
>     We apologize for the confusion: Please note that for the linear equality case, our algorithm accesses $\nabla f$ and $\nabla g$, which makes it first-order. So, even though it *looks* like a zeroth-order approach (due to the finite differences), it is actually a first-order method. We will clarify this.
>
> ---
> 5. ## Assumption on $\lambda^\*$.
>
>     Access to $\lambda^\*$ indeed implies access to $y^\*$. That said, the core difficulty in bilevel programs is access to $\frac{d}{dx}y^\*$. Our technical novelty is precisely to approximate this first-order oracle. Moreover, in practice, we can easily approximate $\lambda^\*$ to high accuracy due to the linear convergence of strongly convex problems with linear constraints --- our experiments demonstrate our algorithm's robustness to the use of such a proxy for $\lambda^\*$.
>
> ---
> 6. ## Clarifying complexity.
>
>     The notation $\tilde{O}(1)$ in Line 229 refers to $\log(1/\epsilon)$, matching the cost in Line 322 — We apologize for our oversight in not stating this and will fix it. Next, the complexities in Lines 226 and 229 are for different operations: Line 226 is the cost of estimating $\nabla F(x)$ and Line 229 that of estimating $F(x)$. For the linear inequality case, the ACGD method in [2] implies linear rate to obtain $y^\*$ and $\lambda^\*$. We will explicitly state these.
>
> ---
> 7. ## Motivating the perturbed lower-level problem.
>
>    For intuition on the perturbation, we demonstrate our method in the unconstrained setting. Let $y_{\delta}^\*:=\text{argmin} (g(x,y)+\delta f(x,y)).$  Optimality of $y_{\delta}^\*$ and differentiating w.r.t. $\delta$ yields $$\frac{dy_{\delta}^\*}{d\delta}=-\left(\delta\cdot\nabla_{yy}^{2}f(x,y_{\delta}^{\ast})+\nabla_{yy}^{2}g(x,y_{\delta}^{\ast})\right)^{-1}\nabla_{y}f(x,y_{\delta}^{\ast}), \implies \frac{dy_{\delta}^{\ast}}{d\delta}\vert_{\delta=0} = -\left(\nabla_{yy}^{2}g(x,y^\*)\right)^{-1}\nabla_{y}f(x,y^\*).$$
>
>    Therefore,
>    $$v_{x}:=\frac{\nabla_{x}g(x,y_{\delta}^{\ast})-\nabla_{x}g(x,y^{\ast})}{\delta}\approx \lim_{\delta\rightarrow0}\frac{\nabla_{x}g(x,y_{\delta}^{\ast})-\nabla_{x}g(x,y^{\ast})}{\delta}=\nabla_{xy}^{2}g(x,y^{\ast})\cdot\frac{dy_{\delta}^{\ast}}{d\delta}\vert_{\delta=0} = \frac{dy^\*}{dx}^{\top} \nabla_y f(x, y^\*)$$ is a valid gradient proxy. Thus, the motivation behind the perturbation is to allow for the construction of a hypergradient approximation by an application of the implicit function theorem.
>
> ---
> 8. ## Strong convexity of $\mathcal{L}$.
>
>     $\mathcal{L}$ is strongly convex when $\alpha_1 > C_f / \mu$. This may be deduced from the smoothness and strong convexity of the components of $\mathcal{L}$. We will clarify this.
>
> ---
> # Conclusion
>
> We again thank the reviewer for their effort in raising pertinent questions; we will incorporate these answers in our paper.
>
> Please let us know if any questions remain. If we adequately answered all the questions, we would like to respectfully ask the reviewer to re-consider their score. We are happy to answer any further questions.
>
> ---
> # References
>
> [1] Kornowski, G., and Shamir, O. Oracle complexity in nonsmooth nonconvex optimization. JMLR (2022).
>
> [2] Zhang, Z., & Lan, G. (2022). Solving Convex Smooth Function Constrained Optimization Is Almost As Easy As Unconstrained Optimization.

---

> > ### Comment · Reviewer_c7kZ · 2024-08-08
> > **Discussions**
> >
> > Thanks for the detailed responses. Most of my concerns are addressed. I realize that the definition for the stationary notion in the linear equality case is with exponent 2 while in the linear inequality cases is with exponent 1, making it very confusing to comprehend. The complexity bounds in the linear equality case seems indeed optimal in terms of accuracy if the definition is with exponent 2. In addition, with the perturbation methods, does the complexity bound in the linear equality constrained case depend on the dimension of the problem?
> >
> > The presentation of the work requires serious improvements.

---

> > > ### Author Response · Authors · 2024-08-09
> > > **Replying to further discussion**
> > >
> > > Thank you very much for your questions! We answer them below.
> > >
> > > ---
> > > 1. ## The exponent of stationarity.
> > >
> > >    Both stationarity notions, namely $\epsilon$-stationarity for the equality case and $(\delta, \epsilon)$-stationarity for the inequality case, are defined and addressed throughout the paper with respect to the norm with exponent 1.
> > >
> > > ---
> > > 2. ## Need for different notions of stationarity.
> > >
> > >    The need for different stationarity notions for the two settings (and hence, the different approaches) stems from the crucial difference that **the hyperobjective is not smooth in the inequality case, whereas in the equality case we prove that it is** — as a result, in the equality setting, we can resort to the simpler (more classically understood) notion of $\epsilon$-stationarity [1, 2]. Moreover, for smooth objectives, these two stationarity notions are equivalent, see e.g. Proposition 6 (ii) in [3].
> > >
> > >    In our paper, we motivate $(\delta, \epsilon)$-stationarity in the (nonsmooth) inequality setting in Lines 45-52. To improve our presentation, we will add a more thorough explanation for these two notions of stationarity in the appendix.
> > >
> > > ---
> > > 3. ## Does the linear equality algorithm incur a dimension dependence?
> > >
> > >    No, our algorithm for the linear equality setting is dimension-free (cf. Theorem 5.1). This is because, as we show, our functions in this setting are already smooth.
> > >
> > >    The purpose served by the perturbation in this setting is to help us generate a sufficiently good hypergradient approximation (as explained in our previous response). We will clarify this better in the text.
> > >
> > > ---
> > > Please feel free to let us know if we can provide any further clarifications! Thank you once again for all your questions and feedback!
> > >
> > > ---
> > > # References
> > >
> > > [1] Lan, G. (2020). First-order and stochastic optimization methods for machine learning. Springer.
> > >
> > > [2] Nesterov, Y., and Polyak, B. Cubic regularization of Newton method and its global performance. Mathematical programming (2006).
> > >
> > > [3] Zhang, J., Lin, H., Jegelka, S., Sra, S., & Jadbabaie, A. (2020, November). Complexity of finding stationary points of nonconvex nonsmooth functions. In International Conference on Machine Learning (pp. 11173-11182). PMLR.

---

> > > > ### Comment · Reviewer_c7kZ · 2024-08-09
> > > > **Discussions**
> > > >
> > > > For the linear equality case, when the hyperobjective function is smooth, as the exponent of the stationary notion (under the smooth setting) is 1, the complexity bounds $O(\epsilon^{-2})$ is optimal if considering stochastic first order algorithms. However, the paper considers deterministic first-order algorithms and thus this complexity bound $O(\epsilon^{-2})$ is no longer optimal. Could you clarify what do you mean by "Nonetheless, we wish to point out that our rate for the linear equality case is in fact optimal" in your initial responses?

---

> ### Author Response · Authors · 2024-08-09
> **Replying to further discussions**
>
> We greatly appreciate the prompt response from the reviewer and clarify our point about optimality below.
>
> ---
>
> 1. ## Optimality in the equality setting
>
>    The work of [1] showed the optimal oracle complexity (in the deterministic setting) to be $O(\epsilon^{-2})$ for unconstrained bilevel programs; our result matches this under the more complicated setting with equality constraints.
>
>
>    More specifically, for our bilevel problem with equality constraints, the upper level objective function is smooth and possibly non-convex, so it covers the more traditional *single-level deterministic smooth non-convex* optimization as a special case. Under this setting, it has been shown in [2] that the number of oracle evaluations required to find an $\epsilon$-stationary point (i.e., a point $x$ satisfying $\|\nabla f(x)\|\leq \epsilon$) is lower bounded by $O(\epsilon^{-2})$. Thus, the $O(\epsilon^{-2})$ oracle complexity we achieved is unimprovable. This point was also recently stated in the work of [3].
>
>
>    We will clarify this point better in the paper. Thank you again for your feedback, and please let us know if we can answer any further questions!
>
> ---
>
> # References
>
> [1] Chen, L., Ma, Y., & Zhang, J. (2023). Near-optimal fully first-order algorithms for finding stationary points in bilevel optimization. arXiv preprint arXiv:2306.14853.
>
> [2] Carmon, Y., Duchi, J. C., Hinder, O., & Sidford, A. (2020). Lower bounds for finding stationary points I. Mathematical Programming, 184(1), 71-120.
>
> [3] Kwon, J., Kwon, D., & Lyu, H. On The Complexity of First-Order Methods in Stochastic Bilevel Optimization. In Forty-first International Conference on Machine Learning.

---

> > ### Comment · Reviewer_c7kZ · 2024-08-13
> >
> > Thank you for the clarification. Just to be sure, [3] studied the stochastic case and their complexity bound is $O(\epsilon^{-4})$ with a corresponding lower bounds for stochastic cases.

---

> > > ### Author Response · Authors · 2024-08-13
> > > **Thank you!**
> > >
> > > We want to again thank the reviewer for their insightful review and great questions. We hope we have satisfactorily answered them.
> > >
> > > We would like to respectfully ask if, per their original message, the reviewer would consider increasing their score? Thanks again!

---

> > > > ### Comment · Reviewer_c7kZ · 2024-08-13
> > > >
> > > > The responses addressed my concerns. Please consider improve the presentation for the final version based on the discussions. I raise my score from 5 to 6.

---

### Official Review · Reviewer_GkWW · 2024-07-03

**Soundness:** 4
**Presentation:** 4
**Contribution:** 4
**Rating:** 6
**Confidence:** 3

**Summary:**

This works deals with constrained bilevel optimization problems where the lower-level has linear inequality or equality constraints. A set of algorithms is developed that does not require access to the Hessian, but only to zeroth and first-order information. In the case of inequality constraints, convergence is established to an $(\delta,\epsilon)$-Goldstein stationary point, due to the non-smoothness of the problem, using $O(\delta^{-1}\epsilon^{-4})$ or $O(d\delta^{-1}\epsilon^{-3})$ oracle calls (d: dimension of the upper-level).For equality-constrained problems, the convergence rate is nearly optimal with a rate $O(\epsilon^{-2})$.

**Strengths:**

* This work makes progress in the topic of constrained (in the lower-level) bilevel problems, which is a challenging class of problems, much more difficult than unconstrained problems.
* The development of first-order algorithms for bilevel problems with linear constraints in the lower-level. On the contrary, typically implicit gradient methods for (unconstrained or constrained) bilevel problems require access to the Hessian. In addition, for the case of inequality constraints there are no assumptions involving second-order derivatives.
* The design of algorithms that deal with the non-differentiability of the hyperobjective F.
* Some of the proposed algorithms (e.g., Algorithm 3) appears to be simple and easy to implement.

**Weaknesses:**

* A major weakness of this work is the small number of experiments. There is only a single example bilevel problem over which experiments are performed. In addition, there are no experiments on real applications (e.g., in machine learning).
* In the experiments section the proposed method is compared only with a single baseline, which does not correspond to any published bilevel method. Why aren’t there any comparisons with other bilevel algorithms that deal with constrained bilevel problems? There are both value function and implicit gradient methods that can, at least in theory, deal with the example problem used here, e.g. [37,38].
* The algorithms require access to exact solutions of certain problems, such as $y^{\ast}(x)$ of the lower-level or $y_{\lambda^{\ast},\alpha}^{\ast}(x)$ of problem (3.6). This is not the case in practice.

**Questions:**

* In the gradient of the penalty function (3.5.), there is the term $\nabla_x g^{\ast}(x)$. How do you compute this gradient given that $g^{\ast}(x)$ is a min function? The authors should explain this.
* The authors should explain in more detail Algorithm 2. It is not clear what the utility of each step is.
* The authors derive the equations for the gradient of $F$ and $y^{\ast}(x)$ (eq. 3.3). However, as they note in section 1.1, the hyperfunction $F$ is non-smooth. Then, how can we derive a formula for the gradient of $y^{\ast}(x)$? How do we determine if a given point is differentiable?
* Perhaps the least common assumption in literature is Assumption 2.3. The authors should justify whether this assumption is easy to be satisfied. For instance, is it typically the case that the dual solution is bounded? How about the Lipschitz property of $y^{\ast}(x)$? Are there other relevant (bilevel) works that use the same or similar assumption?
* How does the cvxpyLayer baseline work? This is not explained clearly in the text, besides a minor reference in the Appendix. As this is the only baseline currently used, I believe that some more details are required.

**Limitations:**

* See weaknesses above.
* The authors discuss limitations of their work in section 7.
* No negative societal impact

---

> ### Author Rebuttal · Authors · 2024-08-06
>
> We are extremely grateful to the reviewer for their thoughtful questions and are encouraged by their positive assessment of our contributions, soundness, and presentation.
>
> ---
> # Response to Weaknesses
> ---
> 1. ## Limited experiments.
>
> We acknowledge the importance of testing our algorithms with large-scale experiments. However, our focus has primarily been theoretical since prior work had far weaker theoretical guarantees for our choice of stationarity and oracle.
>
> In the code, we also have results on bilevel problems with bilinear constraints (in figures/exp_bilinear). Per reviewer pjir's suggestion, we have now implemented additional experiments for Algorithm 2 (please see the top-level response), which we will add to our paper. In follow-up work, we hope to expand our implementation to a wider range of problems and large-scale experiments.
>
> ---
> 2. ## Access to $y^\*$
>
> Our algorithm for the inequality setting indeed requires access to $\lambda^\*$ (the lower-level dual solution). In practice, though, we can easily approximate $\lambda^\*$ to high accuracy due to the linear convergence of strongly convex problems with linear constraints --- our experiments demonstrate our algorithm's robustness to the use of such approximations as proxy for the exact solution. That said, we believe that developing an algorithm for this problem without this assumption is an important direction for future work.
>
> ---
> # Response to Questions
>
> ---
> 1. ## Computing $\nabla_x g^*(x)$
>
> We compute this gradient as
> $\nabla_x g(x,y^\*) + \nabla_x h(x,y^\*)^\top \lambda^\*$.
>
> For some intuition, note that $g^\*(x) := \min_y \max_{\lambda \geq 0} g(x,y) + \lambda^\top h(x,y),$ to which applying Theorem 4.13 from [1] immediately yields the claim. We will add this explanation in the text.
>
> ---
> 2. ## Explaining Algorithm 2
>
> Algorithm 2 is a variant of **gradient descent with momentum and clipping**, where $\tilde{g}\_{t}$ is the inexact gradient, $\Delta\_{t}$ is a clipped accumulated gradient (hence accounts for past gradients, which serve as a momentum), and the main iterate is updated as $x_{t+1}=x_t+\Delta_t$. Similar algorithms have appeared in prior work on nonsmooth nonconvex optimization (e.g. [2]). However, none of them accounted for _inexactness_ in the gradient, crucial in the bilevel setting. We will add this in the final version.
>
> ---
> 3. ## Computing $\nabla y^*(x)$
>
> Our algorithm does not require actually computing $\nabla y^*(x)$. The only place where $\nabla y^*(x)$ appears is in our analysis when bounding the difference between our inexact and the actual hypergradients. We use $\nabla y^*(x)$ to mean a Clarke subgradient of $y^{\ast}$. We will clarify this better.
>
> ---
> 4. ## Assumption 2.3
>
> We acknowledge that this assumption is somewhat technical — however, prior work also imposes similar assumptions. For instance, Khanduri et al [3] assume (weak/strong) convexity of the *hyperobjective* $F$ to obtain finite-time guarantees --- our assumption is strictly weaker. Further, Yao et al [4] also assume boundedness of the optimal dual variable.
>
> As to $y^\*(x)$ being Lipschitz, there are some useful settings with this property: e.g., if the lower level feasible region $Y(x):=\\{y: h(x,y) \leq 0\\}$ is independent of $x$, an argument in [5] shows it to be $\frac{C_g}{\mu_g}$ Lipschitz. The situation is more complicated when $Y(x)$ changes with $x$. If $g(x, y)$ is quadratic in $(x,y)$ (e.g., in model predictive control),  $y^\*(x)$ is piecewise affine and Lipschitz [6]. For more general settings, this is related to sensitivity analysis, and some constraint qualifications can establish the property [7].
>
> ---
> 5. ## cvxpyLayer
>
> cvxpylayer is a Python library [8, 9] that uses a second-order method to compute the derivative of the optimal solution $y^*(x)$ of a parametric convex program with respect to the parameter $x$. We invoke cvxpylayer on the lower-level problem (which is parametrized by $x$) to compute $\frac{dy^*(x)}{dx}$. We will add more details in the text.
>
> ---
> # Conclusion
>
> We again thank the reviewer for their time and effort in bringing up pertinent questions; we will incorporate these answers in our paper.
>
> Please let us know if any questions remain. If we adequately answered all the questions, we would like to respectfully ask the reviewer to re-consider their score. We are happy to answer any further questions.
>
> ---
> # References
>
> [1] Bonnans, J. F., & Shapiro, A. (2013). Perturbation analysis of optimization problems.
>
> [2] Cutkosky, A., Mehta, H., & Orabona, F. (2023). Optimal stochastic non-smooth non-convex optimization through online-to-non-convex conversion. ICML
>
> [3] Khanduri, P., Tsaknakis, I., Zhang, Y., Liu, J., Liu, S., Zhang, J., & Hong, M. (2023). Linearly constrained bilevel optimization: A smoothed implicit gradient approach. ICML
>
> [4] Yao, W., Yu, C., Zeng, S., & Zhang, J. (2024) Constrained Bi-Level Optimization: Proximal Lagrangian Value Function Approach and Hessian-free Algorithm. ICLR.
>
> [5] Nesterov, Yu. Smooth minimization of non-smooth functions. Mathematical programming (2005).
>
> [6] Borrelli, F., Bemporad, A., & Morari, M. (2017). Predictive control for linear and hybrid systems.
>
> [7] Minchenko, L. I., and P. P. Sakolchik. Hölder behavior of optimal solutions and directional differentiability of marginal functions in nonlinear programming. Journal of optimization theory and applications (1996).
>
> [8] Amos, Brandon, and J. Zico Kolter. Optnet: Differentiable optimization as a layer in neural networks. ICML, 2017.
>
> [9] Agrawal, A., Amos, B., Barratt, S., Boyd, S., Diamond, S., & Kolter, J. Z. (2019). Differentiable convex optimization layers. Advances in neural information processing systems.

---

> > ### Comment · Reviewer_GkWW · 2024-08-10
> > **Comment by Reviewer**
> >
> > My main concern during the review was the limited number of experiments. The authors have addressed this issue by providing additional experiments. Given that this is primarily a theoretical work, I also understand the current absence of large-scale experiments.
> >
> > I am raising my score to 6.

---

### Official Review · Reviewer_pjir · 2024-07-09

**Soundness:** 3
**Presentation:** 4
**Contribution:** 3
**Rating:** 7
**Confidence:** 3

**Summary:**

The paper provides algorithms for bilevel optimization with linear equality and inequality constraints. The main contribution is that the algorithms are fully first order and do not require Hessian computations. This is achieved by reformulating the linearly constrained bilevel optimization problem using the penalty method and assuming access to the upper-level variable $x$ and approximate dual optimal solution. In this setting an inexact gradient oracle for the problem is constructed. The authors provide stationarity guarantees for their algorithms under certain standard assumptions. They also describe how to use their inexact gradient oracle for non-convex non smooth optimization. The algorithms and theoretical results are backed by small set of proof-of-concept experiments.

**Strengths:**

1) Typically, hessian inverse computation is quite expensive and hence coming up with a fully first order method with guarantees for an optimization problem has both good theoretical and practical significance. The paper will be of interest to the community.
2) The paper is written very nicely. There is a lot of clarity regarding the related works, notations, main contributions and techniques. Proof sketches are also provided for some results in the main paper. Though the paper is heavy on technical material, the organization makes it somewhat easier for the reader to follow the arguments. I am not exactly from the same research area but could follow most of the paper.
3) I could not check all the proofs in details, but the main claims of the paper appear correct.
4) Nonconvex non smooth optimization is encountered in many ML problems these days. The applicability of the inexact gradient oracle to these problems with guarantees may be of practical significance.

**Weaknesses:**

1) There are no experiments highlighting the effectiveness of Algorithm 2.
2) The experiments given here are also only of a proof -of -concept nature and are not comprehensive. Some more comparisons with existing algorithms (based on time) and ablation studies may provide more insights regarding the practical applicability of the methods.

**Questions:**

There is a lot of literature available on approximating Hessian computations faster for 2nd order methods (for example using iterative sketching etc.) though not necessarily for bilevel optimization. I would appreciate it if you can comment on such methods when compared to yours in terms of practicality.

**Limitations:**

Yes, the limitations have been clarified in the paper.

---

> ### Author Rebuttal · Authors · 2024-08-06
>
> We are very grateful to the reviewer for their effort in reviewing our submission and are encouraged by their positive assessment of the theory, presentation, and significance of our work.
>
> ---
> # Response to Weaknesses
>
> ---
> 1. ## Implementing Algorithm 2
>
> In our paper, we implemented the simpler algorithm (Algorithm 3). Per the reviewer's suggestion, we additionally implemented Algorithm 2 and repeated our experiments --- we have added the results in the pdf with our top-level response. These experiments show that Algorithm 2 outperforms both Algorithm 3 and the baseline cvxpylayer in terms of convergence, as suggested by our theory (see Theorem 4.2 and Theorem 4.3). We appreciate the reviewer’s suggestion and will update the paper with these experiments.
>
> ---
> 2. ## Large-scale implementation
>
> We acknowledge the importance of testing our algorithms with large-scale experiments. However, our focus has primarily been theoretical since prior work had far weaker theoretical guarantees for our choice of stationarity and oracle. As the reviewer notes, our experiments indeed mainly provide a proof of concept for our algorithms.
>
> In the code, we also have results on bilevel problems with bilinear constraints (in figures/exp_bilinear); in follow-up work, we hope to expand our implementation to a wider range of problems and large-scale experiments.
>
> ---
> # Response to Questions
>
> ---
> 1. ## Hessian approximations
>
> The key step in algorithms for bilevel programming is that of computing $\frac{d y^\*}{dx}$, which in turn is composed of the product of a matrix, an inverse Hessian, and vector --- this therefore is the primary computational bottleneck in these algorithms. As the reviewer suggests, there have been many approaches for Hessian approximations, including in the context of bilevel programming.
>
> For instance, [1] uses conjugate gradient to compute the Hessian-inverse-vector product, [2] uses Neumann approximation (essentially a geometric series for approximating matrix inverse) to approximate the inverse Hessian, and [3] uses a method inspired by Gauss-Newton to approximate the Hessian as the outer product of the corresponding gradients. However, all these works deal with unconstrained bilevel programming. In constrained settings (e.g., the ones we consider), the Hessian becomes significantly more complicated, and it is unclear if these techniques would apply to them (but would, nevertheless, be an interesting question to consider).
>
> Another line of Hessian-free approaches by [4-6] uses value-function reformulation to circumvent the use of Hessians. Finally, we also mention the approach of [7], which approximates the Hessian-inverse-vector product by a linear system solver in conjunction with a finite difference. As we detail in our response to reviewer wH2u, the runtime of our algorithm (when applied in the unconstrained setting) improves upon that of this algorithm by [7]. It would be interesting to extend these approaches to the constrained setting we consider with our choice of stationarity and oracle access.
>
>
> ---
> # Conclusion
>
> We again thank the reviewer for their time and effort in raising thoughtful questions. Please let us know if there are any further questions that we could help clarify!
>
> ---
> # References
>
> [1] Pedregosa, F. (2016). Hyperparameter optimization with approximate gradient. In International conference on machine learning. PMLR.
>
> [2] Lorraine, J., Vicol, P., & Duvenaud, D. (2020). Optimizing millions of hyperparameters by implicit differentiation. In International conference on artificial intelligence and statistics. PMLR.
>
> [3] Giovannelli, T., Kent, G., & Vicente, L. N. (2021). Bilevel stochastic methods for optimization and machine learning: Bilevel stochastic descent and darts. arXiv preprint arXiv:2110.00604.
>
> [4] Liu, B., Ye, M., Wright, S., Stone, P., & Liu, Q. (2022). Bome! bilevel optimization made easy: A simple first-order approach. Advances in neural information processing systems.
>
> [5] Kwon, J., Kwon, D., Wright, S., & Nowak, R. D. (2023). A fully first-order method for stochastic bilevel optimization. In International Conference on Machine Learning. PMLR.
>
> [6] Yao, W., Yu, C., Zeng, S., & Zhang, J. Constrained Bi-Level Optimization: Proximal Lagrangian Value Function Approach and Hessian-free Algorithm. In The Twelfth International Conference on Learning Representations.
>
> [7] Yang, Y., Xiao, P., & Ji, K. (2023). Achieving O (ε-1.5) complexity in hessian/jacobian-free stochastic bilevel optimization. In Proceedings of the 37th International Conference on Neural Information Processing Systems.

---

> > ### Comment · Reviewer_pjir · 2024-08-09
> >
> > I have read the rebuttal and the reviews. As of now I will keep my score.

---

### Official Review · Reviewer_wH2u · 2024-07-12

**Soundness:** 3
**Presentation:** 3
**Contribution:** 3
**Rating:** 6
**Confidence:** 2

**Summary:**

This paper studies the linearly constrained bilevel optimization problem and provides a fully first-order method with solid theoretical analysis. To approximate hypergradient, the penalty method seems novel to me and it is applied in two settings where the LL problem is linear inequality or equality constraints.

**Strengths:**

1. The presentation is pretty good and I feel easy to follow.
2. Theoretical analysis is solid and rigorous.
3. Code is provided.

**Weaknesses:**

1. The experiment does not seem sufficient.
2. The checklist should be behind the appendix as I memorize.

**Questions:**

1. Could the authors explain the design of Algorithm 2? Why is there a clip?
2. Why do authors not include the experiment of Algorithm 4?
3. Could authors compare the method in [a] though it studies the unconstrained bilevel optimization problem?
[a] Yang Y, Xiao P, Ji K. Achieving ${O}(\epsilon^{-1.5}) $ Complexity in Hessian/Jacobian-free Stochastic Bilevel Optimization.

**Limitations:**

Please check weaknesses and problems.

---

> ### Author Rebuttal · Authors · 2024-08-06
>
> We are very grateful to the reviewer for their effort in reviewing our submission and are encouraged by their positive assessment of our theory and presentation.
>
> ---
> # Response to Questions
>
>
> ---
> 1. ## Intuition behind clipping
>
> Intuitively, the clipping ensures that consecutive iterates of the algorithm **reside within a $\delta$-ball** of each other — this in turn allows for a guarantee in terms of $(\delta, \epsilon)$-Goldstein stationarity (which, as we recall, is defined in terms of gradients in a $\delta$-ball). We remark that following this intuition, all previous papers on Goldstein stationarity, as discussed in the paper, also utilize normalized and/or clipped steps (a notable exception is [1], where the clipping was dropped by relaxing the stationarity notion accordingly).
>
> ---------------------------------------------------------------
>
> 2. ## Implementation
>
>  We acknowledge that the implementation and testing of our algorithm for the setting in question is an extremely important task. That said, our focus in this section was primarily theoretical since before our result, there did not exist any work at all with finite-time theoretical guarantees using a first-order oracle for the problem in question. We hope to implement our algorithm and perform large-scale experiments in follow-up works.
>
> ---------------------------------------------------------------
>
>
> 3. ## Comparison with Yang-Xiao-Ji [2]
>
> The core idea in both our paper and that by Yang-Xiao-Ji (and indeed, by all papers on bilevel optimization) is an efficient approximation of $\frac{dy^\ast}{dx}^{\top} \nabla_y f(x, y^\ast)$. Recall that this term simplifies to $$\frac{dy^\ast}{dx}^{\top} \nabla_y f(x, y^\ast)=-\nabla_{xy}^{2}g(x,y^{\ast})\cdot\nabla_{yy}^{2}g(x,y^{\ast})^{-1}\nabla_{y}f(x,y^{\ast}).$$
>
> The way the paper of Yang-Xiao-Ji handles this term is by **separately approximating parts** of this product: in particular, it approximates
> $\nabla_{xy}^{2}g(x,y^{\ast})$ via a finite-difference method and
> the term $\nabla_{yy}^{2}g(x,y^{\ast})^{-1}\nabla_{y}f(x,y^{\ast})$
> by a linear system solver.
>
> In contrast to this method, our paper **approximates the entire term**  via a novel application of the implicit function theorem. We now elaborate on this point by illustrating our method for the unconstrained setting.
>
> As stated in our Equation (5.2), our approximation (for this unconstrained setting) of the term $\frac{dy^\ast}{dx}^{\top} \nabla_y f(x, y^\ast)$ is
> $v_{x}:=\frac{\nabla_{x}g(x,y_{\delta}^{\ast})-\nabla_{x}g(x,y^{\ast})}{\delta}$. To see the validity of this approximation, we first note that $\lim_{\delta\rightarrow0}\frac{\nabla_{x}g(x,y_{\delta}^{\ast})-\nabla_{x}g(x,y^{\ast})}{\delta}=\nabla_{xy}^{2}g(x,y^{\ast})\cdot\frac{dy_{\delta}^{\ast}}{d\delta}\vert_{\delta=0}$.
> Next, to approximate $\frac{dy_{\delta}^{\ast}}{d\delta}\vert_{\delta=0},$ we
> observe that $y_{\delta}^{\ast}:=\text{argmin} (g(x,y)+\delta f(x,y)).$
> Applying first-order optimality of $y_{\delta}^{\ast}$ and
> taking the derivative with respect to $\delta$ yields $\frac{dy_{\delta}^{\ast}}{d\delta}\vert_{\delta=0}=-\left(\nabla_{yy}^{2}g(x,y^{\ast})\right)^{-1}\nabla_{y}f(x,y^{\ast}).$ Combining this with the first step proves the claimed approximation.
>
> Finally, when measured according to the stationarity criterion of Yang-Xiao-Ji (in the unconstrained setting), our work's **oracle complexity** is $O(\epsilon^{-1})$ (improving upon their result of $O(\epsilon^{-1.5})$).
>
> ----------------------------------
>
> # Conclusion
>
> We again thank the reviewer for their time and effort. Please let us know if there are any further questions that we could help clarify!
>
> ---------------------------------------
>
> # References
>
> [1] Zhang, Qinzi, and Ashok Cutkosky. "Random scaling and momentum for non-smooth non-convex optimization." arXiv preprint arXiv:2405.09742 (2024).
>
> [2] Yang, Y., Xiao, P., & Ji, K. (2023). Achieving O (ε-1.5) complexity in hessian/jacobian-free stochastic bilevel optimization. In Proceedings of the 37th International Conference on Neural Information Processing Systems.

---

> > ### Comment · Reviewer_wH2u · 2024-08-09
> >
> > Thank you for answering my questions and they are resolved. I would like to keep my score.

---

### Author Rebuttal · Authors · 2024-08-06

We are grateful to all the reviewers for well-thought-out reviews of our submission, appreciation of our ideas, and constructive efforts in helping us strengthen our work. We address all questions and remarks by responding directly to each review (and will incorporate all suggestions). Here we briefly reiterate our key contributions.

---
# Contributions

1. For linear-equality-constrained bilevel problems, we provide a first-order method via a novel application of the implicit function theorem. Our rate of convergence for this problem is optimal.

2. For linear-inequality-constrained bilevel problems, assuming access to the optimal dual variable, we provide a first-order method with finite-time guarantees on convergence to a Goldstein stationary point of the hyperobjective.

3. Along the way, we design an algorithm for nonsmooth nonconvex optimization with an inexact gradient oracle, which could be of independent interest.

---
# Additional Experiments

Following reviewer pjir's suggestion, we implemented our algorithm for nonsmooth nonconvex optimization with an inexact gradient oracle (Algorithm 2). As suggested by our theory, Algorithm 2 outperforms both Algorithm 3 and the baseline cvxpylayer in terms of convergence rate. We provide these results in the attached PDF and will include them in our paper.

In the code, we also have results on bilevel problems with bilinear constraints (in figures/exp_bilinear). Overall, our experiments mainly provide a proof of concept for our algorithms. In follow-up work, we hope to expand our implementation to a wider range of problems and large-scale experiments.


---

---

### Decision · Program_Chairs · 2024-09-25

**Decision:**

Accept (poster)

**Comment:**

This work studies constrained bilevel optimization problems, for which it presents fully first-order methods (thereby avoiding Hessian computations). There is overall consensus among the reviewers that this is an interesting work that is worthy of acceptance, and I am inclined to agree. It would be strongly suggested that, for the camera-ready version, the authors clarify the presentation/conventions of the convergence bounds, in order to better facilitate comparison with existing upper and lower bounds.